JCB Journal of Cell Biology

# *Drosophila* SPG12 ortholog, reticulon-like 1, governs presynaptic ER organization and Ca²⁺ dynamics

Juan José Pérez-Moreno[1]*, Rebecca C. Smith[1]*, Megan K. Oliva[1], Filomena Gallo[2], Shainy Ojha[1], Karin H. Müller[2], and Cahir J. O'Kane[1]

Neuronal endoplasmic reticulum (ER) appears continuous throughout the cell. Its shape and continuity are influenced by ER-shaping proteins, mutations in which can cause distal axon degeneration in Hereditary Spastic Paraplegia (HSP). We therefore asked how loss of *Rtnl1*, a *Drosophila* ortholog of the human HSP gene *RTN2* (*SPG12*), which encodes an ER-shaping protein, affects ER organization and the function of presynaptic terminals. Loss of Rtnl1 depleted ER membrane markers at *Drosophila* presynaptic motor terminals and appeared to deplete narrow tubular ER while leaving cisternae largely unaffected, thus suggesting little change in resting Ca²⁺ storage capacity. Nevertheless, these changes were accompanied by major reductions in activity-evoked Ca²⁺ fluxes in the cytosol, ER lumen, and mitochondria, as well as reduced evoked and spontaneous neurotransmission. We found that reduced STIM-mediated ER-plasma membrane contacts underlie presynaptic Ca²⁺ defects in *Rtnl1* mutants. Our results show the importance of ER architecture in presynaptic physiology and function, which are therefore potential factors in the pathology of HSP.

## Introduction

Synapses, the computational units of nervous systems, are immensely adaptable. The ER is a site of some of the processes that tune synaptic strength. It is a major Ca²⁺ store of the cell, a site of lipid biosynthesis, and can regulate the physiology of other organelles through contact sites that support exchange of Ca²⁺ and lipids. Axonal ER comprises an extensive continuous network formed mainly of tubules (Wu et al., 2017) that are shaped by proteins of the reticulon and REEP families, among others. These contain intramembrane hairpin domains that can curve ER membrane (Shibata et al., 2009; Voeltz et al., 2006); loss of these proteins can reduce tubule number and curvature of axonal ER and disrupt network continuity (O'Sullivan et al., 2012; Yalçın et al., 2017).

The ER network appears to influence axon survival and maintenance; mutations affecting ER-shaping proteins can lead to the axon degeneration disease, Hereditary Spastic Paraplegia (HSP; Öztürk et al., 2020). HSPs show degeneration of the distal regions of longer upper motor axons, consistent with "dying back" degeneration from axon terminals, preferentially in the longest upper motor axons. The links between HSP and ER-shaping proteins suggest a critical relationship between ER architecture and axonal and presynaptic maintenance. However, we lacked a good model for how ER spatial organization affects axonal or presynaptic function and degeneration. One way in which ER might do this is as an intracellular Ca²⁺ store. ER Ca²⁺ stores can either contribute to presynaptic cytosolic Ca²⁺ responses or sequester excess Ca²⁺ from the cytosol in response to high frequency stimulation (Chanaday et al., 2021; de Juan-Sanz et al., 2017; Kuromi and Kidokoro, 2002; Lindhout et al., 2019; Sanyal et al., 2005; Singh et al., 2021; Tran and Stricker, 2021). ER might potentially also influence presynaptic physiology via its interactions with mitochondria. Like ER, mitochondria can also take up Ca²⁺ when cytosolic Ca²⁺ is elevated, although this process appears unable to buffer Ca²⁺ sufficiently to affect synaptic transmission (Chouhan et al., 2010; Verstreken et al., 2005); rather its role appears to be to stimulate ATP synthesis in response to the Ca²⁺ influx that occurs on synaptic activity (Chouhan et al., 2012; Datta and Jaiswal, 2021; Jouaville et al., 1999). The cytosol, rather than ER, seems to be the main source of Ca²⁺ for mitochondria in presynaptic terminals (Ashrafi et al., 2020; Chouhan et al., 2012).

[1]Department of Genetics, University of Cambridge, Cambridge, UK; [2]Development and Neuroscience, Cambridge Advanced Imaging Centre, Cambridge, UK.

*J.J. Pérez-Moreno and R.C. Smith contributed equally to this paper. Correspondence to Juan José Pérez-Moreno: jpmoreno@us.es; Cahir J. O'Kane: c.okane@gen.cam.ac.uk

J.J. Pérez-Moreno's current affiliation is Instituto de Biomedicina de Sevilla (IBiS), Hospital Universitario Virgen Del Rocío/CSIC/Universidad de Sevilla, and Departamento de Biología Celular, Facultad de Biología, Universidad de Sevilla, Seville, Spain. R.C. Smith's current affiliation is Laboratory of Neural Genetics and Disease, Brain Mind Institute, EPFL — Swiss Federal Institute of Technology Lausanne, Lausanne, Switzerland. M.K. Oliva's current affiliation is Ion Channels and Disease Group, The Florey Institute of Neuroscience and Mental Health, University of Melbourne, Parkville, Australia.

As well as being a bulk $Ca^{2+}$ source or sink, the ER dynamically responds to lumenal $Ca^{2+}$ levels. STIM (stromal interacting molecule) proteins in the ER membrane, when activated by depletion of ER $Ca^{2+}$ stores, interact with effectors in the plasma membrane (PM), such as the Orai1 $Ca^{2+}$ channel, leading to influx of extracellular $Ca^{2+}$ into the cytosol (store-operated $Ca^{2+}$ entry, SOCE) and refilling of ER stores (Serwach and Gruszczynska-Biegala, 2020). In primary hippocampal neurons, evoked neurotransmitter release correlates with ER $Ca^{2+}$ content due to a feedback loop dependent on inhibition of voltage-gated $Ca^{2+}$ channels in the plasma membrane by activated STIM1 (de Juan-Sanz et al., 2017). Also in primary hippocampal neurons, SOCE, in a STIM2-dependent manner, can transiently increase presynaptic $Ca^{2+}$ levels and robustly augment spontaneous glutamate vesicular release (Chanaday et al., 2021).

Since mutations in several ER-shaping proteins can cause axon degeneration, altered ER architecture may elicit degenerative changes in axonal or presynaptic function. One mechanism could be local dysregulation of $Ca^{2+}$—for example through reduced availability of $Ca^{2+}$ for synaptic signaling or capacity to remove cytosolic $Ca^{2+}$, or interference with regulatory effects of the ER STIM $Ca^{2+}$ sensors (Chanaday et al., 2021; de Juan-Sanz et al., 2017). Presynaptic ER comprises a network of interconnected tubules with occasional cisternae (Wu et al., 2017). While the presence of cisternae might indicate regions specialized in storing $Ca^{2+}$, the predominance of narrow ER tubules suggests a local regulatory role for this organelle. To test how spatial organization of ER could impair synaptic function, we used *Drosophila* to study the effects of removing the tubular ER-shaping protein Rtnl1 on presynaptic ER organization, and on the $Ca^{2+}$ fluxes between compartments during synaptic activity. *Rtnl1* is one of two documented reticulons in *Drosophila*, orthologous to the four RTNs found in mammals (RTN1-4), including the HSP causative gene RTN2 (Montenegro et al., 2012). It is distributed continuously in motor axons all the way to presynaptic termini (O'Sullivan et al., 2012), and its loss leads to partial depletion of axonal ER (O'Sullivan et al., 2012; Yalçın et al., 2017). Loss of Rtnl1 greatly decreases Excitatory Junction Potential (EJP) amplitude at the larval neuromuscular junction (NMJ; Summerville et al., 2016), implying major effects on presynaptic physiology, although its effects on presynaptic ER and its role in presynaptic physiology has not been well defined.

Here we reveal how loss of Rtnl1 depletes tubular ER but not cisternae or ER volume at NMJs, and how this loss significantly decreases ER, mitochondrial, and cytosolic evoked $Ca^{2+}$ fluxes. We found that this impact of Rtnl1 loss on presynaptic $Ca^{2+}$ handling is caused by a decrease of STIM-mediated ER-PM contacts. Our results suggest a feedback loop, whereby tubular ER and inactivated STIM control evoked $Ca^{2+}$ entry into the presynaptic compartment and neurotransmitter release.

While Rtnl1 loss reduces axonal ER levels mainly in longer motor axons (O'Sullivan et al., 2012; Yalçın et al., 2017), we observed that it reduces presynaptic tubular ER independently of axonal length. This phenotype provides a unique model to study the contribution of the presynaptic tubular ER network and ER-shaping HSP proteins to presynaptic $Ca^{2+}$ handling. Our

findings display a local role for presynaptic ER tubules, and how synaptic dysfunction might be a consequence of impairing functions of proteins that are mutated in HSP.

## Results

### *Rtnl1^18* is a null mutant allele

To understand how Rtnl1 contributes to presynaptic ER organization and physiology, we generated new mutant alleles of *Rtnl1*. We had previously analyzed a loss-of-function allele, *Rtnl1^1* (Wakefield and Tear, 2006), with an internal deletion of around 5 kb (Yalçın et al., 2017) and obtained ER phenotypes that were broadly consistent with those of RNAi knockdown, but obtained only partial rescue using a genomic *Rtnl1* construct (O'Sullivan et al., 2012; Yalçın et al., 2017). Since additional alleles could allow us to exclude phenotypes due to second-site mutations, we generated new CRISPR/Cas9 mutant *Rtnl1* alleles. We targeted these using two *gRNAs*, each just upstream of the sequence encoding each intramembrane domain, present in all splice variants (Figs. 1 and S1). We characterized three new alleles; one of these, *Rtnl1^18*, had multiple changes: an inversion of the region between both *gRNA* sequences, a 6-bp deletion at the site of the upstream *gRNA*, and a 19-bp deletion at the site of the downstream *gRNA* (Fig. 1 and Fig. S1 A). This lesion inverts the sequence encoding the first intramembrane domain; and it deletes some of the sequence encoding the second intramembrane domain, leaving only the C-terminal half of this domain potentially able to be translated from an in-frame *AUG* codon (Fig. S1 B). While *Rtnl1^18* does not abolish *Rtnl1* transcription (Fig. S2), it is likely to be a null allele, since the rearrangements downstream of the first *gRNA* site make it impossible to express more than a fragment of the protein, which lacks an intact intramembrane domain.

### *Rtnl*1 loss-of-function decreases presynaptic ER membrane labeling

We next tested *Rtnl1* mutant NMJs for altered ER distribution. We expressed tdTom::Sec61β, which labels presynaptic ER (Summerville et al., 2016), using *GAL4* drivers specific for each main type of larval excitatory motor neuron (MN), Ib and Is (Pérez-Moreno and O'Kane, 2019), which differ structurally (Menon et al., 2013) and physiologically (Aponte-Santiago et al., 2020; Aponte-Santiago and Littleton, 2020; Chouhan et al., 2010). Compared with *wild-type (WT)* NMJs of the same genetic background, *Rtnl1^18* showed decreased tdTom::Sec61β levels in both Type Ib (Fig. 2 A) and Is (Fig. 2 B) NMJs. This decrease was not due to lowered *GAL4*-dependent expression; the plasma membrane marker, CD4::tdGFP (Han et al., 2011), expressed at NMJs using the same *GAL4* lines, was not affected by *Rtnl1^18*. We found a similar reduction in tdTom::Sec61β using a general MN driver, *D42-GAL4* (Sanyal, 2009) in *Rtnl1^18* NMJs (Fig. 2 C).

Loss of *Rtnl1* reduces ER levels specifically in distal long motor axons (O'Sullivan et al., 2012; Yalçın et al., 2017); however, lowered presynaptic tdTom::Sec61β levels in *Rtnl1^18* NMJs were independent of motor axon length (Fig. 2, A and B). *Rtnl1^18* presynaptic terminals still showed an interconnected and mostly continuous ER network (insets in Fig. 2 A and Fig. S3 A), with a distribution resembling that of *WT*. These findings suggest that

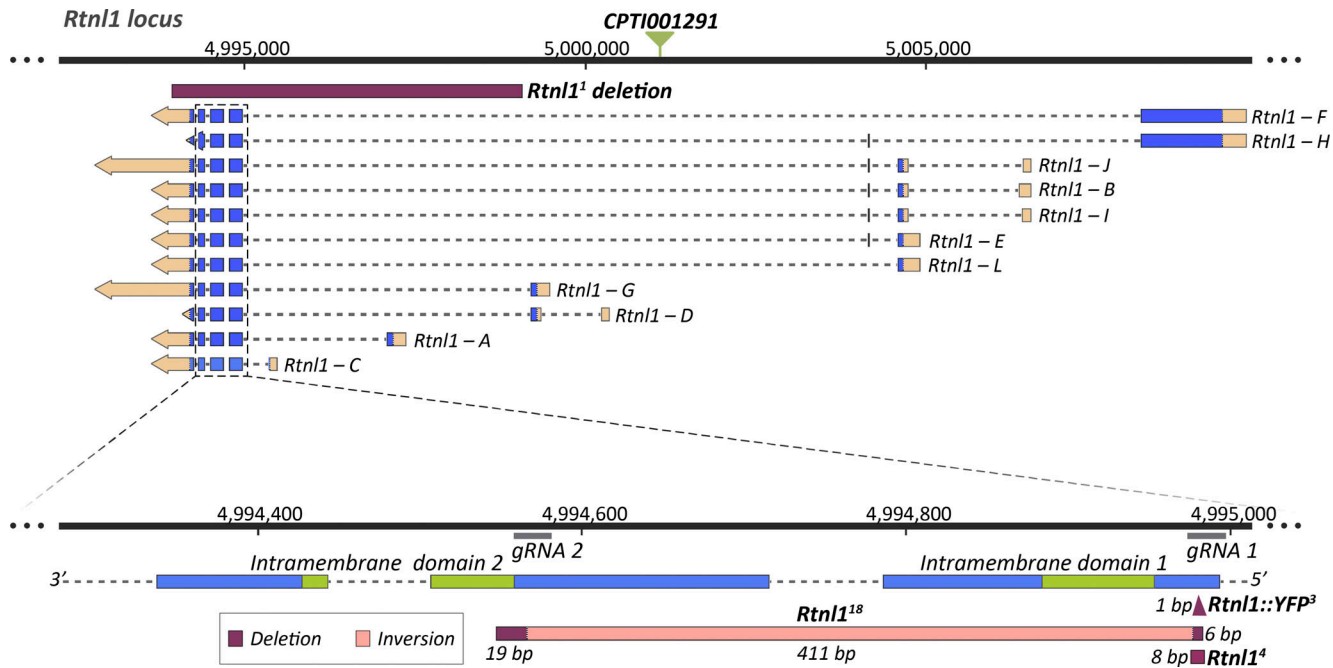

Figure 1. **Generation of new *Rtnl1* null mutant alleles.** For the different *Rtnl1* splice variants, blue boxes indicate coding sequence, broken lines introns, light orange boxes UTR sequences, and arrows the direction of transcription. Green triangle indicates the position of the *Rtnl1::YFP* exon trap insertion (*CPTI001291*). Genomic coordinates are from Release 6.26 of the *Drosophila melanogaster* genome (http://flybase.org). Magnified detail shows exons encoding Rtnl1 intramembrane domains (green; bottom), locations recognized by gRNAs, the *Rtnl1^18* lesion, and the 8-bp (*Rtnl1^4*) and the 1-bp (*Rtnl1::YFP^3*) frameshift deletions, generated at the site of the upstream gRNA on *Rtnl1+* and *Rtnl1^CPTI001291* backgrounds respectively, but not used further in this work. The length of each mutated region is shown. In *Rtnl1^18*, the region encoding the first intramembrane domain is completely disrupted by an inversion, while the initial part of the sequence encoding the second intramembrane domain is absent due to a deletion and frameshift (see Fig. S1 for more molecular details of alleles).

loss of *Rtnl1* leads to less presynaptic ER membrane, but that its general organizational features are unaffected. The phenotype appears to affect ER membrane in general rather than tdTom::Sec61β in particular; a second ER membrane marker, Sturkopf::GFP (Thiel et al., 2013; Yalçın et al., 2017), also showed decreased levels in *Rtnl1^18* NMJs (Fig. 2 D), and an apparently normal ER network organization (Fig. 2 E and Fig. S3 B). Type Is terminals, but not Type Ib, showed occasional apparent gaps in the ER network (Fig. 2 B) in both WT and *Rtnl1^18* NMJs, perhaps due to having a smaller ER network that is more sensitive to partial loss of tubules.

The partial depletion of ER membrane was specifically due to loss of *Rtnl1*. First, we also observed it at Type Ib and Type Is NMJs in a second mutant allele, *Rtnl1^1* (Fig. S4 A). Second, the expression of *UAS-Rtnl1::HA* (Summerville et al., 2016) under GAL4 control fully rescued tdTom::Sec61β levels in *Rtnl1^18* NMJs (Fig. 2 C). Surprisingly, a *UAS-Rtnl1::GFP* construct (Rao et al., 2016), appeared to be dominant negative; it produced a similar tdTom::Sec61β decrease as *Rtnl1* loss-of-function, and even made a *Rtnl1* loss-of-function phenotype slightly more severe (Fig. S4 B). Taken together, our results in a variety of genotypes and with two different markers support the conclusion that *Rtnl1* controls presynaptic ER levels.

### *Rtnl1* loss-of-function specifically decreases presynaptic tubular ER network

The reduced levels of ER membrane markers in *Rtnl1* mutants contrasted with the unchanged resting fluorescence of the $Ca^{2+}$ sensor GCaMP6-210 in the ER lumen, on loss of multiple ER-

shaping proteins including Rtnl1 (Oliva et al., 2020). While the fluorescence of GCaMP6-210 is also $Ca^{2+}$-dependent, we hypothesized that the role of reticulons in generating highly curved ER tubules (Voeltz et al., 2006) might explain the apparent discrepancy by preferentially removing ER tubules compared to cisternae, due to their greater curvature, in *Rtnl1* mutant NMJs. Similar to mammalian neurons (Wu et al., 2017), we observed both ER tubules with a lumen too small to be visible (as reported previously in axons [Terasaki, 2018; Yalçın et al., 2017]) and cisternae with a larger lumen, in presynaptic motor terminals, using electron microscopy (Fig. 3 A). Tubules were observed most clearly in regions of the bouton where synaptic vesicles (SVs) were sparse, but harder to distinguish from SVs in regions of the bouton that were rich in SVs due to the similar diameters of ER tubules and SVs, thus preventing reconstruction of the whole ER network. Using the ER lumen marker BiP::sfGFP::HDEL (Summerville et al., 2016), hereafter referred to as GFP::HDEL, we observed larger and brighter puncta that we interpret as cisternae with a large lumen, joined by stretches of fainter signal that we interpret as ER tubules with a narrow lumen, that we observed using electron microscopy (Fig. 3, B–D). These bright puncta often, but not always, overlap with puncta of tdTom::Sec61β (Fig. S5 A), suggesting that enrichment of these two markers highlights different features of ER that are sometimes found together—indeed, electron microscopy of axonal ER shows both cisternae (with a noticeable lumen) and small sheets (without a noticeable lumen) that are sometimes, but not always, apposed to each other (Yalçın et al., 2017).

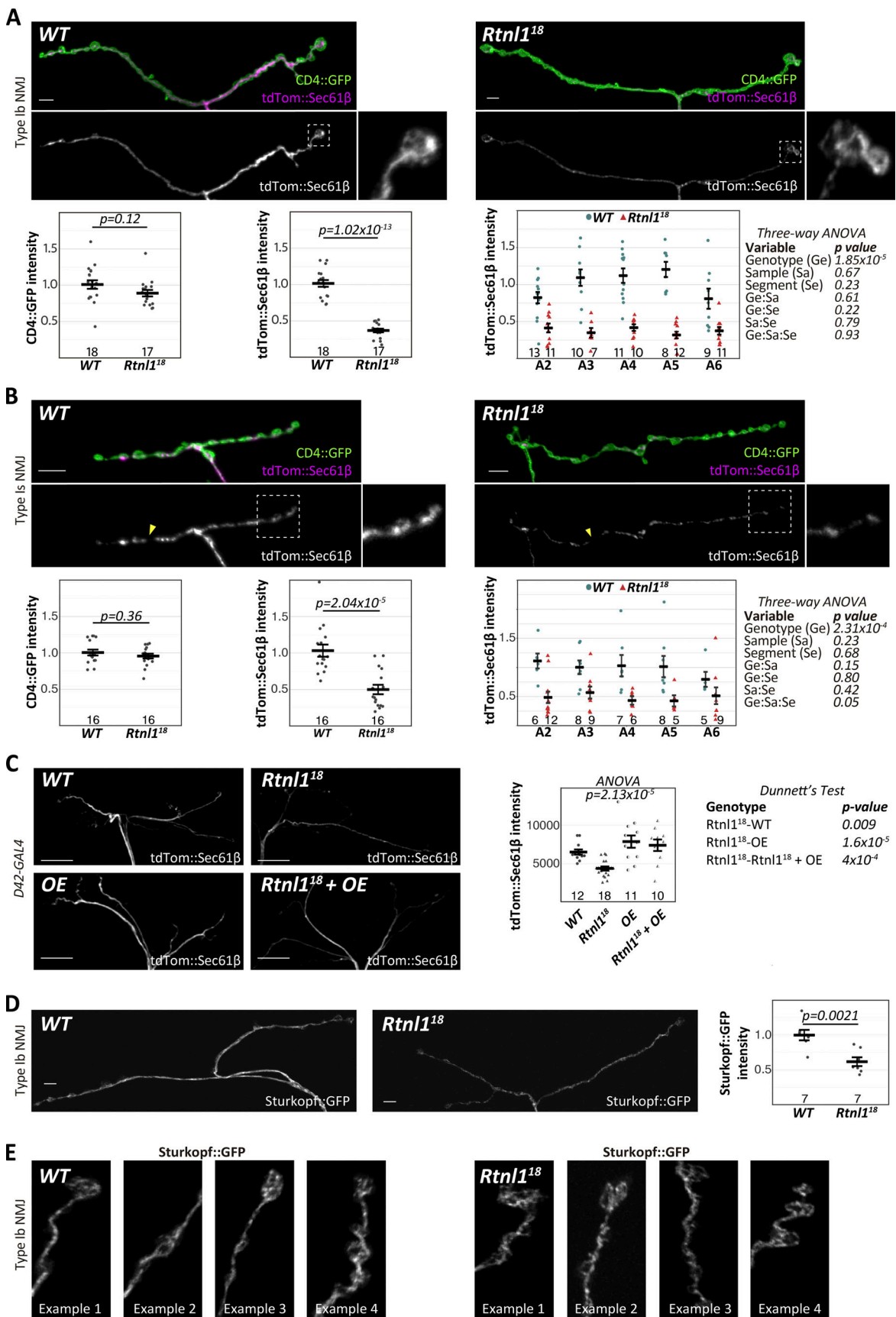

Figure 2. **Rtnl1 loss depletes presynaptic ER membrane markers. (A and B)** Representative confocal projections and quantifications of ER marker tdTom::Sec61β in (A) Type Ib terminals on muscle 1 and (B) Type Is terminals on muscles 1–9 of *WT* and *Rtnl1[18]* larvae. Insets show magnified views of the areas inside

broken lines. Arrowheads in B indicate gaps in tdTom::Sec61β signal. **(C)** Rtnl1::HA overexpression (OE) rescues the depletion of tdTom::Sec61β in *Rtnl1^18^* mutant NMJs at muscle 1/9. Since Type Ib and Is *GAL4* constructs, *UAS-Rtnl1::HA*, and *UAS-tdTom::Sec61β*, are all inserted at *attP2*, we used *D42-GAL4* which could be recombined with the *UAS* insertions. **(D and E)** Sturkopf::GFP in Type Ib muscle 1 NMJs, with intensity analyzed as above, and high magnification examples. All plots show individual larval datapoints and mean ± SEM; y-axes indicate arbitrary units (au) after normalization to *WT*; sample size is within the plots for each genotype. Where tdTom::Sec61β signal is compared between segments (A and B), sample size is the number of hemisegments (NMJs); for other plots, sample size is the number of larvae. For each larva, several NMJs across segments A2-A6 were randomly analyzed, and the mean value used as a larval datapoint. For each larva in C, we analyzed (but did not distinguish) Type Ib and Is branches on muscles 1 and 9 in a single segment chosen randomly between A4-A6. Student's *t* tests were used for pairwise comparisons; ANOVA was used as shown for comparisons of more than two categories, or of multiple factors, with Dunnett's post-hoc testing where appropriate. Scale bars for A, B, and D are 5 µm; for C, 20 µm. Insets in A and B are 8 × 8 µm. Panel width in E is 10 µm. Genotypes: A and B, *GAL4, UAS-CD4::tdGFP/UAS-tdTom::Sec61β*; C, *D42-GAL4, UAS-tdTom::Sec61β* and *D42-GAL4, UAS-tdTom::Sec61β/UAS-Rtnl1::HA* in either *Rtnl1^+^* (*WT*) or *Rtnl1^18^* background (OE and *Rtnl1^18^* + OE respectively); D–E, *Ib-GAL4, UAS-Sturkopf::GFP/+*; all in either a *WT* or *Rtnl1^18^* background.

In *Rtnl1^18^* NMJs, labeling of bright GFP::HDEL puncta was not affected, but labeling between puncta was significantly decreased compared to *WT* (Fig. 3, B–E and Fig. S5 B). We concluded that the tubular ER network, but not the amount or volume of ER cisternae, is reduced in *Rtnl1* mutant NMJs; and that the stronger depletion of ER membrane markers in *Rtnl1* mutant genotypes reflects the higher proportion of these markers in tubules compared to cisternae, due to the high surface area/volume ratio of presynaptic ER tubules.

### *Rtnl*1 loss reduces STIM-mediated ER-PM contact sites
The decreased levels of ER membrane and tubules, but not cisternae, in *Rtnl1* mutant presynaptic termini provided a unique model to explore the specific functional and physiological roles of presynaptic ER tubules. Since contact sites must depend on availability of ER membrane, we first tested two critical roles of ER-PM contact sites, in lipid and Ca²⁺ exchange between both compartments (Öztürk et al., 2020).

The PM phospholipid phosphatidylinositol 4,5-bisphosphate (PI(4,5)P2) is derived from phosphatidylinositol (PI) formed at the ER membrane. PI(4,5)P2 binds to ER membrane proteins to mediate ER-PM tethering (Muallem et al., 2017), and it is also required for presynaptic function (Lauwers et al., 2016). However, using the sensor PLCδ::PH::GFP (Verstreken et al., 2009), we did not detect any difference in PI(4,5)P2 levels between *WT* and *Rtnl1* NMJs (Fig. S6). Although we cannot exclude that other ER-PM contact-site-dependent lipids are affected in *Rtnl1* mutants, a decrease in presynaptic tubular ER does not necessarily impact lipid homeostasis.

ER-PM contact sites can also mediate Ca²⁺ exchange. STIM proteins are ER Ca²⁺ sensors that can also bind to the PM at ER-PM contact sites, promoting SOCE upon emptying of ER Ca²⁺ stores (Serwach and Gruszczynska-Biegala, 2020). STIM proteins are distributed throughout the ER membrane, but enriched at ER-PM contacts, which increase in number when ER is depleted of Ca²⁺ (Liou et al., 2005; Shen et al., 2021). *Drosophila* has one STIM gene, orthologous to mammalian STIM1 and STIM2 (Williams et al., 2001). In *WT* motor neuron terminals, STIM::mCherry (Bi et al., 2014) showed, as expected, a mostly continuous distribution, with some foci characteristic of STIM accumulation at ER-PM contact sites. STIM:mCherry levels were generally reduced in *Rtnl1* NMJs (Fig. 4), like other ER membrane markers (Fig. 2), consistent with less presynaptic ER surface. STIM::mCherry foci were slightly less intense and took up less area in *Rtnl1* mutants than in *WT* larvae, suggesting a reduction

of these ER-PM contact sites (Fig. 4), and that Ca²⁺ stores are not depleted in *Rtnl1* mutants. STIM::mCherry foci area was highly variable among wild-type NMJs, but consistently low in *Rtnl1* NMJs, suggesting that *WT* presynaptic ER could be responding to physiological demands (e.g., variable emptying of ER Ca²⁺) by adjusting the number of STIM-containing ER-PM contact sites.

### *Rtnl*1 loss reduces neurotransmitter release and evoked presynaptic Ca²⁺ influx by impairing STIM-dependent ER-PM contacts
Synaptic strength is regulated by Ca²⁺, both extracellular and in the ER. Alterations to ER as the largest intracellular Ca²⁺ store could therefore potentially alter presynaptic Ca²⁺ physiology, with consequences for processes such as synaptic strength and ATP generation that could contribute to neurodegeneration in conditions including the HSPs. However, since the levels of the lumenal marker GFP::HDEL (Fig. 3) and the lumenal Ca²⁺ sensor GCaMP-210 (Oliva et al., 2020) are relatively unaffected by loss of *Rtnl1*, processes that depend simply on the amount or capacity of presynaptic ER Ca²⁺ storage might not necessarily be affected. We therefore tested how far synaptic function and presynaptic Ca²⁺ fluxes were affected by the loss of *Rtnl1*, which reduced presynaptic ER membrane and ER tubules, but left ER volume and Ca²⁺ storage capacity largely intact.

We first tested a readout of synaptic transmission using a Ca²⁺ indicator inserted in the postsynaptic muscle membrane, Mhc-SynapGCaMP6f (Newman et al., 2017). *Rtnl1* loss caused a large decrease in postsynaptic Ca²⁺ responses to low-frequency stimulation, suggesting decreased presynaptic neurotransmitter release (Fig. S7, A–C; and Videos 1 and 2). This result is consistent with strongly reduced EJP amplitudes in *Rtnl1^1^* NMJs (Summerville et al., 2016). Loss of *Rtnl1* also strongly reduced the frequency of miniature transmission (spontaneous vesicle release, seen as transient highly localized elevations in SynapGCaMP6f fluorescence; Fig. S7, D and E; and Fig. S8; and Videos 3 and 4). Loss of *Rtnl1* had no effect on the amplitude of miniature transmission (Fig. S7 F) indicating that loss of *Rtnl1* does not change neurotransmitter load per vesicle.

We next tested whether reduced synaptic transmission at *Rtnl1* mutant NMJs could be due to low presynaptic Ca²⁺ responses to neuronal activity. We tested Type Ib and Is presynaptic terminals separately, due to their different physiological properties that could potentially be linked to Ca²⁺ storage and fluxes, and their different contributions to postsynaptic responses (Aponte-Santiago et al., 2020; Aponte-Santiago and

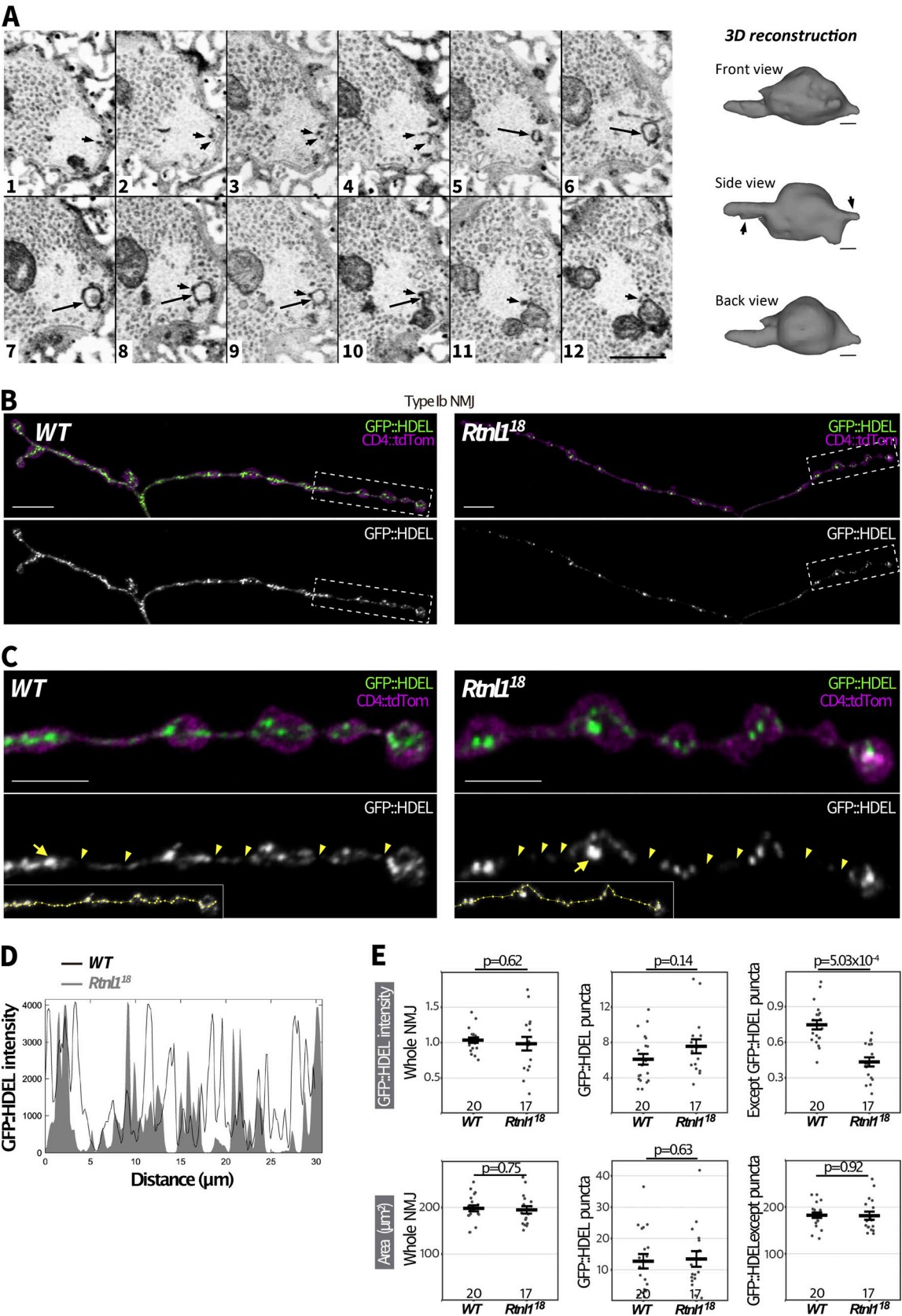

Figure 3. **Rtnl1 loss causes localized depletion of presynaptic ER lumenal marker. (A)** A series of 12 serial ATUMtome 50-nm sections through a ROTO-stained larval bouton, visualized by scanning EM and inverted grayscale images. An example is shown of a cisterna (large arrows in sections 5–10) that appears to be continuous with tubules in the same and neighboring sections (smaller arrows), in a region with few synaptic vesicles. Tubules are identified as darkly

staining puncta in successive sections; their lumen is too small to detect using this method (Terasaki, 2018; Yalçın et al., 2017). Scale bar in panel 12, 500 nm. The cisterna highlighted by the large arrow and its attached tubules (arrows in "Side View") were used to generate a reconstruction, shown on the right and viewed from three different angles. EM section 1 is on the left of the reconstruction, section 12 on the right. Scale bars, 70 nm. Larval genotype is *Rtnl1^18^/WT*. **(B)** Representative confocal projections showing GFP::HDEL in Type Ib muscle 1 NMJs of *WT* and *Rtnl1^18^* larvae. Scale bars, 10 μm. **(C)** Magnified views of the areas inside broken lines in B. Scale bars, 5 μm. Arrowheads indicate regions with low levels of GFP::HDEL, which we interpret as tubules, and arrows indicate examples of puncta which we interpret as cisternae. **(D)** Insets show the lines drawn along GFP::HDEL signal to plot intensity profiles. **(E)** GFP::HDEL intensity across the whole NMJ, within an ROI containing all puncta, and in the NMJ excluding the puncta ROI. Plots show individual larval datapoints and mean ± SEM; intensity levels indicate arbitrary units (au) after normalization to control (*WT*); intensity values are relative to CD4::tdTom (Han et al., 2011) signal. Larval datapoints were calculated and are shown and compared using Student's *t* tests as in Fig. 2. Genotypes are *Ib-GAL4, UAS-CD4::tdTom/UAS-BiP::sfGFP::HDEL*, in either a *Rtnl1^+^ (WT)* or *Rtnl1^18^* background. Transheterozygous *Rtnl1^18^/Rtnl1^1^* mutant larvae also showed a similar phenotype to *Rtnl1^18^* (Fig. S5 B).

Littleton, 2020; Chouhan et al., 2010). A Ca²⁺ sensor targeted to the cytosolic face of the presynaptic plasma membrane, myrG-CaMP6s, showed no significant change in resting fluorescence in both Type Is and Type Ib *Rtnl1^18^* boutons compared to *WT* (Fig. 5 A and Fig. S9 A), indicating that *Rtnl1* loss does not change resting cytosolic [Ca²⁺]. We corroborated this conclusion using a ratiometric cytosolic Ca²⁺ sensor *UAS-tdTom-p2a-GCaMP56* (Daniels et al., 2014), which also detected no difference in resting cytosolic Ca²⁺ in Type Ib *Rtnl1^18^* boutons compared to *WT* (Fig. 5 B). Evoked myrGCaMP6s responses (Videos 5 and 6) were reduced in Type Ib and Is *Rtnl1^18^* boutons at all stimulation frequencies tested (Fig. 5, C and D; and Fig. S9 B; and Fig. S10 A). This reduction appeared to be due to loss of *Rtnl1*; it could be rescued by a *UAS-Rtnl1::HA* transgene (Fig. 5 E); furthermore, transheterozygous *Rtnl1^1^/Rtnl1^18^* mutant larvae displayed the same phenotype as homozygous *Rtnl1^18^* mutants (Fig. S9, B and C). Interestingly, synaptic size was not reduced in *Rtnl1^18^* NMJs; instead, they showed a small but significant increase in the number of synaptic boutons (muscle 1 Type Ib NMJ; *WT*: 24.6 ± 0.5 [mean ± SEM], n = 18 larvae; *Rtnl1^18^*: 27.8 ± 1.0 [mean ± SEM], n = 17 larvae; P = 0.0075**, Student's *t* test), consistent with previous data from *Rtnl1^1^* (Summerville et al., 2016). Loss of *Rtnl1* did not affect the dynamics of the response, as neither the time to peak response, nor the recovery half-time of Type Is or Ib boutons were affected (Fig. S9, D–F and Fig. S10, B–D). Taken together, our data suggested reduced neurotransmission in *Rtnl1^18^* NMJs. This reduced neurotransmission could potentially be accounted for by reduced evoked presynaptic cytosolic Ca²⁺ responses.

Since STIM-mediated ER-PM contact sites are known to mediate Ca²⁺ exchange, and both STIM levels and foci were reduced in *Rtnl1* mutants (Fig. 4), we tested whether STIM levels were associated with the observed defects in presynaptic Ca²⁺ influx. We found that the overexpression of STIM::mCherry in motor neurons rescued cytosolic Ca²⁺ influx in *Rtnl1* mutants (Fig. 5 E), suggesting that the depletion of the presynaptic ER surface may lower presynaptic Ca²⁺ influx by downregulating STIM signaling.

### *Rtnl1* loss decreases evoked ER Ca²⁺ uptake at Type Is and Type Ib NMJs
Further, we examined whether depletion of ER tubules differentially affected ER Ca²⁺ fluxes in NMJs with phasic or sustained firing profiles, Type Is and Type Ib, respectively.

At *WT* Type Is terminals, an ER lumenal GCaMP6-210 reporter showed rapid transient decreases in ER lumenal fluorescence immediately after stimulation, suggesting release of Ca²⁺ to the cytosol evoked by repetitive stimulation, followed by

a slow increase in fluorescence, which was more pronounced at higher stimulation frequencies (Fig. 6; and Video 7), and lagged and outlasted the evoked elevation of cytosolic Ca²⁺ (Figs. 5, S9, and S10). Rtnl1 loss did not significantly change resting fluorescence (Fig. 6, A and B), nor the rapid release (Fig. 6 C) across most stimulation frequencies, except that release was slightly higher at the lowest frequency, 5 Hz (Fig. 6 D). However, *Rtnl1^18^* mutants showed less Ca²⁺ uptake by ER than *WT*, after the initial rapid release (Fig. 6 C) across all stimulation frequencies tested (Fig. 6 E, Fig. S11 A, and Video 8), suggesting that the ER tubule depletion in *Rtnl1* mutants primarily diminishes the uptake of ER Ca²⁺ in Type Is boutons.

*WT* Type Ib terminals only occasionally showed the initial rapid evoked decrease in ER lumenal fluorescence seen in Type Is termini, and showed a strong and sustained increase in lumenal fluorescence that lagged and outlasted the cytosolic Ca²⁺ response, similar to Type Is boutons, but stronger (Fig. 7; and Video 9). Rtnl1 loss again did not alter resting ER lumenal GCaMP fluorescence relative to *WT* (Fig. 7, A and B), but often caused the initial rapid Ca²⁺ release to appear (Fig. 7, C and D) at every stimulation frequency tested (Fig. 7 E, Fig. S11 B, and Videos 10 and 11), and led to lower evoked ER Ca²⁺ uptake (Fig. 7 F), similar to Type Is termini.

### *Rtnl1* loss decreases mitochondrial Ca²⁺ uptake at Type Is and Type Ib NMJs
Mitochondrial Ca²⁺ stimulates ATP production (Datta and Jaiswal, 2021), and since Ca²⁺ released from ER and/or present in the cytosol can potentially be taken up by mitochondria, we hypothesized that mitochondrial Ca²⁺ could be impaired by the changes in ER architecture elicited by *Rtnl1* loss. We tested this hypothesis using a Ca²⁺ sensor adapted for mitochondrial conditions, CEPIA3mt (Suzuki et al., 2014).

In *WT* Type Is and Ib NMJs, mitochondria showed a rapid increase in [Ca²⁺] in response to repetitive stimulation, followed by a rapid partial decline, and a slow return to baseline over a period greater than 40 s (usually 1–2 min; Fig. 8; and Video 12). As with ER lumenal Ca²⁺, this response followed and outlasted the evoked cytosolic Ca²⁺ response. Loss of Rtnl1 did not affect resting CEPIA3mt fluorescence in either Type Is or Ib boutons (Fig. 8, A and B; Fig. S12 A; Fig. S13, A and B; and Video 13). Immunostaining for the myc epitope tag of the CEPIA3mt sensor showed that CEPIA3mt sensor levels were not altered between *WT* and *Rtnl1^18^* larvae (Fig. S13, A and C), implying that the levels of expression of the sensor were comparable and the CEPIA3mt fluorescence is reporting relative [Ca²⁺] in these genotypes.

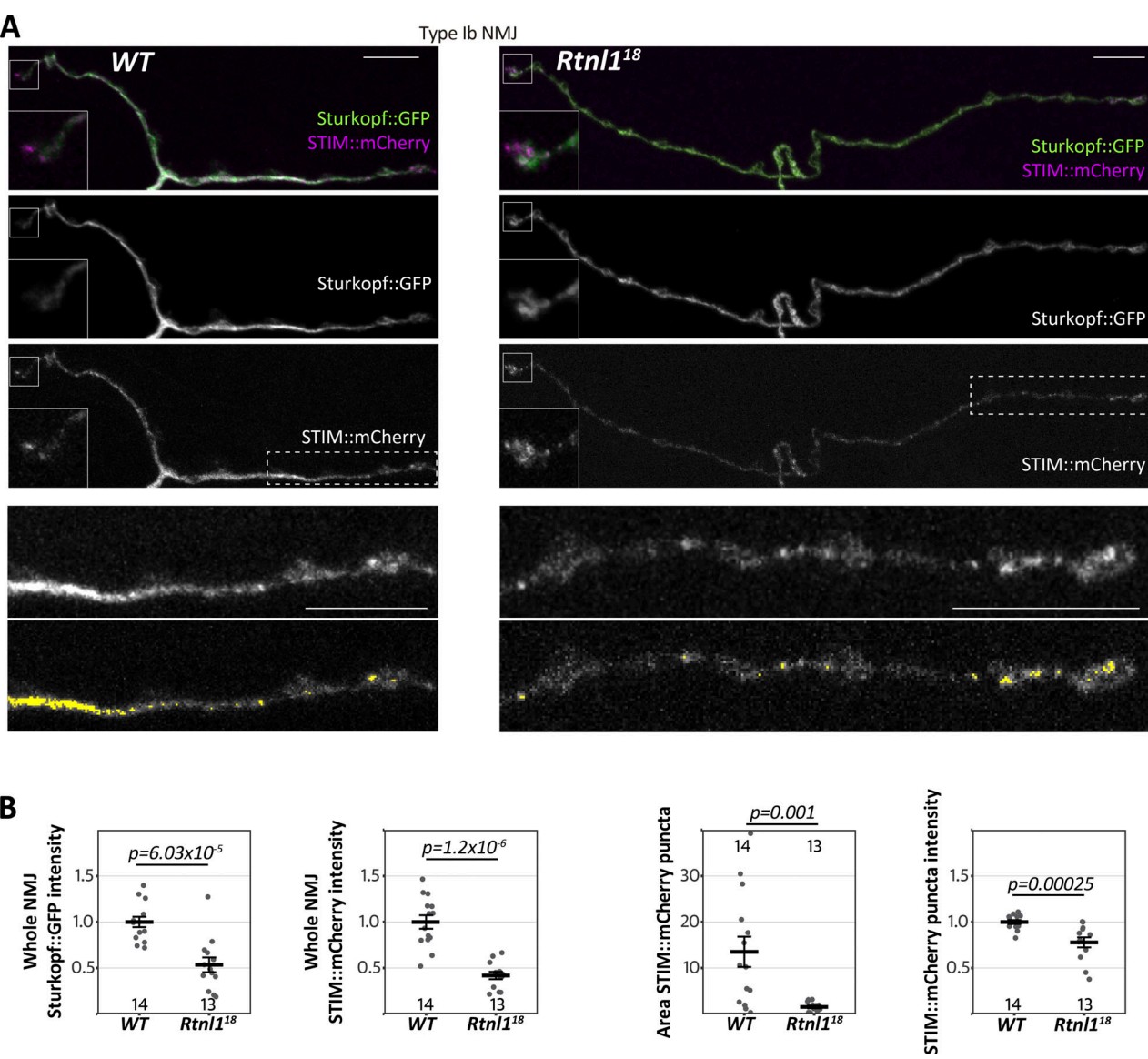

**Figure 4. Rtnl1 loss depletes presynaptic STIM and reduces STIM foci. (A)** Representative confocal projections showing the ER markers STIM::mCherry and Sturkopf::GFP in Type Ib muscle 1 NMJs of *WT* and *Rtnl1¹⁸* larvae. Insets in the top three rows show a magnified view of terminal boutons. Bottom panels show magnified views of the areas indicated with a broken line. Areas of elevated STIM ("foci") are sometimes punctate and sometimes extended and are highlighted in yellow in the bottom panels. Scale bars, 10 μm. Insets, 6 × 6 μm. **(B)** Plots show individual larval datapoints and mean ± SEM; intensity levels indicate arbitrary units (au) after normalization to control (*WT*). Larval datapoints were calculated and are shown as in Fig. 2; Except for Sturkopf::GFP intensity (Student's *t* test), pairwise comparisons were performed with Mann–Whitney U tests. Genotypes are *Ib-GAL4, UAS-Sturkopf::GFP/UAS-STIM::mCherry*, in either a *WT* or *Rtnl1¹⁸* background.

Although *Rtnl1¹* NMJ mitochondria have been reported as larger than *WT* (O'Sullivan et al., 2012), we detected only a small and non-significant increase in mitochondrial size in *Rtnl1¹⁸* muscle 1 NMJs compared to *WT* (Fig. S13 D).

However, *Rtnl1* loss significantly lowered evoked NMJ mitochondrial Ca²⁺ uptake compared to *WT* at every stimulation frequency tested (Fig. 8, C and D; and Fig. S12 B and Fig. S13 E). Transheterozygous mutant controls had statistically similar responses to homozygous *Rtnl1¹⁸* larvae (Fig. S12, B and C), corroborating that the phenotype is due to loss of Rtnl1. Further, expression of *UAS-Rtnl1::HA* under GAL4 control fully rescued evoked mitochondria Ca²⁺ uptake in Ib presynaptic boutons (Fig. 8 E).

Although Type Is and Ib NMJs have different mitochondrial densities, responses to nerve stimulation, and Ca²⁺ sequestration capacities (Chouhan et al., 2010), *Rtnl1* loss reduced evoked mitochondria Ca²⁺ responses in both bouton classes. Heterozygous *Rtnl1¹⁸/WT* NMJ mitochondria Ca²⁺ responses were similar to *WT* larvae at the higher stimulation frequencies, but were reduced at lower frequencies, suggesting a mild dominant effect of *Rtnl1¹⁸* (Fig. S12, B and C).

Genotype did not affect time to peak evoked Ca²⁺ in either Is or Ib boutons (Fig. S12 D and Fig. S13 F), nor time to 50% recovery in Ib boutons (Fig. S13 G). In mitochondria of Type Is boutons, loss of Rtnl1 elicited faster 50% recovery times, in both

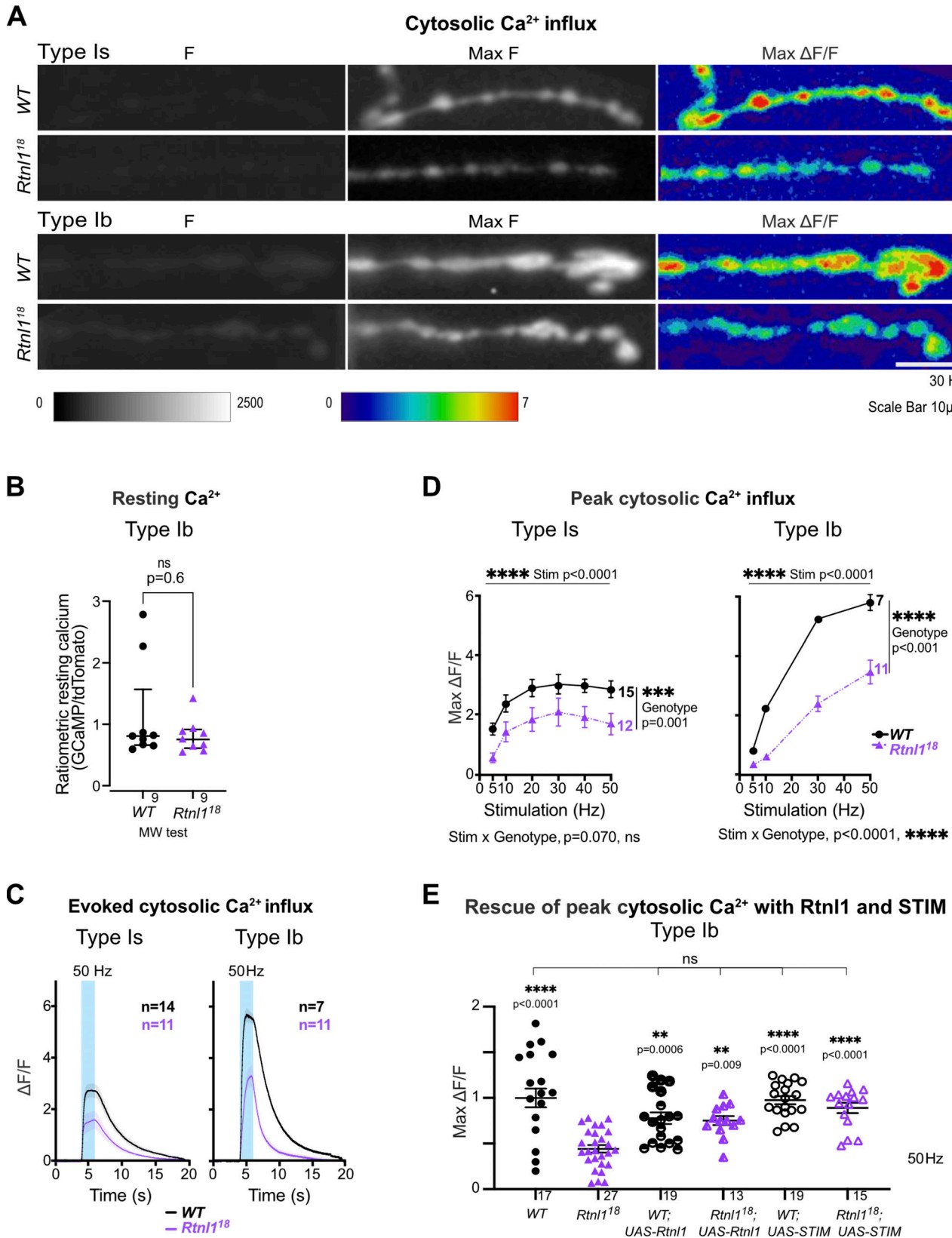

Figure 5. **Loss of Rtnl1 decreases evoked cytosolic Ca²⁺ responses in presynaptic NMJs.** Cytosolic Ca²⁺ responses to 2 s of 30-Hz stimulation were measured at Type Is and Type Ib termini at muscle 1 in segment A4-A6, using myr::GCaMP. **(A)** Panels show GCaMP fluorescence at rest (F), maximum fluorescence (Max F), and maximum relative change in fluorescence (Max ΔF/F) in representative examples of *WT* and *Rtnl1¹⁸* presynaptic terminals. **(B)** The ratiometric cytosolic Ca²⁺ sensor tdTom-p2a-GCaMP56 detected no difference in resting cytosolic Ca²⁺ in Type Ib *Rtnl1¹⁸* boutons compared to *WT*. Graphs show individual larval datapoints for the ratio of GCaMP to Tomato fluorescence, with one Type Ib muscle 1 NMJ per larva, averaged across the responding

area. Genotypes were compared using a Mann–Whitney U test, with median ± interquartile range shown. **(C)** Timecourses of evoked cytosolic GCaMP responses to 50 Hz stimulation (mean ± SEM of larval datapoints). **(D)** Loss of Rtnl1 significantly decreases cytosolic Ca²⁺ responses across a range of stimulation frequencies. Graph shows mean ± SEM of maximum evoked ΔF/F values (larval datapoints), with comparisons using mixed-effects model repeated-measures ANOVA. **(A–D)** Genotypes: *Is-GAL4* or *Ib-GAL4, UAS-myr::GCaMP6s/UAS-tdTom::Sec61β* in either a *WT* or *Rtnl1¹⁸* mutant background. **(E)** Expression of *UAS-Rtnl1::HA* or *UAS-STIM::mCherry* using *Ib-GAL4* rescues evoked cytosolic Ca²⁺ response in mutant larvae. Data were normalized to the *WT* mean of maximum cytosolic GCaMP responses (ΔF/F) evoked by 50 Hz stimulation, from two different microscope setups. Graph shows individual larval datapoints, with one NMJ per larva, averaged across the responding area. Genotypes: *Ib-GAL4, UAS-myr::GCaMP6s/attP2⁺*, or *Ib-GAL4, UAS-myr::GCaMP6s/UAS-Rtnl1::HA*, or *Ib-GAL4, UAS-myr::GCaMP6s/UAS-STIM::mCherry*, in either a *WT* or *Rtnl1¹⁸* mutant background; the *attP2* landing site acts as a *WT* control for insertion of *UAS-Rtnl1::HA* at *attP2*.

*Rtnl1¹⁸* mutants and transheterozygous controls, compared to *WT* controls (Fig. S12 E).

### Comparison of Ca²⁺ responses and their Rtnl1 dependence across compartments

Comparison of the above Ca²⁺ responses in different organelles (Fig. 9) revealed a rapid rise and fall in cytosolic Ca²⁺, followed by strong and longer lasting Ca²⁺ responses in both mitochondria and ER. These time courses reveal that loss of *Rtnl1* affects the evoked Ca²⁺ responses of the three presynaptic compartments with similar severity, with Ca²⁺ responses being reduced to just over a half of those of *WT* in Ib boutons and Is cytosolic responses, and just under half in Is mitochondria and ER responses.

## Discussion

From a body of work mostly in cultured non-neuronal cells, we know that the ER controls Ca²⁺ handling (Öztürk et al., 2020). This role appears conserved in presynaptic ER, where both

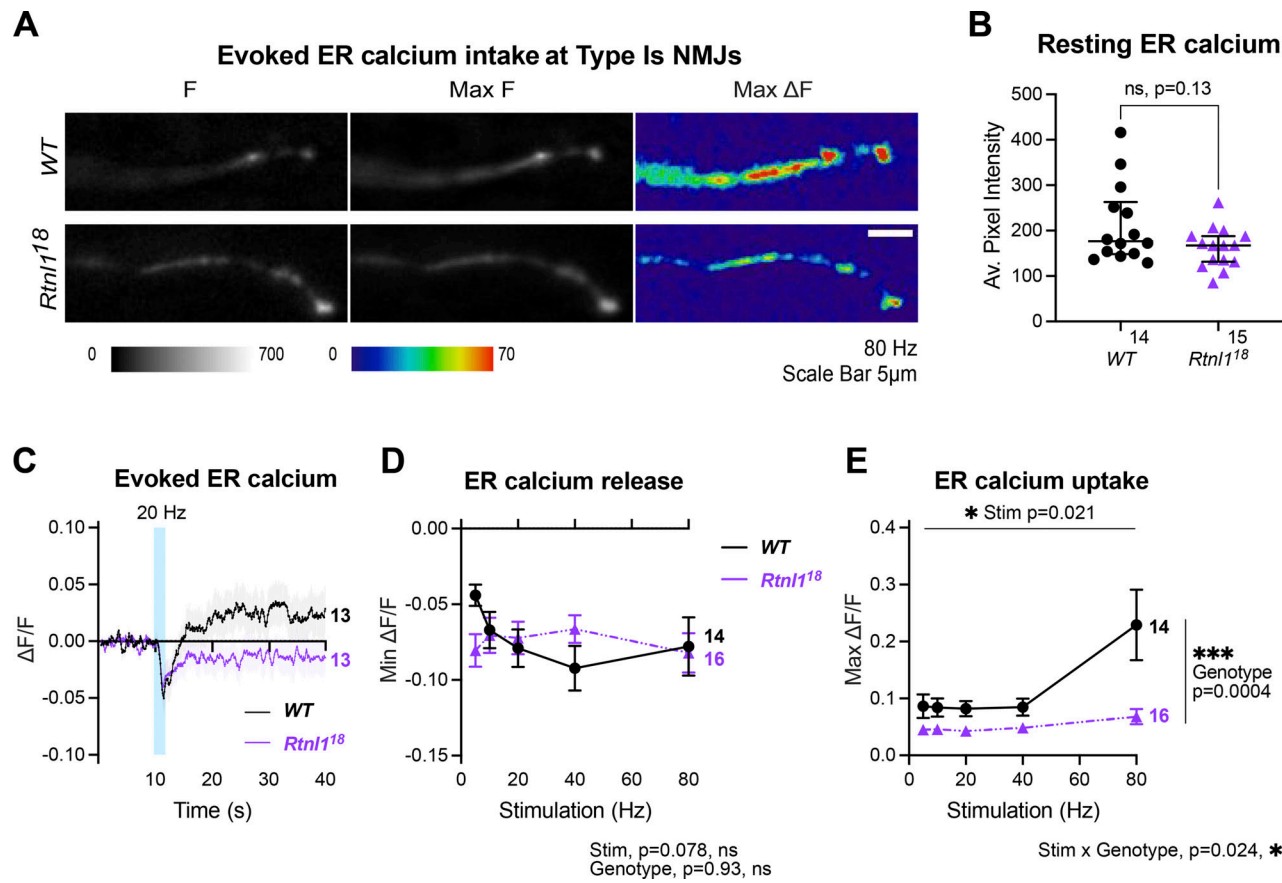

Figure 6.  **Rtnl1 loss decreases evoked ER Ca²⁺ uptake at Type Is termini.** Evoked ER Ca²⁺ responses were measured at Type Is termini at muscle 1 in segment A4-A6. **(A)** Lumenal GCaMP fluorescence at Type Is NMJs, presented at rest (F), maximum fluorescence (Max F), and maximum change in fluorescence (Max ΔF) in representative examples of *WT* and *Rtnl1¹⁸* presynaptic terminals. **(B)** Loss of Rtnl1 does not affect resting ER Ca²⁺. The graph shows larval datapoints as in Fig. 5 B, with median ± interquartile ranges, compared using a Mann–Whitney U-test. **(C)** ΔF/F timecourse of evoked lumenal GCaMP responses to 20 Hz stimulation (mean ± SEM). **(D)** Loss of Rtnl1 does not affect ER Ca²⁺ release immediately following stimulation. The graph shows the minimum ΔF/F value after stimulation. **(E)** Loss of Rtnl1 significantly decreases evoked ER Ca²⁺ uptake over a range of stimulation frequencies. The graph shows the maximum ΔF/F value reached after stimulation. In D and E, graphs and analyses are as for Fig. 5 D. Genotypes: *Is-GAL4, UAS-ER-GCaMP6-210/UAS-tdTom::Sec61β* in either a *WT* or *Rtnl1¹⁸* mutant background.

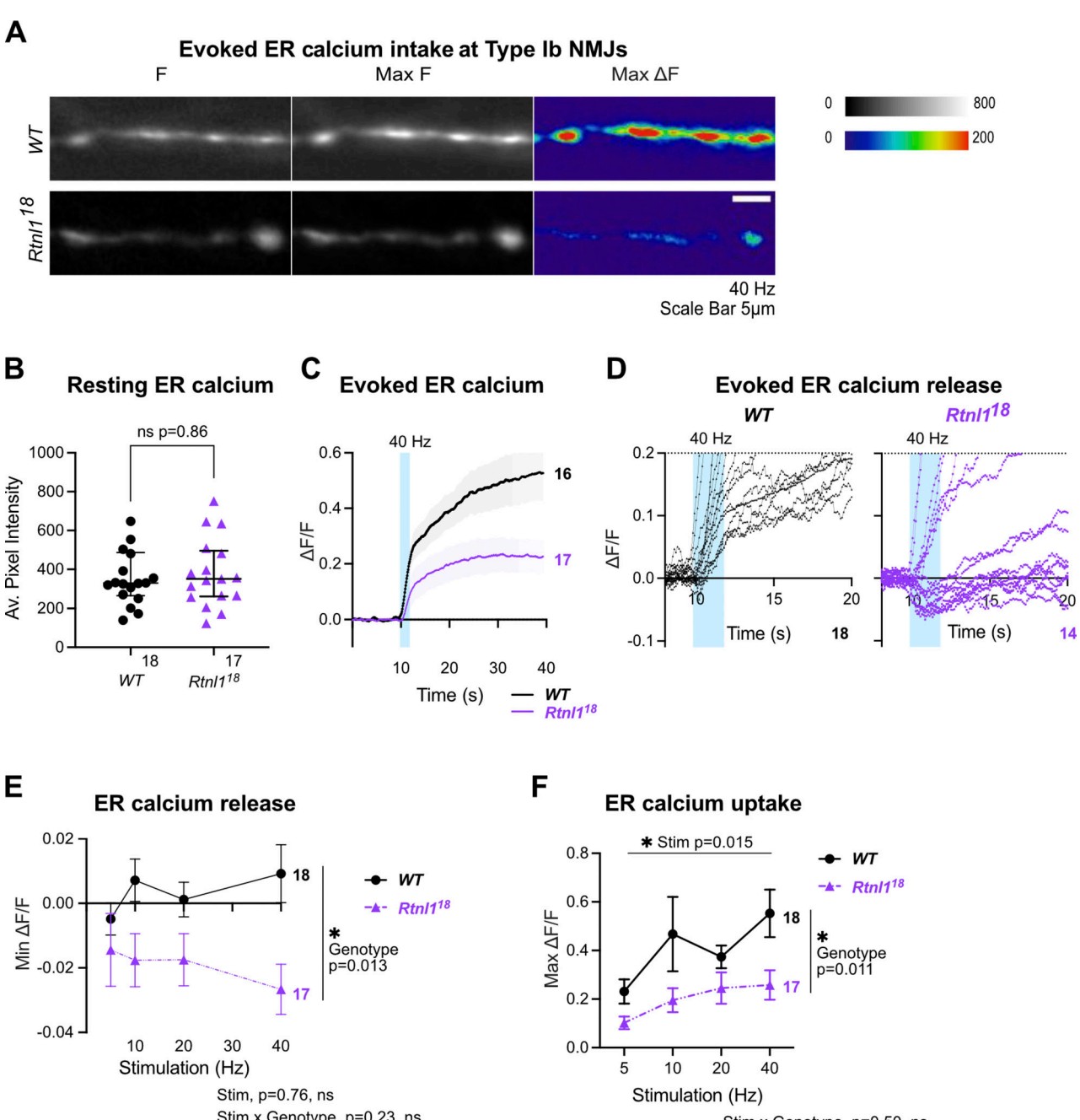

Figure 7. **Rtnl1 loss decreases evoked ER Ca²⁺ uptake in Type Ib termini.** Evoked ER Ca²⁺ responses were measured at Type Ib termini at muscle 1 in segment A4-A6. **(A)** Lumenal GCaMP fluorescence at Type Ib NMJs presented as in Fig. 6 A. **(B)** Loss of Rtnl1 does not affect resting ER resting lumen GCaMP. Graphing and analysis are as for Fig. 6 B. Some outlier datapoints are excluded from the graph but included in statistics. **(C)** ΔF/F time course of evoked lumenal GCaMP responses to 40 Hz stimulation (mean ± SEM). **(D)** ΔF/F time courses from individual larvae show that transient evoked Ca²⁺ release from ER, common in Type Is termini (Fig. 6 C) is mostly undetectable in WT Type Ib termini but found in over half of Rtnl1¹⁸ Type Ib termini tested. **(E)** Loss of Rtnl1 increases ER Ca²⁺ release across a range of stimulation frequencies. **(F)** Loss of Rtnl1 significantly decreases evoked ER Ca²⁺ uptake over a range of stimulation frequencies. Graphs in E and F are presented and analyzed as in Fig. 5 D. Genotypes: *Ib-GAL4, UAS-ER-GCaMP6-210/UAS-tdTom::Sec61β* in either a WT or Rtnl1¹⁸ mutant background.

ER-PM contacts (de Juan-Sanz et al., 2017) and the amount of ER (Kuijpers et al., 2021) are critical for neurotransmission. Given the relatively larger Ca²⁺ storage capacity of ER cisternae, are presynaptic ER tubules contributing to Ca²⁺ dynamics? In this work, we have disrupted the tubular ER-shaping protein *Rtnl*1 and found a decrease of presynaptic ER tubules (schematized in Fig. 10). Rtnl1

loss results in a unique model to study the role of ER tubules at the presynaptic region. Here, we used this model to analyze Ca²⁺ handling, which relates with neurotransmission and the biological basis of neurological disorders associated with tubular ER.

Loss of either *Rtnl*1 or REEPs causes a loss of axonal ER preferentially in longer motor neurons of *Drosophila* larvae

Figure 8. ***Rtnl1* loss decreases evoked mitochondrial Ca²⁺ uptake in Ib and Is presynaptic termini.** Evoked mitochondrial Ca²⁺ responses were measured at Type Ib and Type Is termini at muscle 1 in segment A4-A6. **(A)** Mitochondrial CEPIA3mt fluorescence presented as in Fig. 6 A. **(B)** Loss of Rtnl1 does not alter resting mitochondrial Ca²⁺ levels in Type Is or Type Ib terminals. Some outlier datapoints are excluded from the graph but included in statistics. Graphs and analyses are as in Fig. 6 B, except that Type Ib are shown as mean ± SEM and compared using Student's *t* test. **(C)** Time courses of evoked mitochondrial

CEPIA3mt responses to 80 Hz stimulation (mean ± SEM) show reduced Ca²⁺ uptake by mitochondria on loss of Rtnl1. **(D)** Loss of Rtnl1 significantly decreases mitochondrial Ca²⁺ uptake across a range of stimulation frequencies. Graphs and analyses are as for Fig. 5 D. **(A–D)** Genotypes: *Is-GAL4* or *Ib-GAL4, UAS-CEPIA3mt/UAS-tdTom::Sec61β* in either a *WT* or *Rtnl1¹⁸* mutant background. **(E)** Expression of *UAS-Rtnl1::HA* under *Ib-GAL4* control rescues evoked Ca²⁺ uptake by mitochondria in mutant larvae. Data shows maximum evoked mitochondrial CEPIA3mt responses (ΔF/F) after 80 Hz stimulation. Graph shows individual larval datapoints, with one NMJ datapoint per larvae, averaged across the responding area; statistical comparisons were performed using one-way ANOVA followed by Dunnett's post-hoc multiple comparisons. Genotypes: *Ib-GAL4, UAS-CEPIA3mt/UAS-Rtnl1::HA* or *Ib-GAL4, UAS-CEPIA3mt/attP2⁺* in either a *WT* or *Rtnl1¹⁸* mutant background; the *attP2* landing site acts as a *WT* control for insertion of *UAS-Rtnl1::HA* at *attP2*.

(Yalçın et al., 2017). In contrast, the decrease of presynaptic ER network in *Rtnl1* mutants is independent of axon length (Fig. 2, A and B). The tubular ER network in presynaptic terminals (mouse nucleus accumbens) is more extensive than in axons (Wu et al., 2017). It is therefore possible that presynaptic ER is more dependent on the function of tubular ER-shaping proteins than axonal ER. Regarding the continuity of the network, it is possible that neurons can better tolerate loss of ER tubules in presynaptic terminals than in axons, where there are relatively few tubules

to begin with. This might explain why loss of *Rtnl1* causes ER network discontinuity in axons (Yalçın et al., 2017), but not at the presynaptic region.

Increasing the levels of axonal and presynaptic ER (by blocking neuronal autophagy) correlates with elevated Ca²⁺ release from ER, which in turn increases neurotransmission (Kuijpers et al., 2021). In agreement with this, we showed that lower ER levels correlate with decreased neurotransmission (Fig. S7), indicating that the relationship works conversely.

## Organelle time trace comparisons

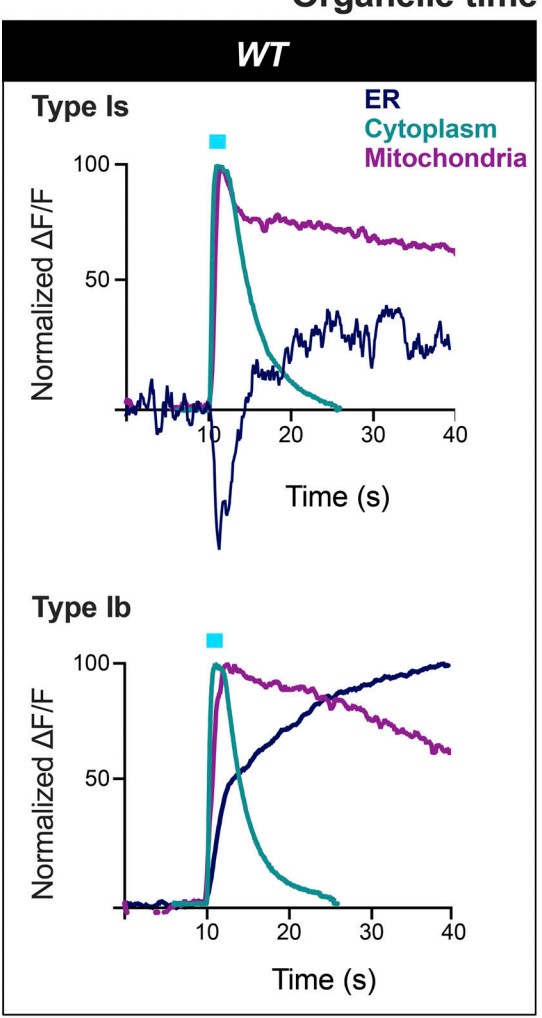
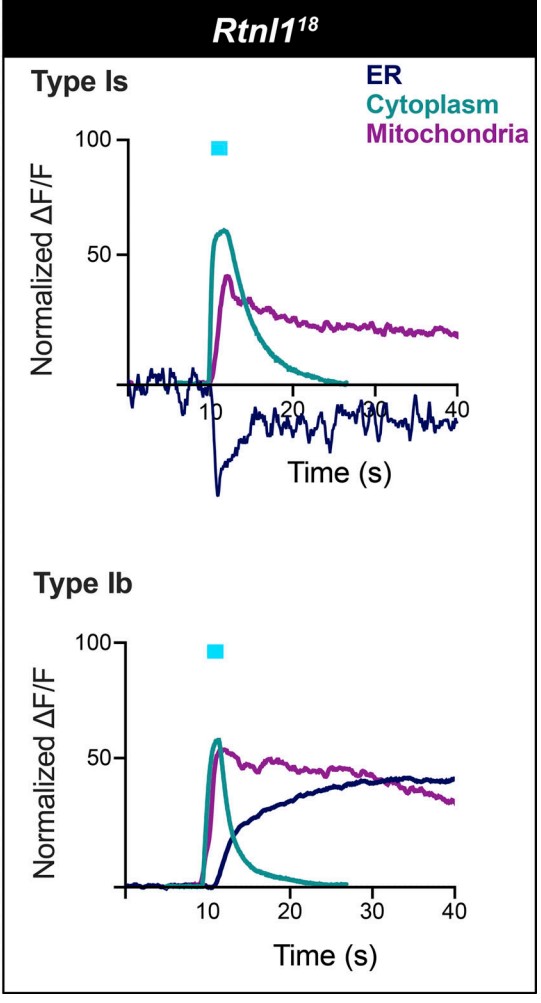

Figure 9. **A comparison of superimposed time courses.** Mean time traces (data from Figs. 5 and 8) from ER, mitochondria, and cytosolic sensors in *WT* and a *Rtnl1* mutant (*Rtnl1⁻*), and in Type Is and Ib excitatory synapses. Plots are normalized so that the difference of minimum to maximum ΔF/F values are set to 100 for the *WT* recordings in each compartment. Most time courses are from 40 Hz stimulation trials. Two exceptions are: the Ib cytosolic trace was with 50 Hz stimulation, as 40 Hz was not tested, Is ER responses were with 20 Hz, as this frequency was most representative of the lack of statistical difference between *WT* and *Rtnl1* mutant responses (Fig. 6 D).

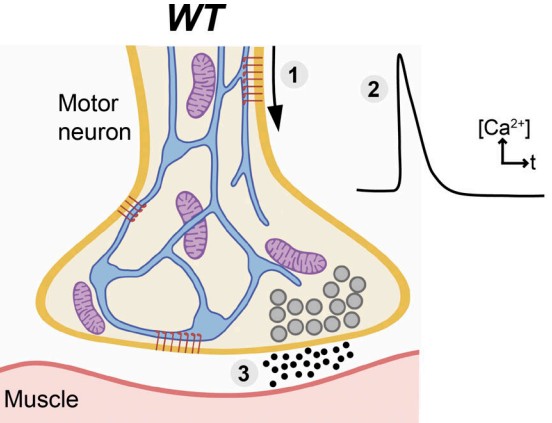

**1** Action potential transmission **2** Ca²⁺ uptake by intracellular compartments **3** Neurotransmission

Figure 10. **Model of effect of Rtnl1 loss on presynaptic ER network and synaptic function.** Schematic diagrams of a *WT* and a *Rtnl1* mutant (*Rtnl1⁻*) presynaptic terminal. The latter shows fewer ER tubules, resulting in less ER network surface, and hence in less ER contact surface with other cell compartments. Reduced presynaptic ER surface is accompanied by generally lower Ca²⁺ fluxes in presynaptic compartments (cytosol, ER lumen, and mitochondria), and by lower neurotransmission. These reductions could be a potential route by which the loss of ER-shaping HSP protein function might lead to motor neuron axonopathy. Key: ER network, blue; mitochondria, purple; synaptic vesicles, gray; MN PM, dark orange; muscle PM, dark red; ER-PM contacts, accumulation of STIM1 (red rods); released neurotransmitter, black dots.

Interestingly, we also found a severe decrease in the frequency of miniature neurotransmission in *Rtnl1* mutants (Fig. S7, D–F and Fig. S8). Since presynaptic ER tubules, but not cisternae, appear to be reduced in *Rtnl1* mutants (Fig. 3), we proposed that tubules help regulate Ca²⁺ handling across presynaptic compartments. Tubule loss abates Ca²⁺ handling in the ER (Figs. 6 and 7), cytosol (Fig. 5), and mitochondria (Fig. 8), in both types of excitatory NMJs in *Drosophila* larvae. As neurotransmitter release is triggered by cytosolic Ca²⁺ influx through PM voltage-gated Ca²⁺ channels, the decrease of postsynaptic Ca²⁺ responses (Fig. S7) and presynaptic cytosolic influx (Fig. 5) corroborate with each other.

ER-PM contacts have an inhibitory role in cytosolic Ca²⁺ influx in primary hippocampal neurons via activated STIM1 on emptying of ER stores (de Juan-Sanz et al., 2017); other studies in neurons have demonstrated that emptying of Ca²⁺ from ER stores (thereby activating STIM) promotes spontaneous (miniature), but not evoked vesicle release at excitatory terminals by the action of STIM2 (Chanaday et al., 2021). Here, we do not expect STIM to be playing an inhibitory role to presynaptic Ca²⁺, as we observed a decrease in STIM levels and foci in *Rtnl1* mutants (Fig. 4), as well as a decrease in evoked Ca²⁺ fluxes, which were rescued by overexpressing STIM (Fig. 5). Work in non-neuronal cells indicate that STIM1/STIM2 promotes Ca²⁺ entry via coupling with Ca²⁺ channels on the PM (Berna-Erro et al., 2009; Chanaday et al., 2021; Gudlur et al., 2018; Jennette et al., 2022), and our in vivo data support a similar relationship in motor neurons for the *Drosophila* STIM1/STIM2 ortholog STIM. Accordingly, we proposed a model for presynaptic compartments, whereby narrow tubular ER modulates Ca²⁺ entry via STIM (Fig. 10). Other components of the network with higher volume, such as cisternae, would work as a Ca²⁺ store, since both ER volume (Fig. 3) and resting ER Ca²⁺ levels (Figs. 6 and 7) are unaffected in *Rtnl1* mutants. Evoked Ca²⁺ uptake by the ER was

also decreased by loss of *Rtnl1* (Figs. 6 and 7). While most lumenal GFP::HDEL or GCaMP, and hence Ca²⁺, appears to be in cisternae that would act as Ca²⁺ reservoirs, evoked ER Ca²⁺ uptake is reduced upon depletion of the tubular ER network in both phasic and tonic firing boutons. Moreover, we saw the tubular network influencing the physiological profile of ER Ca²⁺, with ER Ca²⁺ release occurring on stimulation of *WT* Is boutons, and rarely on stimulation of *WT* Ib boutons. However, with loss of *Rtnl1* and depletion of tubular ER, ER in Ib boutons released Ca²⁺ during neuronal activity in over half of NMJs tested, and thus caused it to act like smaller Is boutons. Our results demonstrate a complex relationship between ER structure, and uptake and release of Ca²⁺ by the ER, suggesting that intact ER cisternae are not sufficient to maintain full levels of synaptic ER Ca²⁺ release and uptake, but that ER tubules are necessary as well.

*Rtnl1* loss also decreased evoked presynaptic Ca²⁺ uptake by mitochondria (Fig. 8), similar to the ER and cytosolic compartments. As mitochondrial Ca²⁺ and ATP production are positively correlated (Datta and Jaiswal, 2021), energy generation in both Type Is and Ib boutons might potentially be impaired on depletion of tubular ER. Mitochondria normally receive Ca²⁺ directly from ER (Rizzuto et al., 1993; de Brito and Scorrano, 2010), although synaptic mitochondria can take up Ca²⁺ from the cytosol (Ashrafi et al., 2020; Chouhan et al., 2010, 2012). However, the comparative kinetics of the Ca²⁺ responses of *WT* cytosol, ER, and mitochondria (Fig. 9) do not allow us to distinguish between mitochondrial Ca²⁺ uptake from the cytosol or from ER in this situation.

Beyond the Ca²⁺ handling defects characterized in this work, we proposed *Rtnl1* mutants as a useful model to explore other biological roles of presynaptic ER, and to understand which presynaptic ER-related processes might be relevant to the mechanisms of HSP. In addition to Ca²⁺ handling, presynaptic ER depletion might affect other cellular processes such as

autophagy, vesicle trafficking, glucose transport, or lipid metabolism (Öztürk et al., 2020). Although we do not observe any PI(4,5)P$_2$ alterations in *Rtnl1* mutants (Fig. S6), it is still possible that other ER-dependent lipids might be affected, since at least some ER-PM contacts seem to be reduced (according to the decreased levels observed for STIM1 foci). Failed lipid transfer at ER-PM contacts has been previously related with reduced expansion of the growth cone and with neuronal PM growth in general (Petkovic et al., 2014), but we did not observe any defect in morphology of mature motor neurons in *Rtnl1* mutants apart from a slight increase in the number of boutons.

Our results reveal at least three aspects of the *Rtnl1* mutant phenotype that could be relevant to the disease mechanisms of HSPs caused by mutations in ER-shaping proteins. First, *Rtnl1* mutants show a reduced frequency of miniature neurotransmission. This induces bouton fragmentation in young adult *Drosophila* so as to resemble that of aged *Drosophila*, whereas elevated minis can delay age-associated fragmentation and prolong motor ability in adult *Drosophila* (Banerjee et al., 2021). Second, *Rtnl1* mutants show decreased evoked cytosolic Ca$^{2+}$ influx and neurotransmission at NMJs, which controls muscle contraction. Third, decreased evoked mitochondrial Ca$^{2+}$ fluxes predict lower ATP generation capacity and potential energy deficits, although these may be mitigated by the reduced levels of synaptic transmission. Our results suggest mechanisms whereby altered organization of ER tubules could be a potential mechanism for diseases including HSP by affecting synaptic Ca$^{2+}$ handling, and thus synapse function.

## Materials and methods

### *Drosophila* genetics
*Drosophila* stocks (Table S1) were maintained on standard culture media at 18°C (Fly Facility from Department of Genetics, University of Cambridge, Cambridge, UK). Crosses used to generate data were performed at 25°C and are listed in Table S2.

### Generation and analysis of Rtnl1 *and* Rtnl1::YFP *mutant alleles*
Disruption of *Rtnl1* was performed by expressing Cas9 together with two guide RNAs (*gRNAs*) that target the genomic sequences encoding Rtnl1 intramembrane domains (Fig. 1 and Fig. S1 A). We used the online tool FIND CRISPRs (2017 version; https://www.flyrnai.org/crispr/) from DRSC/TRiP Functional Genomics Resources (Harvard Medical School) for *gRNA* designing, targeting *Rtnl1*, and filtering for at least four mismatches to any off-target sequences. BLAST sequence searches at www.ncbi.nlm.nih.gov were used to validate the absence of potential off-target sequences for the selected *gRNAs*. *gRNA* sequences were cloned into the tRNA::gRNA plasmid *pCFD5* by following the protocol on http://www.crisprflydesign.org/. In brief, *pCFD5* was digested with Bbs1-HF; DNA fragments containing *gRNAs* were generated using Q5 High-Fidelity DNA Polymerase with the guided sequences 5′-CAGTGGAATCCCTTATCTAC-3′ (1) and 5′-CAAGTTCGGCGTCATTCTGT-3′ (2); Gibson Assembly Master Mix was used to ligate the generated DNA fragments with digested *pCDF5*; High efficiency Transformation Protocol was used to establish bacterial colonies containing the assembled plasmids; colonies

containing inserts with the correct size were sent for sequencing (Source BioSciences) to validate insert orientation. The resulting *pCFD5-2xgRNA* expression plasmid was purified using a QIAprep Spin Miniprep kit and injected into *Drosophila* embryos (microinjection service at Department of Genetics, University of Cambridge). *Rtnl1[18]* and *Rtnl1[4]* were recovered from microinjection into embryos containing *nos-Cas9* on the X chromosome; we used the second chromosome of this stock as *Rtnl1[+] WT*, as the parental chromosome of the *Rtnl1[18]* and *Rtnl1[4]* alleles. Surviving F0 male adults were individually crossed with *If/CyO* virgin females, and the male F1 adults from this cross were crossed with *Sco/CyO* virgin females for a few days, to establish stocks containing any new *Rtnl1* mutations. These males were then sacrificed for genomic DNA extraction, and mutations in *Rtnl1* locus were analyzed by PCR using Surveyor Mutation Detection Kit with Phusion High-Fidelity Polymerase and 5′-GGCAAGGTAAACAGCGAGAC-3′ (*Rtnl1-1A*) and 5′-TGTGTATGGTGGACAAAAGCA-3′ (*Rtnl1-1B*) primers (*WT* amplicon, 629 bp; Thermo Fisher Scientific; Fig. S1 A). The same DNA polymerase and primers were used to generate PCR products from the mutated lines for sequencing (Source BioScience). To generate *Rtnl1[+]* line, a single male from the *nos-Cas9* stock was crossed with *If/CyO* virgin females, and the *Rtnl1[+]/CyO, Tb* males from the progeny were crossed again with *If/CyO* virgin females to establish the stock.

To generate *Rtnl1::YFP[3]*, we first established a stable line expressing the *Rtnl1 gRNAs*. The *pCDF5-2xgRNA* plasmid was injected into *Drosophila* embryos of genotype *nos-phiC31 v[−]* on the X chromosome, and the *attP2* landing site on the third chromosome. Successful phiC31 integrase-mediated integration of *pCDF5*, which contains an *attB* site, was verified by *v[+]* in the resulting F1 adult flies. The *Rtnl1* double *gRNA* line was then used with *nos-Cas9* to generate *Rtnl1* mutations in the germline of *Rtnl1[CPTI001291]* flies.

For RT-PCR, we used a One-Step RT-PCR kit, on total RNA template from a TRIzol reagent extraction (15 males per genotype), and 5′-GGCAAGGTAAACAGCGAGAC-3′ (forward) and 5′-GTTCAAACCCACTGTCCAGG-3′ (reverse) primers (Fig. S2 A), which do not amplify *Rtnl1[1]* (negative control), and amplify only *WT* mRNA (636 bp) but not genomic DNA (13,955 bp).

Unless otherwise specified, PCRs were performed with DreamTaq Green PCR Master Mix. Gel electrophoresis was performed in 2% agarose in TBE. DNA fragment sizes were estimated using a GeneRuler 1 kb Plus DNA Ladder. Primers were designed with Primer3web 4.1.0 (https://primer3.ut.ee). Cloning design and sequence analysis were performed with SnapGene 4.2.11. Bacterial cultures were grown using BD Difco LB Broth, Miller, and BD Bacto Agar.

### Genotyping Rtnl1 *alleles*
We used PCR primer pairs to verify the orientation of the region inverted in *Rtnl1[18]* (Fig. 1 A and Fig. S1 A). Primers 5′-GGCAAGGTAAACAGCGAGAC-3′ (*Rtnl1-1A*) and 5′-CTGTCGCACGAAAAGGTACA-3′ (*Rtnl1-2B*) gave a product of 272 bp in *WT*, but none in *Rtnl1[18]*. Primers 5′-GGCAAGGTAAACAGCGAGAC-3′ (*Rtnl1-1A*) and 5′-GAGGGTTAGGAGCGACAAGT-3′ (*Rtnl1-2A*) primers (both forward in *WT*) gave a PCR product of 242 bp in *Rtnl1[18]*, but none in *WT*. 5′-GGCAAGGTAAACAGCGAGAC-3′ (*Rtnl1-1A*) and 5′-CTG

TCGCACGAAAAGGTACA-3′ (*Rtnl1-2B*) were also used to detect *UAS-Rtnl1::HA*, which gave a 207 bp band (lacking a 65 bp intron) compared to the 272 bp band of *WT* genomic DNA. To genotype *Rtnl1^1^*, 5′-GGAAATTGCGTGGAACTCAT-3′ and 5′-TATTCGCAT TTCCTCGATCC-3′ primers gave a PCR product of 5,623 bp in *WT*, and 496 bp in *Rtnl1^1^* (Yalçın et al., 2017). New stocks were genotyped for *Rtnl1* alleles and *UAS-Rtnl1::HA* after they were constructed to verify their genotypes and at intervals throughout the work.

### Generation of mitochondrial Ca²⁺ sensor
*17xUASTattB-CEPIA3mt::myc* was constructed as for the lumenal ER sensors (Oliva et al., 2020), and integrated by injection at phiC31 landing site *attP86Fb* (Bischof et al., 2007).

### Histology and confocal microscopy
Third instar larvae were dissected in chilled Ca²⁺-free HL3 solution (prefixation HL3; Stewart et al., 1994), and fixed for 10 min in PBS with 4% formaldehyde. Unless otherwise specified, visualization of fluorescent tags was performed via direct imaging without immunostaining. For immunostaining, the dissected preparations were permeabilized in PBS containing 0.1% Triton X-100 (PBT) at room temperature for 1 h. After permeabilization, samples were blocked in PBT with 4% bovine serum albumin for 30 min at room temperature, incubated with primary antibodies (Table S3) overnight at 4°C, and finally incubated with secondary antibodies (Table S3) for 2 h at room temperature. For myc and HA immunostaining, samples were permeabilized for 30 min with PBT, blocked for 1 h with 5% Normal Goat Serum in PBT, and incubated with anti-myc or anti-HA overnight at 4°C. Processed preparations were mounted in Vectashield, and images were collected using EZ-C1 acquisition software (Nikon) on a Nikon Eclipse C1si confocal microscope (Nikon Instruments) with 488 nm (for GFP and Alexa-488 signals), 561 nm (for tdTom, mCherry, and Alexa-594 signals) and 638 nm (for Alexa-647 signal) lasers. Images were captured using a 40×/1.3NA oil Nikon Plan Fluor DIC H/N2 infinity-0.17 WD 0.2 objective, or a 60×/1.4NA oil Plan Apo VC infinity-0.17 DIC N2.

### Live imaging
Wandering third instar larvae were fillet dissected in ice-cold Schneider's *Drosophila* medium and imaged in ice-cold HL3 of the appropriate kind. Presynaptic responses were imaged in low-Mg²⁺ HL3 containing 1 mM Ca²⁺ (Oliva et al., 2020). Postsynaptic responses were imaged in low-Mg²⁺ HL3, with 1 mM *L*-glutamic acid to partially saturate postsynaptic receptors; higher concentrations of *L*-glutamic acid blocked postsynaptic responses, and lower concentrations did not inhibit muscle contractions enough. Muscle contractions often obscured postsynaptic responses even with 1 mM *L*-glutamic acid. New HL3 was kept no longer than 3 d at 4°C, as Ca²⁺ responses became unreliable with HL3 older than this.

Nerves were cut at the base of the VNC with dissection scissors, suctioned into a heat-polished glass pipette (Macleod et al., 2002), attached to a stimulator and train generator, and imaged on an upright widefield Olympus BX50 microscope as described previously (Oliva et al., 2020) using LED illumination (Cairn Instruments): a 470 nm LED with an ET470/40× T495LPXR excitation cube, a cool white LED with an ET572/X35 holder and ET500/20xT485/68dcrb excitation cube, a 59,022 bs infinity cube, and a Cairn Optosplit II beamsplitter with an ET520/40m T565LPXR-UF2 ET632/60m emission cube. ER lumen and mitochondrial responses were imaged at 10 frames per second and EM gain level 100. Presynaptic cytosolic and postsynaptic responses were imaged at 50 frames per second and EM gain level 100. Live images were acquired using a 40×/1.0NA long working distance, water-immersion objective (W Plan-Apochromat 40×/1.0 DIC M27), a 2× C-mount fixed focal length lens extender, and an Andor EMCCD camera (iXon Ultra model 897_BV, 512 × 512 pixels, Andor Technology). Imaging data were acquired using Micro-Manager (Edelstein et al., 2014) and saved as multilayer TIFF files.

### Image analysis and figure preparation
Confocal stack images (.nd2) were processed with Fiji ImageJ (Schindelin et al., 2012). Selection of specific z-stack ranges, generation of maximum intensity projections, brightness adjustment (identical for all images within the same experiment), and/or image cropping were performed as required. Thresholding of the resulting images was used to select regions of interest (ROIs) for quantification. For overall axon and NMJ structures, thresholding was manually adjusted on the PM marker channel (when present), and the resulting ROIs were applied to the remaining channels. For punctate signals, to reduce variability in ROI selection between samples, one of the predefined Fiji threshold algorithms was chosen, based on its ability to filter puncta previously identified by eye in multiple random images. The selected ROIs were used to measure mean pixel intensity and/or area values, which were recorded in a Microsoft Excel 2019 file (.xlsx). Files generated during successive steps in image processing, the threshold algorithms used in each experiment (when required), the threshold values used for all ROIs, and the quantification datasets, can be found in the underpinning dataset for this paper.

Wide-field multilayer .tif time course files were opened in Fiji ImageJ (Schindelin et al., 2012), and channels were background subtracted (Rolling Ball Radius: 50 pixels). NMJs were stabilized using Register ROI from the cookbook menu with an ROI containing the entire NMJ. Some NMJs needed to be registered multiple times. Fluorescent intensity time traces were obtained for ROIs traced around the entire NMJ, or around all individual mitochondria in the NMJ, in each .tif frame in a given data set using the Time Series Analyzer V3 plugin and ROI manager, recording the average fluorescence in each ROI (or across all mitochondria ROIs). Mitochondria or NMJs that moved and could not be corrected with registration software, or drifted out of focus during recording, were discarded from analysis. If only one or two mitochondria/boutons in the NMJ were available for analysis, due to focus or drift, data were discarded from analysis. Miniature neurotransmission frequencies were manually counted (Fig. S8) at the distal bouton over a 20-s interval. Miniature neurotransmission amplitude was calculated via a maximum ΔF/F value reached at the distal bouton over a 20-s

interval in R. Time traces were saved as CSV files and NMJ averages were fed into R scripts (R Core Team, 2020) to perform bleach correction and rolling averages (5 frames). Bleach correction was performed by fitting a bleach correction power curve ($PC = a*x^b$) to pre-stimulation fluorescence (a and b coefficients calculated in R) and dividing raw fluorescence over the entire time course by this fitted curve. R scripts were also used to collect resting fluorescence (immediate pre-stimulation value of the fitted power curve), maximum fluorescence (maximum raw fluorescent value), $\Delta F$, $\Delta F/F$, time to peak, time to 50% recovery, and with the cytosolic sensor, time to 100% recovery. All files generated during successive steps in image processing, ROIs, and the quantification analyses, are available in the dataset for this paper.

## Electron microscopy
### Fixation and embedding
Fixation of dissected third instar larvae was done by replacing HL3 solution with fix solution (0.05 M sodium cacodylate at pH 7.4 containing 4% formaldehyde, 2% vacuum distilled glutaraldehyde, and 2 mM $CaCl_2$). The larvae were incubated at 4°C overnight. After fixation the preparations were washed three times for 10 min each at 4°C using 0.05 M cold cacodylate buffer. Samples were osmicated in 1% osmium tetroxide/1.5% potassium ferricyanide/0.05 M sodium cacodylate buffer for 3 d at 4°C. The preparations were washed four times with deionized water at room temperature. Thiocarbohydrazide solution was prepared at a concentration of 0.1% in deionized water, incubated in a 60°C oven for 1 h while agitated by swirling every 10 min to facilitate dissolution, and filtered through two 9-cm filter papers just before use. The preparations were then incubated in thiocarbohydrazide solution for 20–30 min at room temperature and covered with aluminum foil to protect from light. They were then rinsed with deionized water at room temperature five times for 3 min, incubated in 2% aqueous osmium tetroxide for 30–60 min at room temperature, and rinsed with deionized water at room temperature five times for 3 min. Afterward, the preparations were incubated in 2% uranyl acetate (maleate-buffered to pH 5.5) at 4°C for 3 d and rinsed with deionized water at room temperature 5 times for 3 min. They were then incubated in lead aspartate solution (0.66 $g$ of lead nitrate was dissolved in 100 ml 0.03 M aspartic acid solution and pH adjusted to 5.5 with 1 M KOH in a 60°C oven for 30 min) at 60°C for 30 min, and rinsed with deionized water at room temperature five times for 3 min. They were then dehydrated twice with each of 50, 70, 90, and 100% ethanol and twice with dried ethanol, twice with dried acetone, and twice with dry acetonitrile. The preparations were incubated in 50/50 acetonitrile/Quetol 651 overnight at room temperature. They were then incubated 72 h in Quetol epoxy resin 651 (Agar Scientific) without BDMA and five times for 24 h in Quetol epoxy resin 651 with BDMA (dimethylbenzylamine). Afterward, they were incubated at 60°C for a minimum of 48 h.

### Sectioning
The resin blocks were sectioned using a Leica Ultracut E ultramicrotome. Sections were mounted on aluminum SEM stubs using carbon sticky pads and coated with 30 nm carbon for conductivity. The sections were imaged in a Verios 460 SEM at 4 keV and 0.2 nM probe current using the concentric backscatter detector at low magnification, and stitched image maps were acquired using MAPS automated acquisition software to give an overview through the central region of the larva. Once NMJs were identified, blocks were trimmed down to the region containing the structures of interest. The samples were then sectioned with a 4 mm UltraMaxi Diatome 35° knife using an ATUMtome (RMC/Boeckeler Instruments) at a thickness of 50 nm and were collected onto rolling Kapton tape.

### Imaging
Kapton tape strips were mounted on 4-in silicon wafers using double-sided carbon sticky tape. Wafers were then sputter coated with 30 nm carbon using a Quorum Q150 T E carbon coater. Wafers were imaged in a Verios 460 scanning electron microscope (FEI/Thermo Fisher Scientific) at 4 keV accelerating voltage and 0.2 nA probe current in backscatter mode using the concentric backscatter detector (CBS) in immersion mode at a working distance of 3.5–4 mm; 1,536 × 1,024 pixel resolution, 3 μs dwell time, four line integrations. Stitched maps were acquired using FEI MAPS software using the default stitching profile and 5% tile overlap. Images of serial sections were imported into a single file using ImageJ/Fiji (Schindelin et al., 2012), and sections were aligned and reconstructed using the TrakEM2 plugin (Cardona et al., 2012). Reconstructions were further processed with eight smoothing steps of Laplacian smooth function in the program MeshLab (http://meshlab. sourceforge.net). The rendering file generated is available in the underpinning data for the paper.

## Statistical analysis
After quantification, values were exported from .xlsx or .CSV to .txt files to be analyzed with R Studio 1.3.1093 or GraphPad Prism 9 for statistical analysis and plotting. Figures were made using Adobe Illustrator CC 2017 or Affinity Designer 1.10.0. All R scripts were written by the authors and are available in the underpinning data for the paper.

A Shapiro-Wilk test was used to test for normality in the data distribution, and Levene's test was used to test for differences between group variances. Normally distributed data were analyzed with mixed-effects model repeated-measures ANOVA or unpaired two-tailed Student's $t$ tests. Data not normally distributed were analyzed using non-parametric Kruskal–Wallis or Mann–Whitney U-tests. Post-hoc multiple comparisons were then applied where relevant. Tukey HSD test was used to compare each group with the other groups, while Dunnett's test was used for planned comparisons between every group and a single control group.

## Online supplemental material
Fig. S1 shows CRISPR-derived lesions in *Rtnl1* and their effects on the *Rtnl1* coding region. Fig. S2 shows *Rtnl1* expression in *Rtnl1* mutant CRISPR alleles. Fig. S3 shows high-magnification confocal planes of the presynaptic ER network organization in *Rtnl1* mutants. Fig. S4 (tdTom::Sec61β) and Fig. S5 (GFP::HDEL) show

extended data on the effects of *Rtnl1* mutant alleles on presynaptic ER distribution. Fig. S6 shows presynaptic PI(4,5)P$_2$ levels in *Rtnl1* mutants. Fig. S7 shows synaptic transmission in *Rtnl1* mutants. Fig. S8 shows images of miniature neurotransmission in *Rtnl1* mutants. Figs. S9 and S10 show extended data on cytosolic Ca$^{2+}$ handling in Is and Ib boutons, respectively, of *Rtnl1* mutants. Fig. S11 shows extended data on ER Ca$^{2+}$ handling in Is and Ib boutons of *Rtnl1* mutants. Figs. S12 and S13 show extended data on mitochondrial Ca$^{2+}$ handling in Is and Ib boutons, respectively, of *Rtnl1* mutants. Video 1 (*WT*) and Video 2 (*Rtnl1*) show postsynaptic Ca$^{2+}$ response to low-frequency stimulation. Video 3 (*WT*) and Video 4 (*Rtnl1*) show postsynaptic miniature Ca$^{2+}$ events. Video 5 (*WT*) and Video 6 (*Rtnl1*) show cytosolic Ca$^{2+}$ response to stimulation. Video 7 (*WT*) and Video 8 (*Rtnl1*) show ER Ca$^{2+}$ response to stimulation in Is boutons. Video 9 (*WT*) and Video 10 (*Rtnl1*) show ER Ca$^{2+}$ response to stimulation in Ib boutons. Video 11 is a brighter version of Video 10. Video 12 (*WT*) and Video 13 (*Rtnl1*) show mitochondria Ca$^{2+}$ response to stimulation in Ib boutons. Video 13 shows *Rtnl1* mutant mitochondria Ca$^{2+}$ response to stimulation in Ib boutons. Table S1 shows the *Drosophila* stocks used in this work. Table S2 shows the crosses used in this work. Table S3 shows the reagents used in this work.

### Data availability

The underpinning dataset for this article is openly available at the University of Cambridge Data Repository (https://www.repository.cam.ac.uk): https://doi.org/10.17863/CAM.93878.

## Acknowledgments

We thank Dion Dickman, Xun Huang, Andrea Daga, Guy Tear, Addgene, the Developmental Studies Hybridoma Bank, the Bloomington and Kyoto *Drosophila* Stock Centers, and the University of Cambridge Department of Genetics Fly Facility for antibodies, constructs, and stocks. We thank the University of Cambridge Department of Genetics Fly Facility for embryo injections and the electron microscopy facility of the Cambridge Advanced Imaging Centre. We thank Beatriz Ibañez for helpful discussions, and the HSP Tom Wahlig Foundation for its support and dissemination of our work.

This work was supported by grants from the Medical Research Council (MRC) (MR/S011226/1), the Biotechnology and Biological Sciences Research Council (BBSRC; BB/S001212/1), and the Spastic Paraplegia Foundation, Inc. (SPF) to C.J. O'Kane. J.J. Pérez-Moreno was supported by the SPF, a Marie-Sklodowska-Curie grant from the European Union (745007), and a Juan de la Cierva Incorporación grant (IJC2019-038819-I) from the Spanish State Research Agency (MCIN/AEI/10.13039/501100011033). M.K. Oliva was supported by a Marie-Sklodowska-Curie grant from the European Union (660516), and S. Ojha by the SPF.

Author contributions: J.J. Pérez-Moreno: Conceptualization, Resources, Formal analysis, Funding acquisition, Investigation, Visualization, Writing—original draft, Writing—review and editing; generated and characterized new *Rtnl1* alleles, performed and analyzed most confocal microscopy, participated in the generation and imaging of EM samples. R.C. Smith: Conceptualization, Resources, Formal analysis, Investigation, Visualization, Writing—original draft, Writing—review and editing; performed and analyzed most Ca$^{2+}$ imaging, performed and analyzed confocal microscopy. M.K. Oliva: Conceptualization, Resources, Formal analysis, Investigation, Visualization, Writing—review and editing; generated and characterized GCaMPs, performed Ib ER Ca$^{2+}$ imaging. S. Ojha: Serial EM analysis. F. Gallo: generation of EM samples and serial EM sectioning, Writing—review and editing; K.H. Müller: EM imaging, Writing—review and editing. CJO'K: Conceptualization, Resources, Formal analysis, Supervision, Funding acquisition, Writing—original draft, Project administration, Writing—review and editing.

Disclosures: The authors declare no competing interests exist.

Submitted: 22 December 2021

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

**Supplemental material**

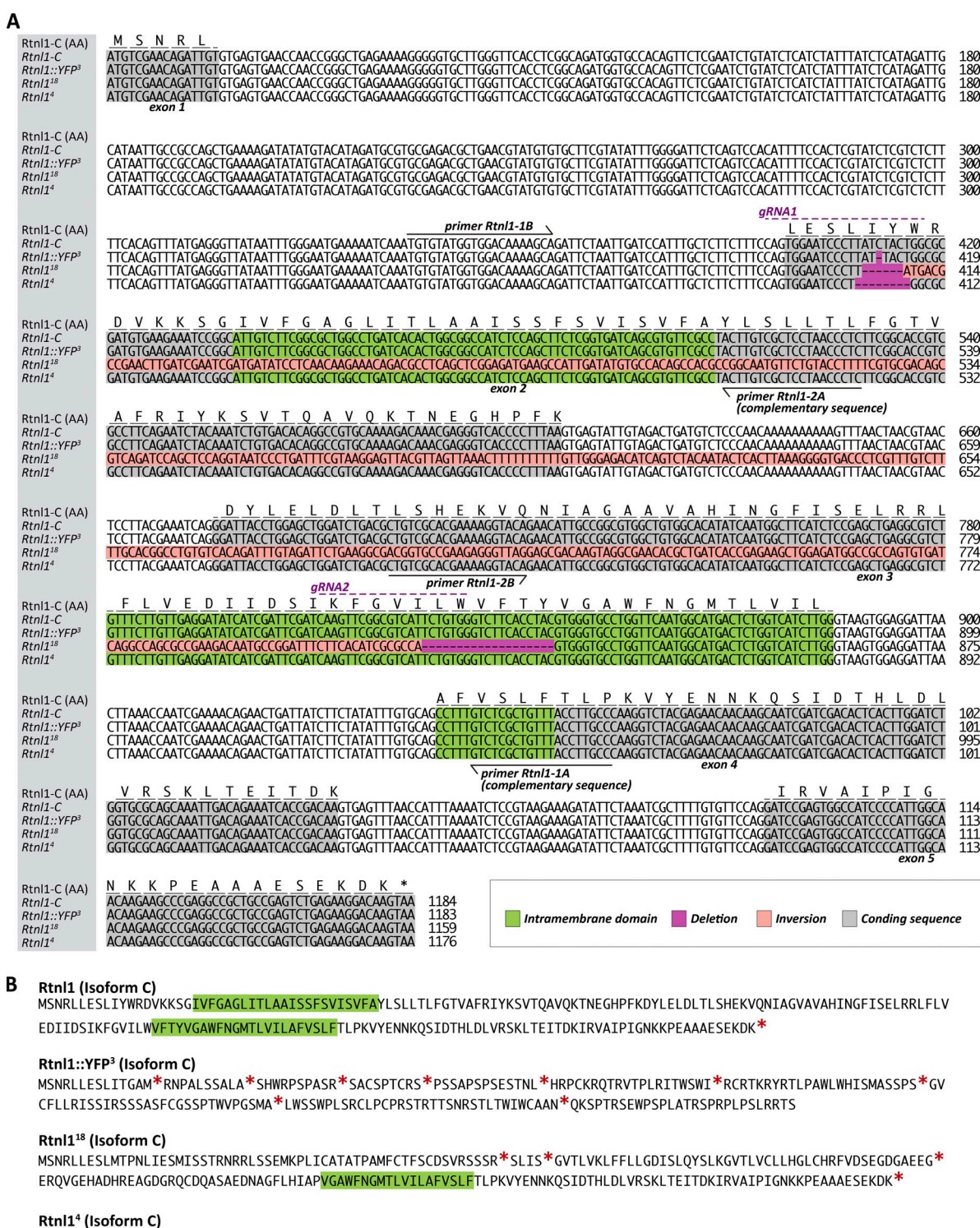

Figure S1. **CRISPR-derived lesions in *Rtnl1* and their effects on the *Rtnl1* coding region. (A)** Manual alignment of the *Rtnl1* transcript C for *WT* (*Rtnl1-C*), and *Rtnl1::YFP[3]*, *Rtnl1[18]* and *Rtnl1[4]* mutant CRISPR alleles, showing the amino acids (AA) encoded by the *WT* (*Rtnl1-C*) allele, the location of the gRNAs used to generate the mutant alleles, and the primers used to genotype them (see Materials and methods for details). Since the sequences encoding intramembrane domains are shared by all *Rtnl1* isoforms, the shortest one, isoform C, was chosen for convenience. Sequences read from 5' to 3'. The *Rtnl1::YFP[3]* allele was generated from *Rtnl1::YFP[CPTI00291]*, has a 1-bp deletion/frameshift at the position of gRNA1 upstream of the first intramembrane domain, and lacks detectable YFP expression (Fig. S2); we did not use it further in this work, but it shows that frameshifts around this position can lead to loss of protein expression. The effects of *Rtnl1[18]* on the protein-coding sequence are described in the main paper. *Rtnl1[4]* has an 8-bp deletion/frameshift at the position of gRNA1, upstream of the first intramembrane domain. **(B)** Predicted Rtnl1-C protein sequences for the *Rtnl1* alleles shown in A. Intramembrane domains are decorated in green and stop codons shown with a red asterisk.

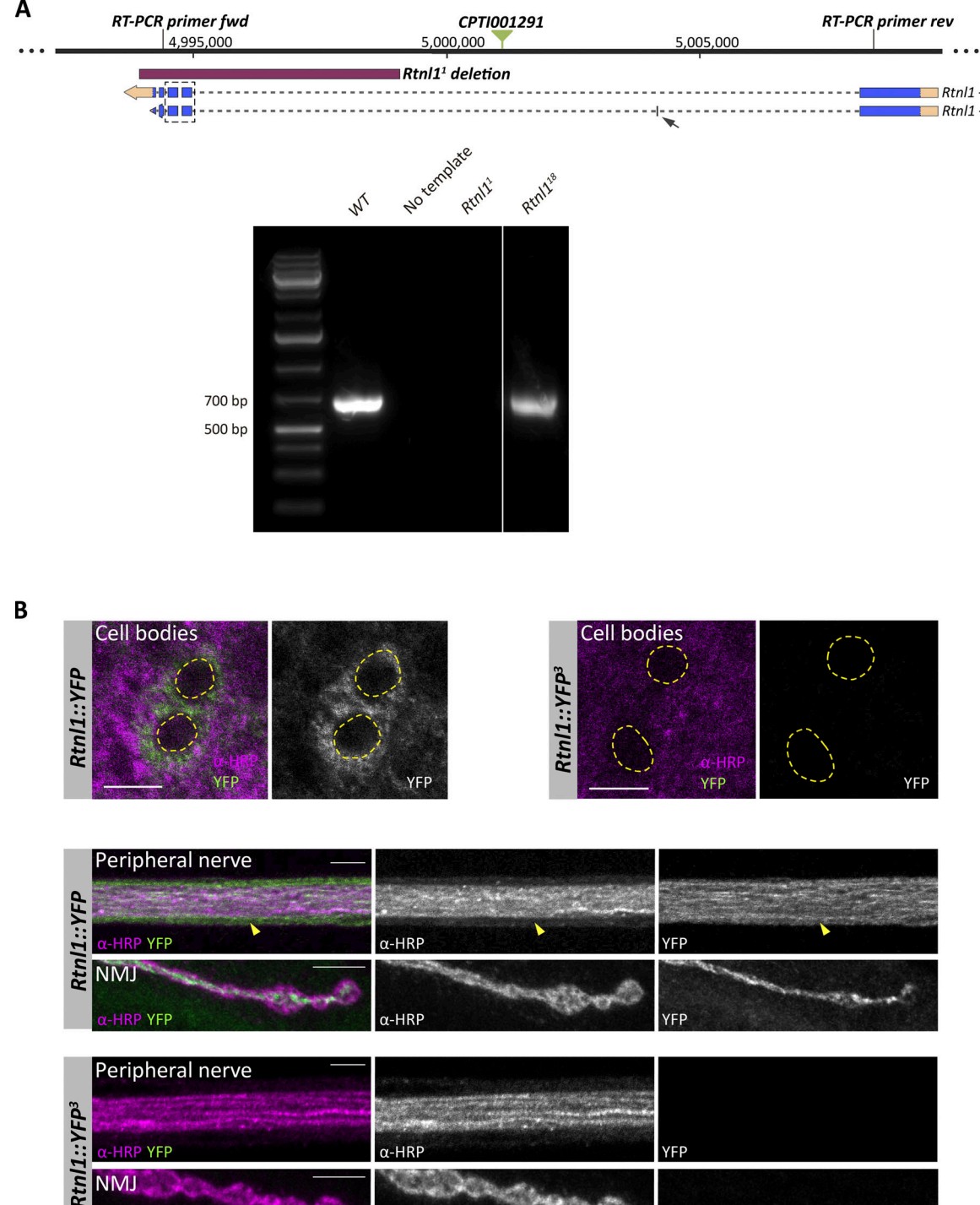

Figure S2. **Effects of *Rtnl1* mutant CRISPR alleles on *Rtnl1* expression. (A)** RT-PCR strategy to test presence of *Rtnl1* transcripts (top). For the *Rtnl1* isoforms shown, blue boxes indicate exons, broken lines indicate introns, light orange boxes indicate 5' and-3' UTRs, and arrows indicate the direction of transcription. A green triangle shows the position of the *Rtnl1::YFP* exon trap insertion (*CPTI001291*). Genomic coordinates are based on Release 6.26 of the *D. melanogaster* genome (http://flybase.org). Using the indicated primers, the expected amplicon from genomic DNA is around 14 kb. The expected amplicon for *Rtnl1* transcript *F* is 636 bp, and for transcript *H* is 672 bp, due to an extra small exon (arrow). The box surrounded by a broken line shows the region containing *Rtnl1[18]* lesions. Agarose gel electrophoresis (1%) of reverse-transcribed cDNA (bottom) shows that *Rtnl1[18]* does not disrupt *Rtnl1* mRNA transcription. For both WT and *Rtnl1[18]* alleles, only one amplicon is seen, which is slightly smaller in *Rtnl1[18]*, due to the small deletions totaling 25 bp within the amplified region (see Fig. 1 A and Fig. S1 A for details). **(B)** Confocal sections of neuronal cell bodies, peripheral nerves and NMJs (muscle 1) show that *Rtnl1::YFP[3]* mutation is enough to abolish Rtnl1::YFP expression. In cell bodies, nuclei are distinguished by the absence of α-HRP signal (dotted line regions). On the peripheral nerve, glia can be distinguished from neuronal axons due to the low levels of α-HRP signal (arrowhead). All larvae are also expressing *Ib-GAL4*, but this is not driving any reporter expression. Scale bars, 5 µm. Source data are available for this figure: SourceData FS2.

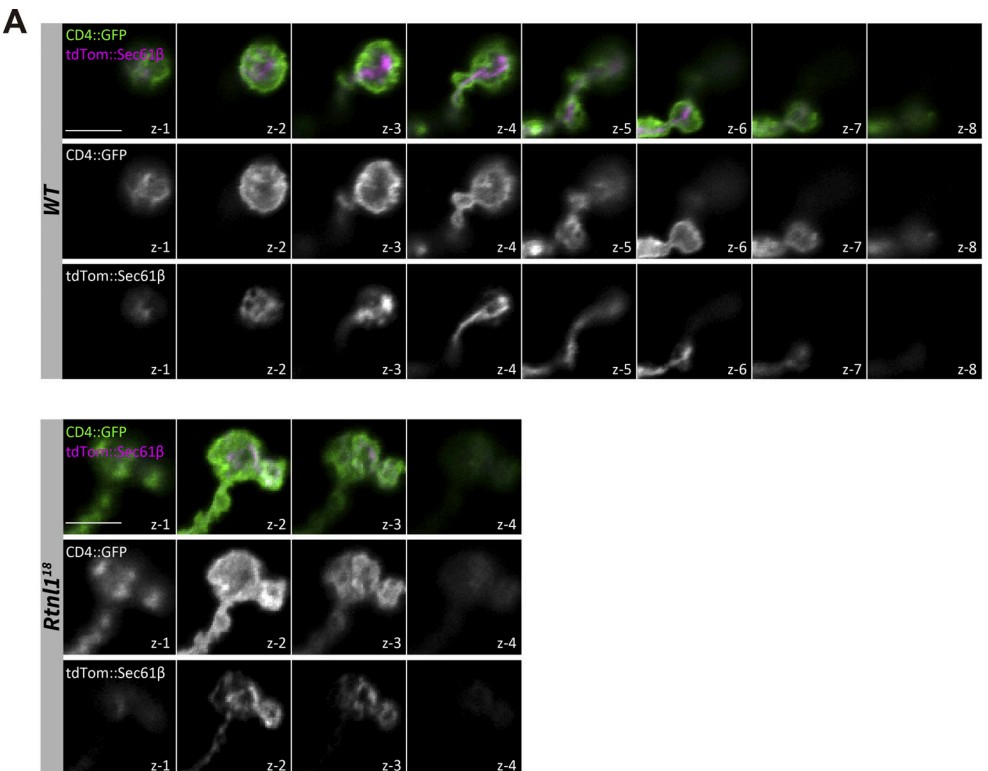

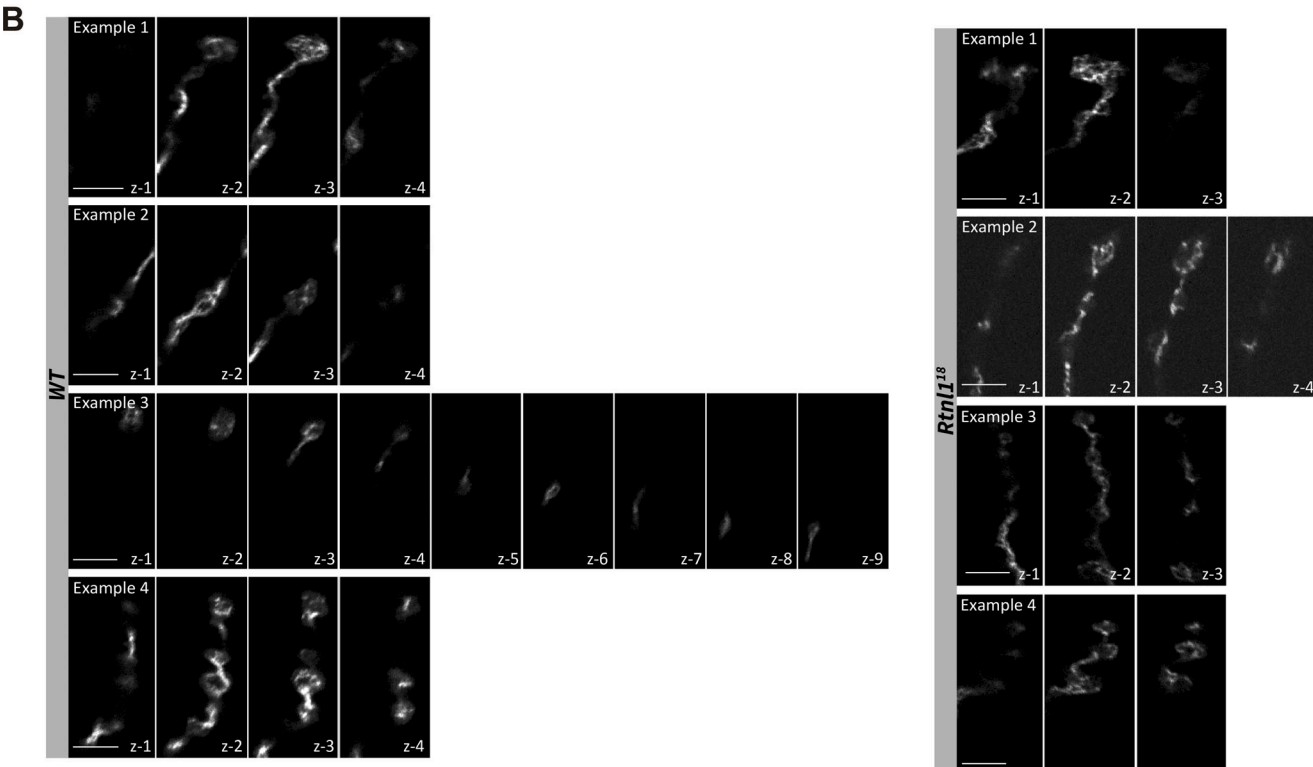

Figure S3. **Rtnl1 loss does not noticeably disrupt presynaptic ER network organization. (A)** Confocal sections (z steps, 1 µm) of the projections presented in the magnified areas of Fig. 2 A, showing the distribution of the ER marker tdTom::Sec61β in Type Ib muscle 1 NMJ of *WT* and *Rtnl1[18]* larvae. CD4::GFP labels the plasma membrane. Genotypes are *Ib-GAL4, UAS-CD4::tdGFP/UAS-tdTom::Sec61β*, in either a *WT or Rtnl1[18]* background. **(B)** Confocal sections (z steps, 1 µm) of the projections presented in the magnified areas of Fig. 2 E, showing the distribution of the ER marker Sturkopf::GFP in Type Ib muscle 1 NMJ of *WT* and *Rtnl1[18]* larvae. Genotypes are *Ib-GAL4, UAS-Sturkopf::GFP/+*, in either a *WT* or *Rtnl1[18]* background. Scale bars, 5 µm.

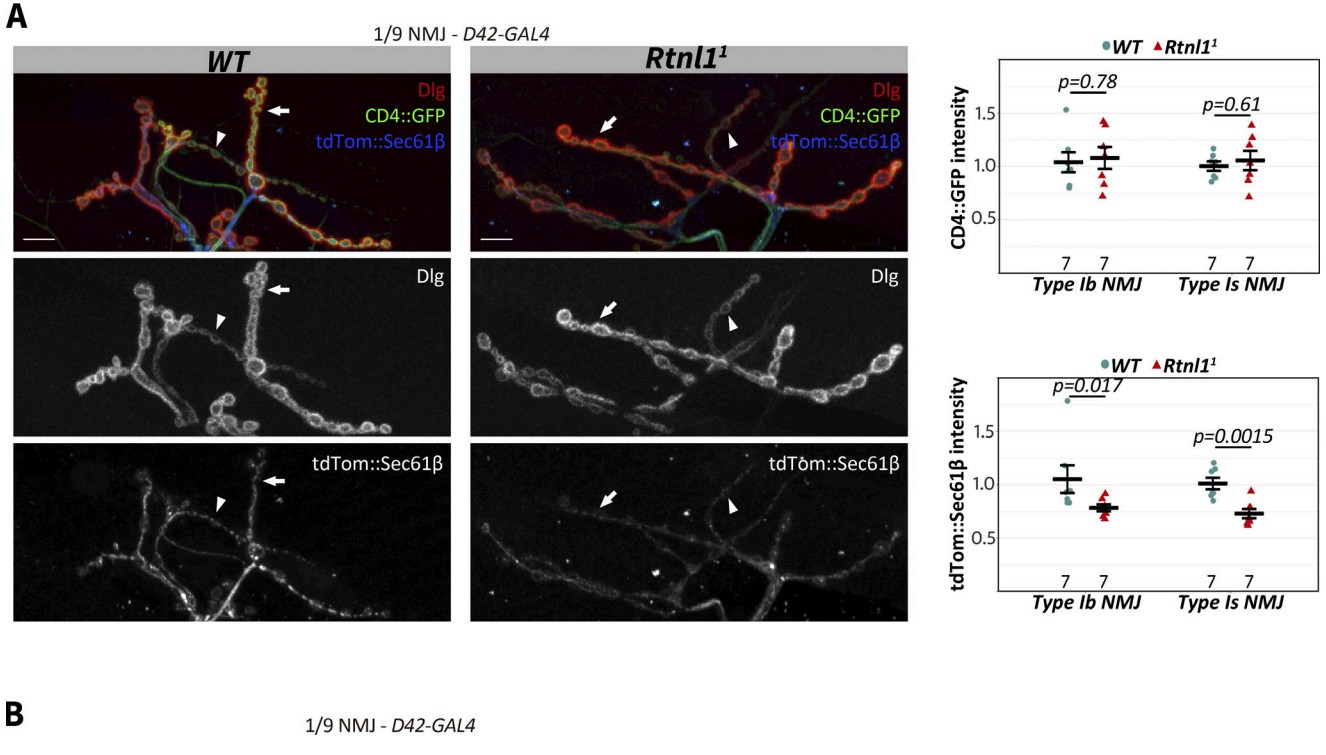

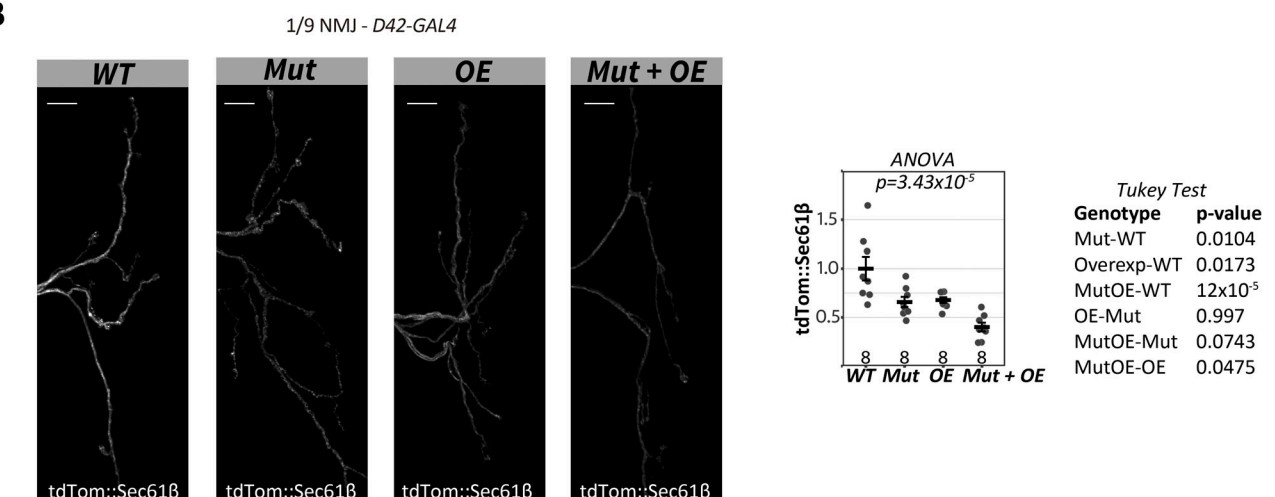

Figure S4.  **Effects of *Rtnl1* mutant alleles on presynaptic tdTom::Sec61β levels. (A)** Representative examples of confocal projections and quantifications of *WT* and *Rtnl1[1]* larvae showing the distribution of the ER marker tdTom::Sec61β in Type I NMJs (muscles 1/9 NMJ). Samples were immunostained for tdTom, GFP, and Dlg to distinguish between Type Ib (arrows) and Type Is (arrowheads) NMJs. Genotypes: *D42-GAL4, UAS-tdTom::Sec61β, UAS-CD4::tdGFP/+*, in either *WT* or *Rtnl1[1]* background. **(B)** Representative examples of confocal projections and quantification of *WT*, *Mut*, *OE*, and *Mut + OE* larvae showing the distribution of the ER marker tdTom::Sec61β in Type I NMJs (muscles 1/9 NMJ). Genotypes: *WT - Rtnl1+; D42-GAL4, UAS-tdTom::Sec61β/+, Mut - Rtnl1[18]/Rtnl1[1]; D42-GAL4, UAS-tdTom::Sec61β/+ OE - Rtnl1+; D42-GAL4, UAS-Rtnl1::GFP/UAS-tdTom::Sec61β Mut + OE-Rtnl1[18]/Rtnl1[1] ; D42-GAL4, UAS-Rtnl1::GFP/UAS-tdTom::Sec61β*. For A and B, plots show individual larval datapoints and mean ± SEM; y-axis indicates arbitrary units (au) after normalization to control (*WT*); sample size (number of larvae) is indicated within the plots for each genotype. For each larva, several NMJs between A2-A6 segments were analyzed, and the mean value is shown as a larval datapoint. Student's *t* tests were performed for pairwise comparisons, except for tdTom::Sec61β intensity in Type Ib NMJ comparison (A), where a Mann-Whitney U test was performed. Scale bars, 10 μm.

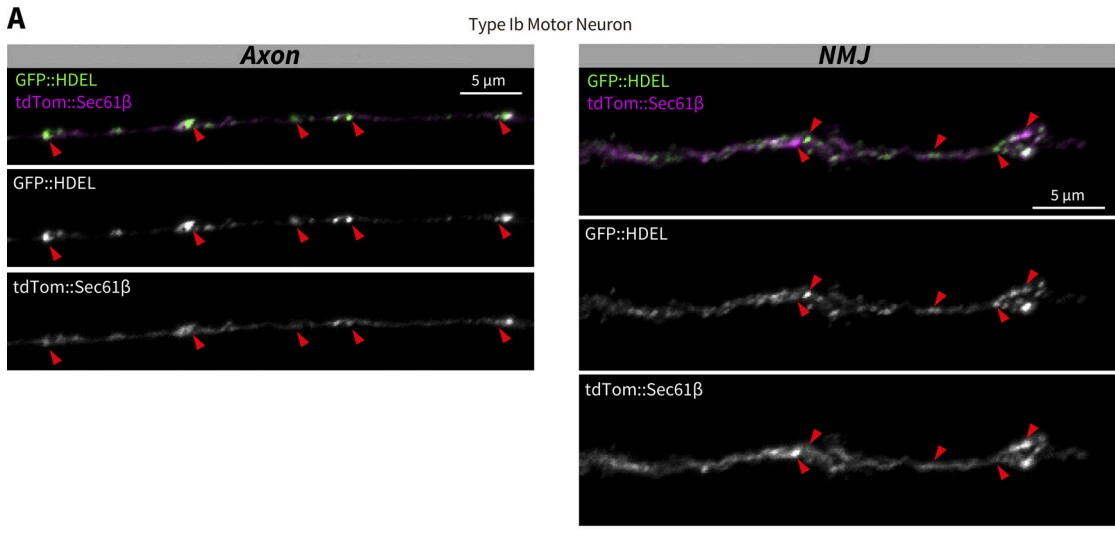

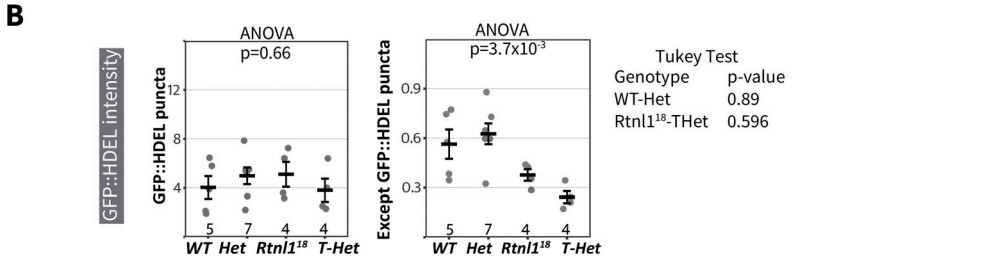

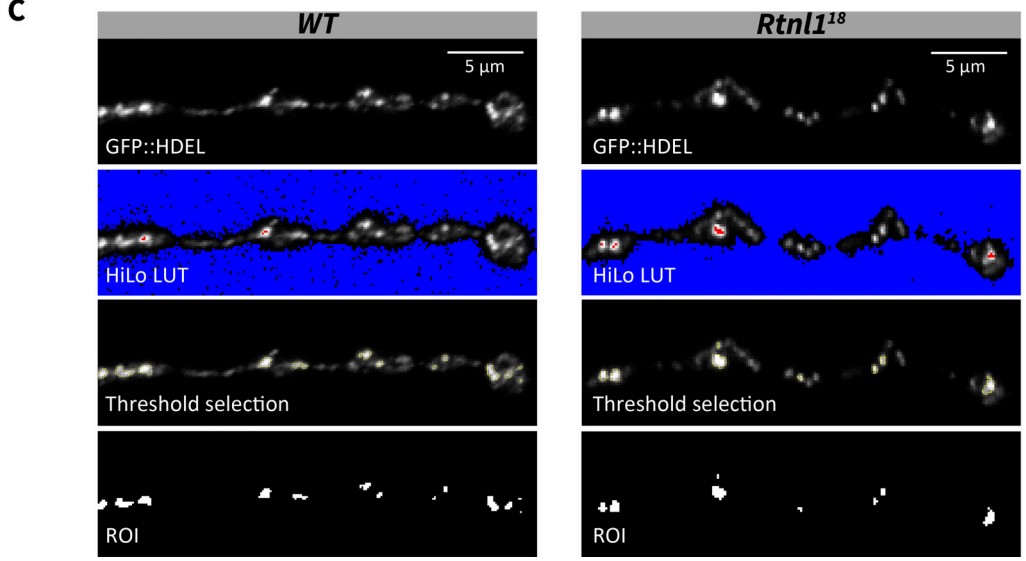

**Figure S5. GFP::HDEL distribution. (A)** Representative examples of confocal projections of *WT* larvae showing the distribution of the ER markers tdTom:: Sec61β and GFP::HDEL in a Type Ib motor neuron. Genotype: *Rtnl1+/UAS-tdTom::Sec61β; Ib-GAL4/UAS-GFP::HDEL.* **(B)** Quantification of the ER marker GFP:: HDEL in Type I NMJs (muscles 1/9 NMJ) of *WT, Het, Rtnl1[18]*, and *T-Het* larvae. Genotypes: *WT—Rtnl1+; Ib-GAL4, UAS-CD4::tdTom/UAS-GFP::HDEL Het - Rtnl1+/ Rtnl1[18]; Ib-GAL4, UAS-CD4::tdTom/UAS-GFP::HDEL Rtnl1[18] - Rtnl1[18]; Ib-GAL4, UAS-CD4::tdTom/UAS-GFP::HDEL T-Het - Rtnl1[18]/Rtnl1[1] ; Ib-GAL4, UAS-CD4::tdTom/ UAS-GFP::HDEL.* Plots show individual larval datapoints and mean ± SEM; y-axis indicates arbitrary units (au) after normalization to control (*WT*); intensity values are relative to CD4::tdTom signal; sample size (number of larvae) is indicated within the plots for each genotype. For each larva, several NMJs between A2-A6 segments were analyzed, and the mean value is shown as a larval datapoint. **(C)** Representative examples showing the selection of GFP::HDEL puncta in confocal projections of Type Ib muscle 1 NMJ (magnified areas from Fig. 3 C). HiLo lookup table (LUT) shows saturated pixels in red and pixels with no detectable signal in blue. Intermodes thresholding of GFP::HDEL intensity was used to select only those regions with high GFP::HDEL levels (region of interest, ROI). Genotypes are *Ib-GAL4, UAS-CD4::tdTom/UAS-BiP::sfGFP::HDEL*, in either a *Rtnl1+* (*WT*) or *Rtnl1[18]* background.

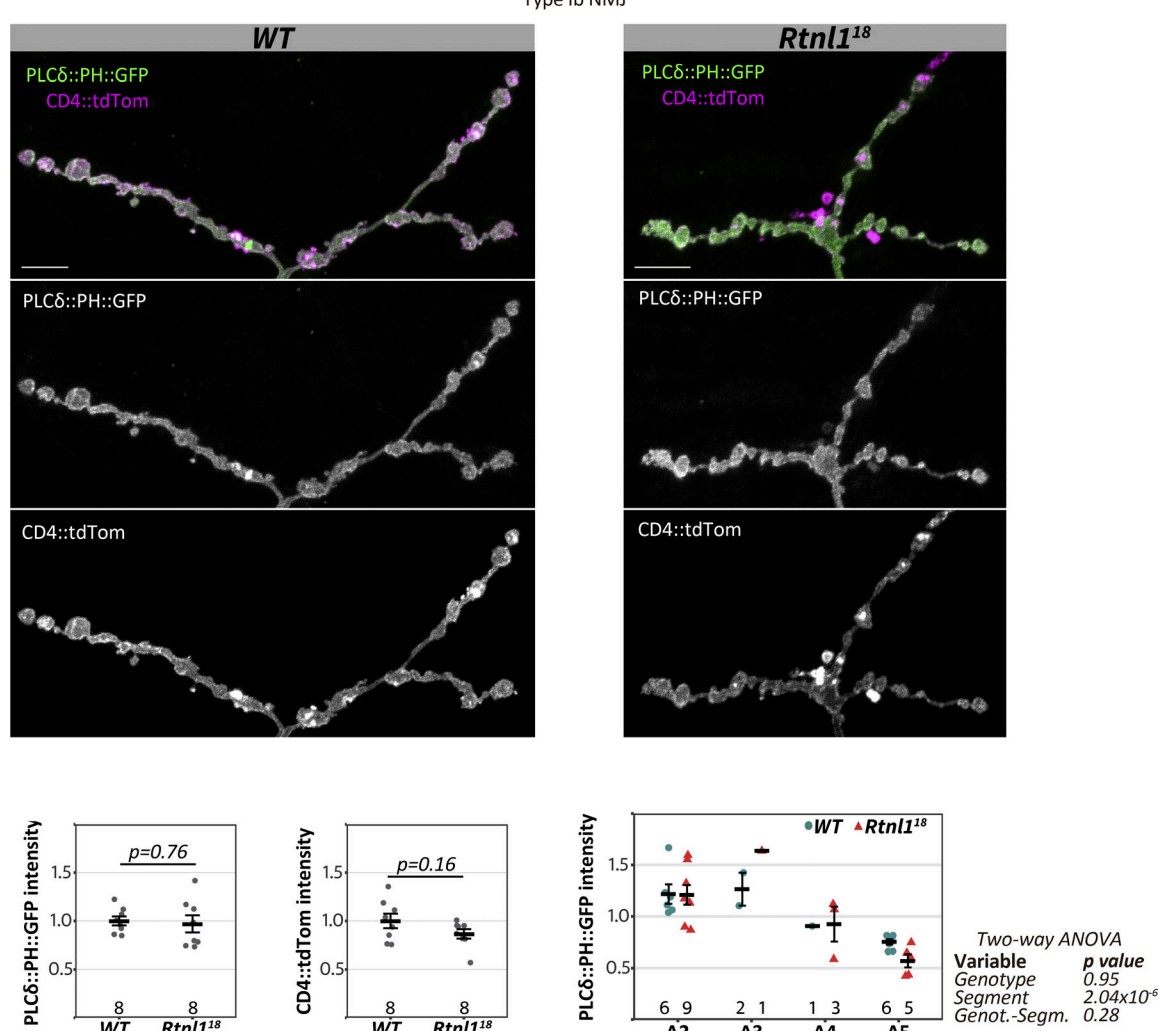

Figure S6. **Rtnl1 loss does not affect presynaptic PI(4,5)P$_2$ levels.** Representative examples of confocal projections and quantifications of *WT and Rtnl1[18]* larvae showing the distribution of the PI(4,5)P$_2$ marker PLCδ::PH::GFP in Type Ib muscle 1 NMJ. Scale bars, 10 μm. Plots show individual larval datapoints and mean ± SEM; y-axis indicates arbitrary units (au) after normalization to control (*WT*); sample size (larvae) is indicated within the plot for each genotype. For each larva, several NMJs between A2-A6 segments were analyzed, and the mean larval value is shown as a datapoint. Student's *t* tests were performed for pairwise comparisons. Genotypes are *Ib-GAL4, UAS-CD4::tdTom/UAS-PLCδ::PH::GFP*, in either a *Rtnl1+* (*WT*) or *Rtnl1[18]* background.

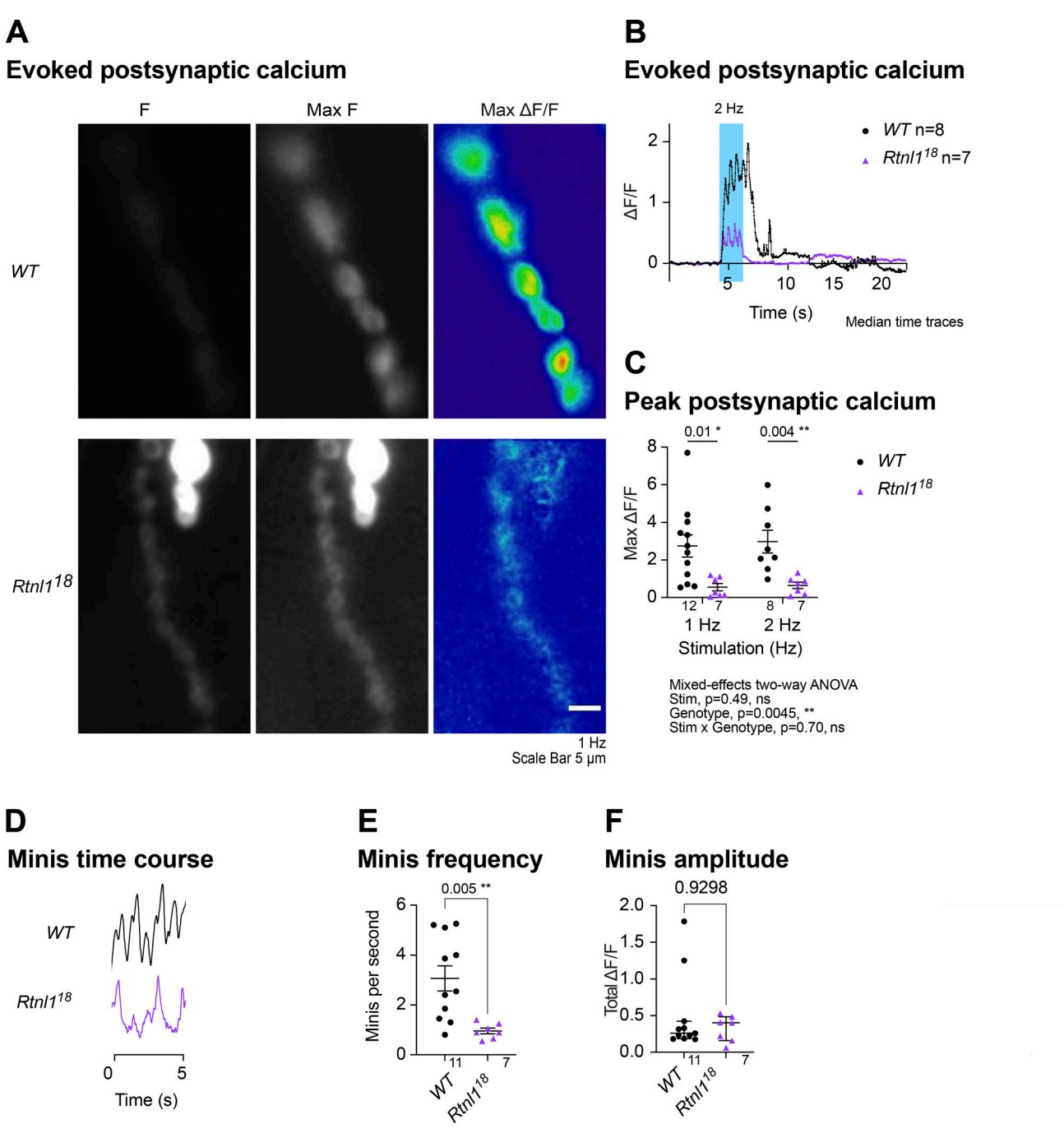

**A**

**Evoked postsynaptic calcium**

F　　　Max F　　　Max ΔF/F

*WT*

*Rtnl1$^{18}$*

1 Hz
Scale Bar 5 μm

**B**

**Evoked postsynaptic calcium**

2 Hz

- *WT* n=8
- *Rtnl1$^{18}$* n=7

Median time traces

**C**

**Peak postsynaptic calcium**

0.01 *　　0.004 **

- *WT*
- *Rtnl1$^{18}$*

12　7　　8　7

Stimulation (Hz)

Mixed-effects two-way ANOVA
Stim, p=0.49, ns
Genotype, p=0.0045, **
Stim x Genotype, p=0.70, ns

**D**

**Minis time course**

*WT*

*Rtnl1$^{18}$*

Time (s)

**E**

**Minis frequency**

0.005 **

11　7

Unpaired t test

**F**

**Minis amplitude**

0.9298

11　7

Mann-Whitney

Figure S7.　**Rtnl1 loss impacts synaptic transmission. (A)** GCaMP fluorescence at rest (F), maximum fluorescence (Max F) in response to 1 Hz stimulation, and maximum change in fluorescence (Max ΔF) in examples of *WT* and *Rtnl1$^{18}$* at muscle 1, Type Is postsynaptic terminals. **(B)** Impact of Rtnl1 loss-of-function on peak evoked postsynaptic Ca$^{2+}$. Plot shows the median responses to a burst of 2 Hz stimulation of larvae from each genotype. **(C)** Impact of Rtnl1 loss-of-function on peak evoked postsynaptic Ca$^{2+}$ responses. Plot shows individual larval datapoints and mean ± SEM; datapoints represent the largest ΔF/F reached after either a 1 Hz or 2 Hz stimulation during the recording. Comparisons were analyzed with a mixed-effects two-way ANOVA. **(D)** Impact of Rtnl1 loss-of-function on postsynaptic resting miniature responses (minis). Plots show representative time traces (close to the mean stimulation frequency) over 5 s from the distal bouton during the recording collected. **(E)** Impact of Rtnl1 loss-of-function on minis frequency. Plot shows individual larval datapoints and mean ± SEM. Frequency datapoints represent the number of minis per second over a 5-s recording. Pairwise comparison was performed using Student's *t* test. **(F)** Rtnl1 loss-of-function did not affect minis amplitude. Plot shows individual larval datapoints and median ± interquartile ranges. For each larva in E and F, minis from a 20-s recording from the distal bouton of one NMJ between segments A4-A6. Amplitude datapoints represent the largest ΔF/F over the recording. Pairwise comparison was performed using Mann–Whitney U-test (maximum mini amplitude). **(B–F)** Sample size (larvae) is indicated within each plot for each genotype. Each recording was from one muscle 1, Type Is NMJ between segments A4-A6. Genotypes are *Is-GAL4, mhc-SynapGCaMP6f/UAS-tdTom::Sec61β*, in either a *WT* or *Rtnl1$^{18}$* background.

*WT*

*Rtnl1*[18]

100 frames in 8 seconds, 12.5 frames/second

Scale bar 5 μm

Figure S8. **Rtnl1 loss decreases miniature neurotransmission frequency.** (Extended data from Fig. S7, D–E.) Representative examples (close to the median frequency) of miniature events in 100 frames over 8 s (12.5 frames/s). Panels show time-lapse GCaMP fluorescence at rest in distal boutons of muscle 1, Type Is postsynaptic terminals, in *WT* (two boutons) and *Rtnl1*[18] (four boutons). Arrows indicate miniature events counted. Genotypes are *Is-GAL4, mhc-SynapGCaMP6f/UAS-tdTom::Sec61β*, in either a *WT* or *Rtnl1*[18] background.

Figure S9. **Rtnl1 loss decreases cytosolic Ca²⁺ handling in Type Is boutons.** (Extended data from Fig. 5.) **(A)** Loss of Rtnl1 does not affect resting myrGCaMP6s fluorescence. *Rtnl1¹⁸/Rtnl1¹* flies show a decrease in myrGCaMP6s fluorescence, but this effect is not replicated in a homozygous *Rtnl1¹⁸* background, indicating that loss of Rtnl1 does not affect resting cytosolic Ca²⁺. The plot shows individual larval datapoints and median ± interquartile ranges; sample size (larvae) is indicated within the plot for each genotype. The pairwise comparison performed was a Kruskal–Wallis test. **(B)** Impact of Rtnl1 loss-of-function on peak evoked cytosolic Ca²⁺ responses. Plots show all single time traces, as well as mean ± SEM time traces in every genotype for six stimulation frequencies tested. Sample size is indicated within the plot for each genotype. **(C)** Rtnl1 loss of function decreases peak evoked cytosolic Ca²⁺. Datapoints represent mean ± SEM of the largest ΔF/F reached after stimulation during the recording, shown in B. **(D)** Rtnl1 loss did not affect time to peak cytosolic Ca²⁺. Datapoints represent the time between stimulation and peak ΔF/F. Comparisons were analyzed as in C. **(E and F)** Rtnl1 loss did not affect time recovery from evoked cytosolic Ca²⁺ influx. Although the mixed-effects two-way ANOVA records a significant genotypic effect in both time to 50% recovery (E) and 100% recovery (F), two-way ANOVAs without the *Rtnl1¹⁸* group erase this significance, indicating that *Rtnl1¹⁸/Rtnl1¹* transheterozygotes are not significantly different from controls in these assays and that the effect is likely due to the homozygosity of the chromosome carrying *Rtnl1¹⁸*, and not to the loss of Rtnl1. Datapoints represent the time between peak ΔF/F and 50% or 100% recovery. Comparisons were analyzed as in C. In C–F, plots show mean ± SEM of every genotype for each stimulation frequency tested; sample size (larvae) is indicated within the plot for each genotype. For each larva, responses from a 20 s recording from one muscle 1 NMJ between segments A4-A6 were analyzed. Comparisons were analyzed with a mixed-effects two-way ANOVA. **(A–F)** Genotypes are *Is-GAL4, UAS-myrGCaMP6s/UAS-tdTom::Sec61β*, in either a *WT*, or *Rtnl1¹⁸*, *Rtnl1¹⁸/Rtnl1¹*, or *WT/Rtnl1¹⁸* background.

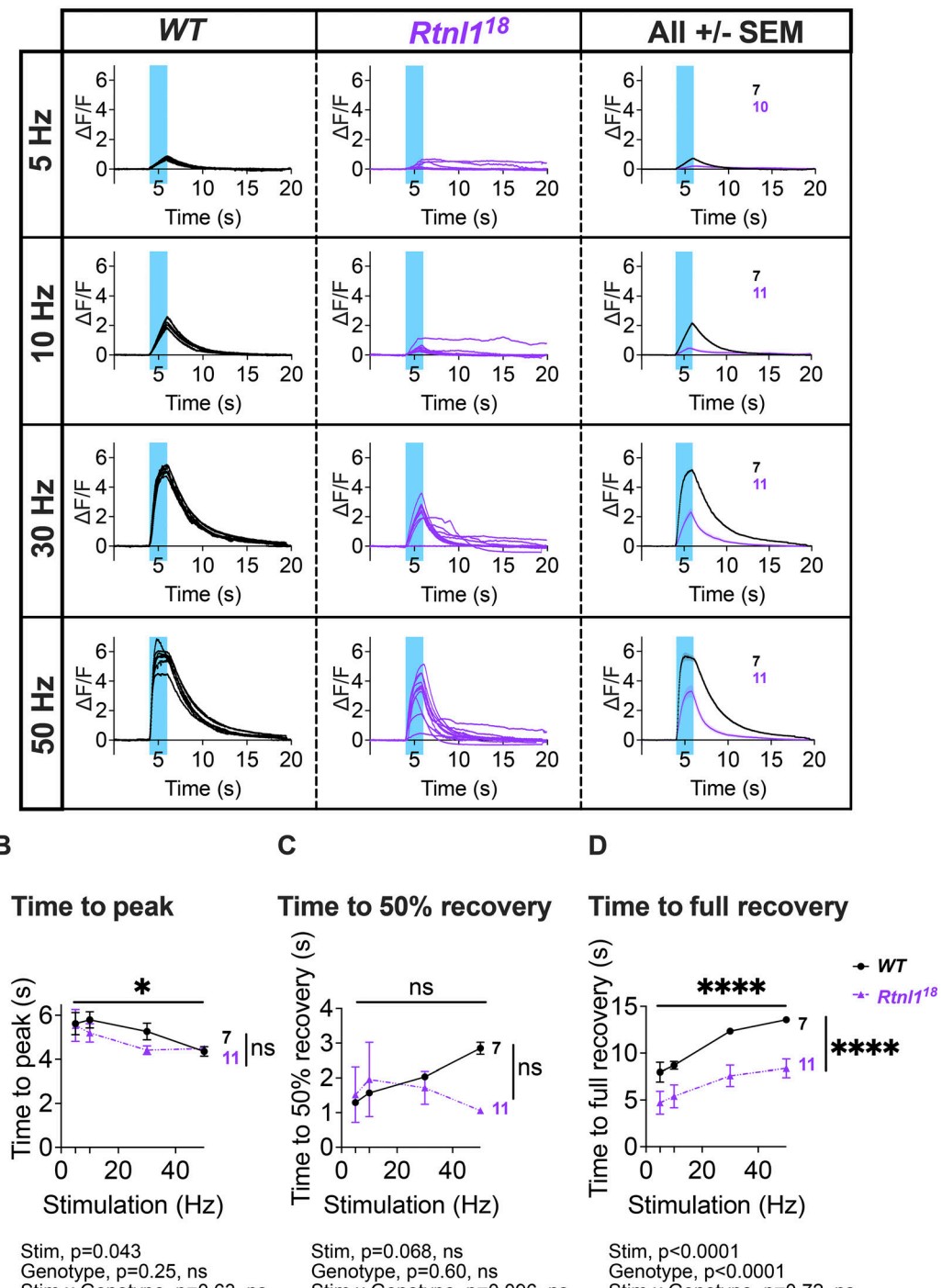

Figure S10. **Rtnl1 loss decreases cytosolic Ca²⁺ handling in Type Ib boutons.** (Extended data from Fig. 5.) **(A)** Impact of Rtnl1 loss of function on peak evoked cytosolic Ca²⁺. Plots show all single time traces and mean ± SEM of time traces in both genotypes for the four stimulation frequencies tested. Sample size (larvae) is indicated within the plot for each genotype. **(B and C)** Rtnl1 loss of function does not affect time to peak cytosolic Ca²⁺, or time to 50% recovery. **(D)** Rtnl1 loss of function decreases time to 100% recovery in every stimulation frequency tested. In B–D, plots show mean ± SEM of every genotype for each stimulation frequency tested; sample size (larvae) is indicated within the plot for each genotype. For each larva, we analyzed responses from a 20-s recording from one muscle 1 NMJ between segments A4-A6. Datapoints represent the time to peak ΔF/F and 50% or 100% recovery. Comparisons were analyzed with a mixed-effects two-way ANOVA. Genotypes are *Ib-GAL4, UAS-myrGCaMP6s/UAS-tdTom::Sec61β*, in either a *WT* or *Rtnl1¹⁸* background.

**A**

Evoked ER calcium response, all trials, Type Is boutons

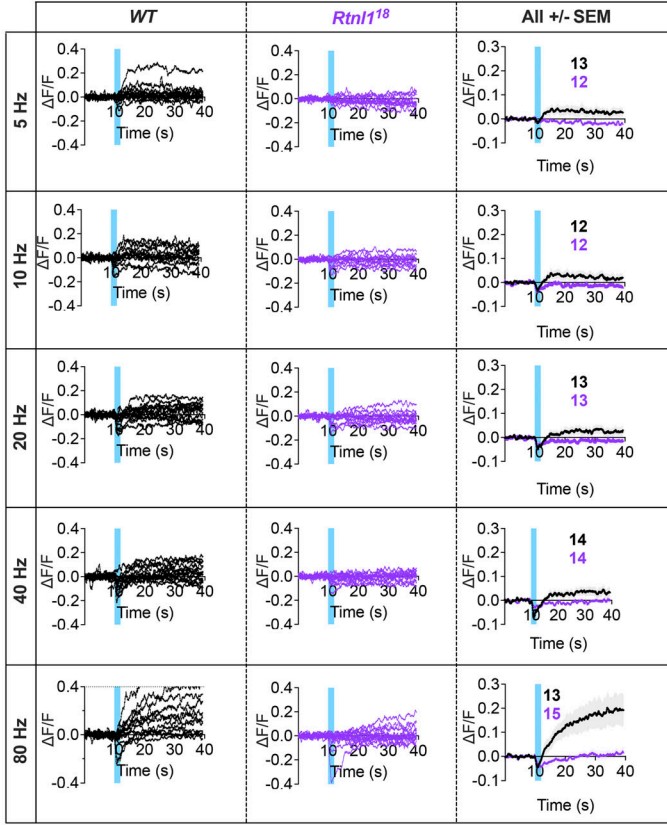

**B**

Evoked ER calcium response, all trials, Type Ib boutons

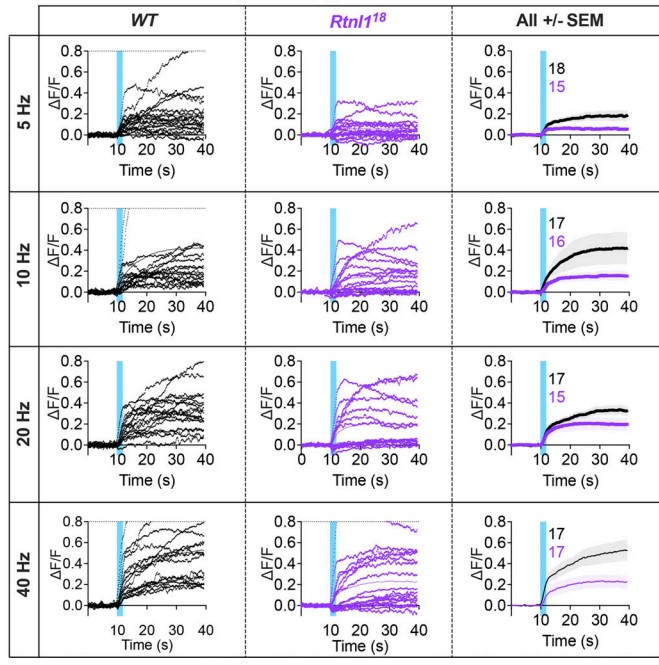

Figure S11. **Rtnl1 loss decreases ER Ca²⁺ handling in Type Is and Ib boutons.** (Extended data from Figs. 6 and 7.) **(A)** Impact of Rtnl1 loss-of-function on evoked ER Ca²⁺ in Is boutons. Plots show all single time traces and mean ± SEM time traces in both genotypes for the five stimulation frequencies tested. Sample size (larvae) is indicated within the plot for each genotype. Genotypes are *Is-GAL4, UAS-ER-GCaMP6-210/UAS-tdTom::Sec61β*, in either a *WT* or *Rtnl1¹⁸* background. **(B)** Impact of Rtnl1 loss of function on evoked ER Ca²⁺ in Ib boutons, plotted as in A. Genotypes are *Ib-GAL4, UAS-ER-GCaMP6-210/UAS-tdTom:: Sec61β*, in either a *WT* or *Rtnl1¹⁸* background.

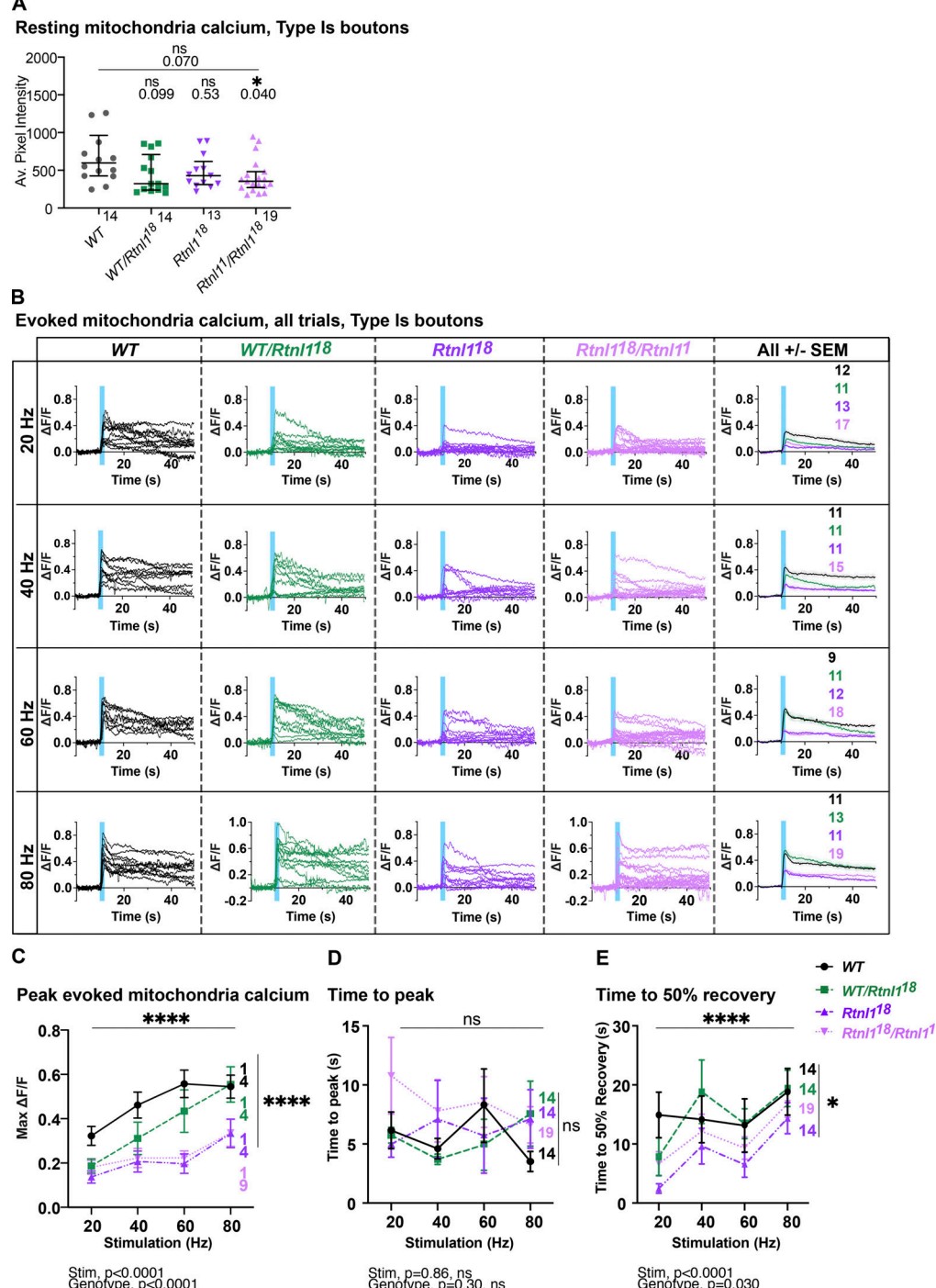

**Figure S12. Rtnl1 loss decreases mitochondrial Ca²⁺ handling in Type Is boutons.** (Extended data from Fig. 8.) **(A)** Loss of Rtnl1 does not affect resting CEPIA3mt fluorescence. Although *Rtnl1¹⁸/Rtnl1¹* flies show a marginally significant decrease in CEPIA3mt fluorescence, this effect is not found in a homozygous *Rtnl1¹⁸* background, indicating that loss of Rtnl1 does not affect resting mitochondrial Ca²⁺. Plot shows individual larval datapoints and median ± interquartile ranges; sample size (larvae) is indicated within the plot for each genotype. Comparisons were performed using Kruskal–Wallis tests. **(B)** Impact of Rtnl1 loss-of-function on peak evoked mitochondrial Ca²⁺. Plots show all single time traces and mean ± SEM time traces in every genotype for four stimulation frequencies tested. Sample size (larvae) is indicated within the plot for each genotype. **(C)** Impact of Rtnl1 loss-of-function on peak evoked mitochondrial Ca²⁺. Plots show mean ± SEM of every genotype for each stimulation frequency tested; sample size (larvae) is indicated within the plot for each genotype. For each larva, responses from a 50-s recording from one muscle 1 NMJ between A4-A6 segments were analyzed. Comparisons were analyzed with a mixed-effects two-way ANOVA. **(D)** Rtnl1 loss did not affect time to peak mitochondria Ca²⁺. Data analyzed as in C. **(E)** Impact of Rtnl1 loss on time to 50% recovery of mitochondria Ca²⁺. Rtnl1 loss in both *Rtnl1¹⁸* and *Rtnl1¹⁸/Rtnl1¹* backgrounds slightly decreases time to 50% recovery when compared to *WT* and *WT/Rtnl1¹⁸* backgrounds, indicating that mitochondria in Rtnl1 mutants lose their Ca²⁺ slightly faster than in *WT*. Data analyzed as in C. In D and E, datapoints represent the time to peak ΔF/F and 50% or 100% recovery. Genotypes are *Is-GAL4, UAS-CEPIA3mt/UAS-tdTom::Sec61β*, in either a *WT*, or *Rtnl1¹⁸*, *Rtnl1¹⁸/Rtnl1¹*, or *WT/Rtnl1¹⁸* background.

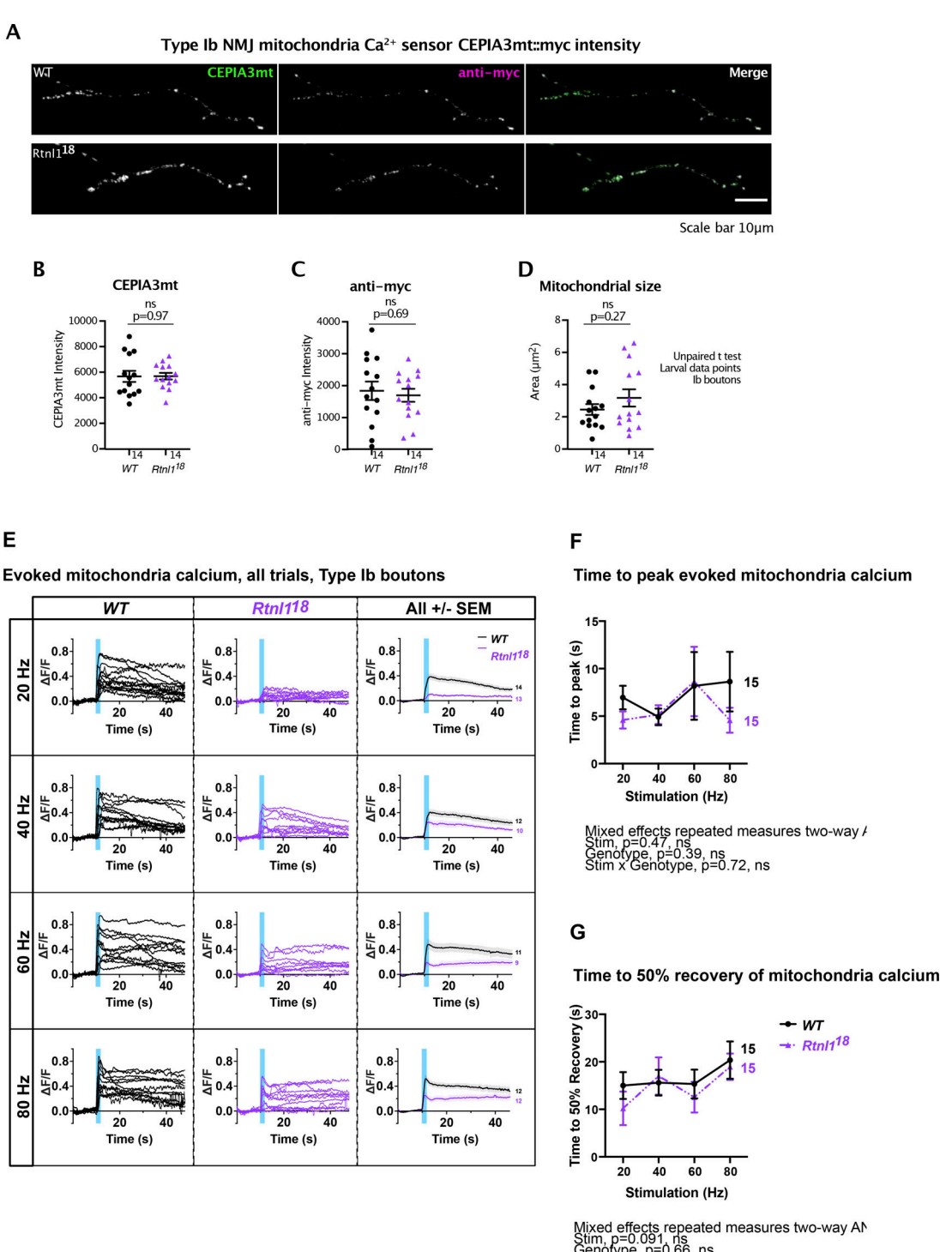

Figure S13. **Effect of Rtnl1 loss on mitochondrial Ca²⁺ handling in Type Ib boutons.** (Extended data from Fig. 8.) **(A)** Panels show mitochondrial CEPIA3mt fluorescence and anti-myc signal of CEPIA3mt::myc in typical examples of *WT* and *Rtnl1¹⁸* muscle 1, Type Ib postsynaptic terminals. **(B–D)** Rtnl1 loss-of-function does not impact CEPIA3mt fluorescence intensity (B), anti-myc signal intensity (C) of CEPIA3mt::myc, or mitochondrial size (D). Plots shows individual larval datapoints and mean ± SEM; sample size (larvae) is indicated within the plot for each genotype. For each larva, all mitochondria from several muscle 1 NMJs between A2-A6 segments were analyzed, and each mean larval value is shown as a datapoint. Pairwise comparisons were performed using Student's *t* tests. Genotypes are *Ib-GAL4, CEPIA3mt::myc*, in either a *WT* or *Rtnl1¹⁸* background. **(E)** Impact of Rtnl1 loss-of-function on peak evoked mitochondria Ca²⁺. Plots show all single time traces and mean ± SEM time traces in both genotypes for the four stimulation frequencies tested. Sample size (larvae) is indicated within the plot for each genotype. **(F and G)** Rtnl1 loss of function does not affect time to peak mitochondria Ca²⁺ or (G) time to 50% recovery of mitochondria Ca²⁺. Plots show mean ± SEM of every genotype for each stimulation frequency tested; sample size (larvae) is indicated within the plot for each genotype. For each larva, responses from a 50-s recording from one muscle 1 NMJ between A4-A6 segments were analyzed. Datapoints represent the time between stimulation and peak ΔF/F or peak ΔF/F and half recovery. Comparisons were analyzed with a mixed-effects two-way ANOVA. Genotypes are *Ib-GAL4, UAS-CEPIA3mt::myc/UAS-tdTom::Sec61β*, in either a *WT* or *Rtnl1¹⁸* background.

Video 1.  **WT postsynaptic Ca²⁺ response to low-frequency stimulation.** Evoked postsynaptic Ca²⁺ response to a 1 Hz stimulation. Genotype is *WT; Is-GAL4, Mhc-SynapGCaMP6f/UAS-tdTom::Sec61β*. In this and in all subsequent videos, Ca²⁺ sensor GCaMP is in green, ER marker tdTom::Sec61β in magenta; time is shown in seconds, and stimulation period (when present) as "STIM"; scale bar, 10 µm.

Video 2.  ***Rtnl1* mutant postsynaptic Ca²⁺ response to low-frequency stimulation.** Evoked postsynaptic Ca²⁺ response to a 1 Hz stimulation. Genotype is *Rtnl1¹⁸; Is-GAL4, Mhc-SynapGCaMP6f/UAS-tdTom::Sec61β*.

Video 3.  **WT postsynaptic miniature Ca²⁺ events.** Spontaneous postsynaptic Ca²⁺ at rest. Genotype is *WT; Is-GAL4, Mhc-SynapGCaMP6f/UAS-tdTom::Sec61β*.

Video 4.  ***Rtnl1* mutant postsynaptic miniature Ca²⁺ events.** Spontaneous postsynaptic Ca²⁺ at rest. Genotype is *Rtnl1¹⁸; Is-GAL4, Mhc-SynapGCaMP6f/UAS-tdTom::Sec61β*.

Video 5.  **WT cytosolic Ca²⁺ response to stimulation.** Evoked cytosolic Ca²⁺ response to a 30 Hz stimulation. Genotype is *WT; Ib-GAL4, UAS-myr::GCaMP6s/UAS-tdTom::Sec61β*.

Video 6.  ***Rtnl1* mutant cytosolic Ca²⁺ response to stimulation.** Evoked cytosolic Ca²⁺ response to a 30 Hz stimulation. Genotype is *Rtnl1¹⁸; Ib-GAL4, UAS-myr::GCaMP6s/UAS-tdTom::Sec61β*.

Video 7.  **WT ER Ca²⁺ response to stimulation in Is boutons.** Evoked ER Ca²⁺ response to a 40 Hz stimulation. Genotype is *WT; Is-GAL4, UAS-ER-GCaMP6-210/UAS-tdTom::Sec61β*.

Video 8.  ***Rtnl1* mutant ER Ca²⁺ response to stimulation in Is boutons.** Evoked ER Ca²⁺ response to a 40 Hz stimulation. Genotype is *Rtnl1¹⁸; Is-GAL4, UAS-ER-GCaMP6-210/UAS-tdTom::Sec61β*.

Video 9.  **WT ER Ca²⁺ response to stimulation in Ib boutons.** Evoked ER Ca²⁺ response to a 40 Hz stimulation. Genotype is *WT; Ib-GAL4, UAS-ER-GCaMP6-210/UAS-tdTom::Sec61β*.

Video 10.  ***Rtnl1* mutant ER Ca²⁺ response to stimulation in Ib boutons.** Evoked ER Ca²⁺ response to a 40 Hz stimulation. We provided two video versions, the first with the brightness comparable to the WT Video 9, Genotype is *Rtnl1¹⁸; Ib-GAL4, UAS-ER-GCaMP6-210/UAS-tdTom::Sec61β*.

Video 11.  ***Rtnl1* mutant ER Ca²⁺ response to stimulation in Ib boutons, brighter image.** A brighter version of Video 10.

Video 12.  **WT mitochondria Ca²⁺ response to stimulation in Ib boutons.** Evoked mitochondria Ca²⁺ response to a 40 Hz stimulation. Genotype is *WT; Ib-GAL4, UAS-CEPIA3mt/UAS-tdTom::Sec61β*.

Video 13.  ***Rtnl1* mutant mitochondria Ca²⁺ response to stimulation in Ib boutons.** Evoked mitochondria Ca²⁺ response to a 40 Hz stimulation. Genotype is *Rtnl1¹⁸; Ib-GAL4, UAS-CEPIA3mt/UAS-tdTom::Sec61β.*

**Provided online are Table S1, Table S2, and Table S3. Table S1 lists *Drosophila* stocks. Table S2 lists crosses used in this work. Table S3 shows reagents used in this work.**

