## [Peer Review File · The Journal of Cell Biology]

Drosophila SPG12 ortholog, reticulon-like 1, governs presynaptic ER organization and Ca²⁺ dynamics

Juan José Pérez-Moreno, Rebecca Smith, Megan Oliva, Filomena Gallo, Shainy Ojha, Karin Müller, and Cahir O'Kane

Corresponding Author(s): Cahir O'Kane, University of Cambridge and Juan José Pérez-Moreno, Institute of Biomedicine of Seville

Review Timeline:

Submission Date:	2021-12-22
Editorial Decision:	2022-02-03
Revision Received:	2023-01-01
Editorial Decision:	2023-02-08
Revision Received:	2023-02-21
Accepted:	2023-02-24

Monitoring Editor: Patrik Verstreken

Scientific Editor: Tim Spencer

Transaction Report:

DOI: <https://doi.org/10.1083/jcb.202112101>

February 3, 2022

Re: JCB manuscript #202112101

Prof. Cahir J O'Kane
University of Cambridge
Department of Genetics
Downing Street
Cambridge, England CB2 3EH
United Kingdom

Dear Prof. O'Kane,

Thank you for submitting your manuscript entitled "Spastic paraplegia protein ortholog, reticulon-like 1, governs presynaptic ER organization and Ca²⁺ dynamics". The manuscript has been evaluated by expert reviewers, whose reports are appended below. Unfortunately, after an assessment of the reviewer feedback, our editorial decision is against publication in JCB.

As you will see, the reviewers appreciate that your study presents findings that are potentially of interest to a wide readership. However, we agree with their concerns that the current degree of analysis and level of mechanistic insight need to be expanded for the paper to be suitable for JCB. The reviewers have provided thoughtful and constructive suggestions, which we hope you agree will improve your study. In particular, as requested by all three reviewers it is essential to provide EM data to further investigate the ER architecture as well as membrane contact sites. A more detailed investigation of the relationship between ER architecture and Ca²⁺ also must be attempted, though in depth mechanistic insight into the role of Rtnl1 in tubular ER formation can be the subject of future studies. Reframing your study as suggested by reviewer #2 will also help improve the readability for a broad audience by more clearly articulating the questions being addressed.

Therefore, although your manuscript is intriguing, it seems the points raised by the reviewers are more substantial than can be addressed in a typical revision period. If you wish to expedite publication of the current data, it may be best to pursue publication at another journal.

Given interest in the topic, we would be open to resubmission to JCB of a significantly revised and extended manuscript that fully addresses the reviewers' concerns as outlined and is subject to further peer-review. If you would like to resubmit this work to JCB, please contact the journal office to discuss an appeal of this decision or you may submit an appeal directly through our manuscript submission system. We request that you submit a detailed revision plan, which we may request feedback on from the reviewers, to help ensure that a revised study will adequately address their concerns. Please note that priority and novelty would be reassessed at resubmission.

Regardless of how you choose to proceed, we hope that the comments below will prove constructive as your work progresses. We would be happy to discuss the reviewer comments further once you've had a chance to consider the points raised in this letter. You can contact the journal office with any questions, cellbio@rockefeller.edu or call (212) 327-8588.

Thank you for thinking of JCB as an appropriate place to publish your work.

Sincerely,

Patrik Verstreken, PhD
Monitoring Editor

Andrea L. Marat, PhD
Senior Scientific Editor

Journal of Cell Biology

Reviewer #1 (Comments to the Authors (Required)):

Pérez-Moreno et al. in this Ms deal with the role of ER-shaping protein Rtnl1 and ER organization in presynaptic physiology. They generated a Rtnl1 null mutant fly, in addition to their earlier characterized Rtnl1-1 loss-of-function mutant (Hum Mol Genet. 2012). They show that loss of Rtnl1 decreases the amount of ER membrane proteins at the presynapse, while leaving luminal ER protein levels unaffected and suggest these changes are due to the depletion of (local) narrow ER tubules. They then use

their model to explore the roles of presynaptic tubular ER using a variety of calcium (Ca²⁺) sensors. While Rtnl1 loss does not affect resting levels of Ca²⁺, they observe decreases in evoked cytosolic Ca²⁺ responses, a direct measure for neurotransmitter release, as well as decreases in ER- and mitochondrial Ca²⁺ uptake. Consistent with the decreases in evoked presynaptic Ca²⁺, loss of Rtnl1 also reduced spontaneous vesicle release.

The overall findings of the paper are of potential interest to a wide readership, as it is relevant for studies of synaptic function and the HSP field as well. However, there are a number of concerns the authors must address, in particular the interpretation of protein changes and ER membrane architecture. I also believe the study as presented, lacks some novel insights or clarity into the mechanisms behind the synaptic Ca²⁺ changes.

General concerns

1) Although partial depletion of synaptic ER is shown with two different markers, both measurements rely on overexpression and protein distribution. From these confocal images the authors then draw rather strong conclusions; on the architecture/continuity of the ER network, the ER amount and volume, and the localization/changes in tubules vs cisternae. However, one would need higher resolution or EM to really show these ER network characteristics and changes. Furthermore, it is difficult to imagine what the proposed ER organization changes are. How can the amount of tubules change so drastically while keeping ER volume intact? Are there so many cisternae in the terminals? Again, EM might clarify this.

2) Rtnl1 loss also causes large changes in postsynaptic Ca²⁺ responses. This could imply changes in presynaptic release but it could also suggest changes in postsynaptic Ca²⁺ handling. Is the ER architecture normal in the postsynaptic muscle? In addition, UAS-Rtnl1::GFP seems to phenocopy/work as dominant negative in both loss of SEC61 and decreased evoked Ca²⁺ responses but how is this reflected in ER network morphology changes?

3) The authors nicely show changes in Ca²⁺ handling in ER, mitochondria and synaptic cytosol upon depletion of ER-protein Rtnl1, however, there is little novel insights into how the depletion of synaptic ER tubules can affect these parameters or how they are related. Regarding the decrease in evoked ER Ca²⁺ fluxes, do the authors suggest this is solely dependent on the changes in ER-tubule number/surface or could, for instance, the ratio of Ca²⁺ release vs uptake channels be affected (as suggested by the fact that Ca²⁺ uptake but not Ca²⁺ release is affected)? In addition, it is unclear to me how the evoked-cytosolic Ca²⁺ fluxes (or reduced spontaneous release) can be changed, especially when authors do not expect an effect on SOCE. What is the potential contribution of ER, ER-calcium channels or SOCE here? Would increase/decrease in extracellular calcium enhance/reduce these phenotypes? Alternatively, how would inhibition/depletion of ER/PM Ca²⁺-channels or STIM affect these parameters?

Specific concerns

4) The authors refer to (Oliva et al 2020) to indicate the unchanged GCAMP6-210 on loss of Rtnl1 (page 4) but could also refer here to their own data in Fig. 6

5) The Ms describes no differences in resting cytosolic Ca²⁺, figure 5B however shows a significant increase of resting Ca²⁺ in Type Ib boutons.

6) The reduction in evoked responses could be (partially) rescued by UAS-Rtnl1::Ha (Fig. 5E), statistics between WT and Rtnl1 are however not indicated. Similar for Fig. 8E.

7) According to the result section, loss of Rtnl1 has no effect on the dynamics of Ca²⁺ responses (Figure S11D-F). In the figure however, while Rtnl1-18/Rtnl1-1 flies exhibit no changes, Rtnl1-18 flies seem to show significant changes in time to recover (faster, both in Ib and Is boutons). For clarity it might be better to only show and do the statistics on the WT and Rtnl1-18 flies (similar as in S12 B-D). Graph titles should be the same for similar experiments.

8) Fig 6C,D. The authors show in 6D that ER-Ca²⁺ release is not affected in Rtnl1-18 flies. The example traces in Figure 6C (and additionally in Fig 9) however, show a large decrease compared to WT.

9) Video 6: split channels for clarity

Other

E_{ijp} amplitude= Excitatory Junction Potential

S9 E,F: name of statistical test is behind graph

S11A,B: labels/stats are shifted

Reviewer #2 (Comments to the Authors (Required)):

The authors undertake a detailed characterization of presynaptic ER in *Drosophila* Rtnl1 mutants using the NMJ as a model system. Although several previous studies by the authors and others have examined Rtnl1, to what extent Ca²⁺ dynamics in distinct presynaptic compartments are altered following loss of this gene has not been well defined. Generating new targeted CRISPR alleles of Rtnl1 is an important addition to this previous work, and much of the data presented in Figures 1-4 are consistent and confirmatory of these previous studies. Although it is not clear precisely how the presynaptic ER is altered in Rtnl1 mutants, the authors interpret the changes presented to be due to specific reductions in "tubular ER", separate from "cisternal ER" structures at presynaptic terminals. For the rest of manuscript, the study becomes really one probing the role of tubular ER in regulating presynaptic Ca²⁺ dynamics, with the authors finding reductions in cytosolic, ER, and mitochondrial Ca²⁺ in Rtnl1 mutants.

The experiments presented in this study are of high quality, the writing is clear, and there are several intriguing findings. The authors are also to be commended for investigating presynaptic ER and related Ca²⁺ dynamics separately at Is vs Ib terminals, distinct neuronal subtypes that have largely been ignored by the field. The author's finding that the tubular ER network, if that is indeed what is specifically perturbed at presynaptic terminals in Rtnl1 mutants, has an underappreciated role in regulating Ca²⁺ dynamics at cytosolic, ER, and mitochondrial compartments, changes which may contribute to defective synaptic transmission and even HSP disease etiology. However, much of the study is rather descriptive, lacking a mechanistic hypothesis for why tubular ER is selectively disrupted in Rtnl1 mutants and what the role of this structure is, distinct from cisternal ER, in regulating presynaptic Ca²⁺. There are some areas in which major improvements can be made (see below), but at this point the study is lacking in the significant conceptual advance that one would expect.

Major points:

1. Framing of the study: This study really uses Rtnl1 mutants as an opportunity to probe the role of tubular ER in presynaptic Ca²⁺ dynamics at distinct compartments. However, it wasn't until the Discussion that this motivation was clearly articulated, and important background information is missing from the Introduction. Why is it important to disentangle functions of tubular vs cisternal ER in presynaptic Ca²⁺ dynamics? What is known currently about the role of tubular ER regulating synaptic transmission? It is clear from the highly cited and discussed de Juan-Sanz 2017 paper that presynaptic ER plays a major role in modulating neurotransmitter release, and a specific mechanism is proposed in that paper as the authors discuss. How does the tubular vs cisternal ER fit into this previously published work, and what is learned from this current study? The answers to these fundamental questions needs to be addressed up front in the Introduction and further in the Discussion.

2. Rtnl1 and the impact on tubular ER: It would be of significant interest to know why Rtnl1 would be selectively needed for proper tubular ER formation. More importantly, it seems important to more fully define what aspects of the presynaptic ER network are perturbed in Rtnl1 mutants. Although this does not seem to be the central interest of the authors in this study, it is not well defined what particular aspects of presynaptic ER are perturbed by loss of Rtnl1. There are clearly reductions in presynaptic ER in Rtnl1 mutants, although it is not clear to what extent this is due specifically to what the authors call "tubular ER". In the absence of clear markers to define tubular vs cisternal ER, it would appear that other approaches not used in this study, such as serial EM reconstruction, would be necessary to really make this claim.

The changes in axonal ER morphology the authors present in Rtnl1 mutants are interesting and consistent with previous work, although it is not clear what the physiological importance is of the inter-boutonal changes and whether they indeed reflect selective "tubular ER" reductions. Nor is it apparent what the physiological function of tubular ER vs cisternal ER are to presynaptic Ca²⁺ physiology. For example, it is not clear how an apparent change in axonal ER (tubule depletion) would lower evoked cytosolic Ca²⁺ influx, particularly given the mechanisms defined by the de Juan-Sanz paper. From the data presented in this study, we don't learn much about Rtnl1 function beyond simply reducing tubular ER surface area, and having more insight into why loss of Rtnl1 would selectively disrupt tubular but not cisternal ER would really improve and extend the impact of this manuscript.

Beyond this phenotype or reduced tubular ER in Rtnl1 mutants, all of the Ca²⁺ imaging expts are simply different ways of showing less response due to reduced ER surface area, without a coherent model for what tubular ER, distinct from cisternal ER, is doing to regulate presynaptic Ca²⁺ dynamics.

3. MN-Is vs -Ib physiology: The authors do a great job of uncoupling ER changes and Ca²⁺ dynamics at MN-Is vs -Ib terminals, something that has been largely ignored by the field. It would be very interesting if the authors could show specialized functions for presynaptic ER in contributing to tonic vs phasic physiology at these distinct motor neuron subtypes. However, this doesn't seem to be the case, and no particularly notable phenotypic differences are reported.

The major approach missing from this current study is electrophysiology. The Ca²⁺ imaging does not add much and is focused on miniature transmission, when the major phenotype is reduced evoked release as reported in previous studies, which failed to separate evoked transmission from Is vs Ib. As the authors know, recent work combining optogenetic stimulation using the Is/Ib Gal4 drivers the authors identified is capable of recording transmission selectively at Is vs Ib (published in recent studies from Graeme Davis and Troy Littleton, among others). This should be done here and may lead to differential functions in presynaptic tubular ER in regulating release from the strong Is vs the weak Ib terminals.

Minor points:

1. Given the observed reduction in ER-PM contact sites reported in Rtnl1 mutants, it would seem of interest to characterize Esyt2 localization in Rtnl1 mutants and maybe to define Rtnl1::YFP localization in Esyt2 mutants. Tagged constructs and Drosophila mutants have been reported (Kikuma et al., 2017).

2. Ca imaging in Fig. 5 and other areas: Accurate estimates of resting Ca²⁺ cannot be obtained by simply measuring basal GCaMP fluorescence between NMJs. A ratiometric indicator is needed, either by dual loading of dyes or co-expression of GCaMP and a Ca²⁺ insensitive fluorophore like mCherry (see work by Greg MacLeod). There are even new ratiometric presynaptic indicators using the newest GCaMP (UAS-Syt::mScarlet::GCaMP8f) recently published by Dion Dickman's lab (Li et al., Science Advances 2021). Doing presynaptic Ca²⁺ imaging using one of these ratiometric approaches is necessary to estimate differences in resting Ca²⁺ levels between Is and Ib and how they are impacted in Rtnl1 mutants.

Similarly, using the newer GCaMP8 variants would provide single action potential resolution of differences in the Ca response to stimulation that is more easily interpreted compared to the 50 Hz stimulation shown in Fig. 5C.

Reviewer #3 (Comments to the Authors (Required)):

In their paper, Pérez-Moreno and co-workers present a mutant reticulon1 fly model that allows to study more selectively the role of presynaptic tubular ER. Deficient expression of this ER-shaping protein indeed results in a synaptic less elaborate ER-tubule network, whereas ER cisternae remain unaffected. Using compartment-specific Ca²⁺ indicators they further demonstrate that an altered ER architecture in the presynapse decreases activity-evoked Ca²⁺ fluxes in the ER, cytosol and mitochondria. As such, the authors demonstrate a more local role for tubular ER in synapse physiology and how this affects neurotransmission. Given that the fly reticulon1 is orthologous to the human reticulons, including RTN2, and as mutations in the RTN2 gene are causal to HSP, these findings are of potential interest in the underlying 'dying back' mechanism of distal axons, and as observed in neurodegeneration. The manuscript builds further on previous work from the group, exploring the role of reticulon1 in the axonal ER, and overall the experiments are well conducted and provide an original insight in connecting ER architecture with presynaptic Ca²⁺ handling. However, essentially a single approach (Ca²⁺ fluxes in different compartments) is used throughout the manuscript, and the data would become more conclusive if independent or orthogonal experiments were included. The provided data need more scrutiny and mechanistic detail to be considered for publication at this level. I can summarize the major concerns as follows.

- Membrane contact sites play an important role in the exchange of lipids and Ca²⁺ handling between associated subcellular compartments. With respect to lipid transfer in the light of mutant Rtn1, the authors limit the study to inquiring PIP2 levels. As these are negative, the remainder of the work is focused on Ca²⁺ handling. However, the authors in addition conclude that on the basis of their findings, lipid handling in general is not affected (also indirectly based on the notion that other features reflecting failed lipid transfer are not observed). Without further support, this remains to my opinion a too bold statement, not supported by the data. Could the authors explore at least one more lipid reporter to corroborate more convincingly their statement?

- Mutant Rtn1 results in less tubular ER, whereas ER cisternae appear unaffected. This is essentially concluded from the reduced fluorescence intensity of bona fide (axonal) ER markers. The authors go even a step further and conclude that regions with essentially no visible staining represent thinned axonal ER tubules. E.g. in figure 3B, arrowheads point in both WT and mutant Rtn1 to gaps in the staining: from these (representative) pictures it is for this reviewer impossible to conclude that the fluorescence intensity is reduced. If reduced, can the authors exclude that in mutant Rtn1 neurons, the axonal/presynaptic ER is not more fragmented (instead of thinned tubular ER)? Likewise for the presynaptic areas: although the fluorescence intensity is reduced, it is difficult to deduce from the figures that actually the presynaptic ER network is reduced (with less tubular ER) (as concluded in their model in figure 10). I assume the authors have in their institute access to super-resolution imaging to provide a better resolution for the ER architecture in the presynapse, or to perform ultrastructural data using TEM to demonstrate that indeed the ER network is significantly affected (overall reduced, less/thinner ER tubules,...).

- The authors base a major part of their conclusions on an altered ER-PM communication, suggesting alterations in membrane contact site functioning. However, the only direct support is a reduced detection of STIM in motor neuron axons. The authors should include a more detailed analysis that morphologically the ER-PM MCSs are affected (altered size, number, frequency) using more advanced imaging with ER-PM MCSs marker proteins.

- With respect to the observed reduced Ca²⁺ uptake by the ER, the authors do not scrutinize the source of the Ca²⁺. Given the role of SOCE in re-filling ER Ca²⁺ stores, this lead should be investigated to provide more mechanistic insights in the observed reduced Ca²⁺ fluxes. A similar remark concerns the mitochondrial Ca²⁺ flux: herein also ER-mito MCSs may play a significant role. The authors might consider exploring specific MCS tethers to test whether indeed these contact sites are affected.

Likewise, data could be supplemented with TEM.

- The authors focus on the role of Rtn1 in axonal ER. For this reviewer it is not clear whether Rtn1 is uniquely expressed in axons, and for instance not in dendritic arbors: if so, mutant Rtn1 could generate combined defects in axonal and dendritic Ca²⁺ handling (maybe more clear in CNS neurons?).

Minor comments:

- p.6: "Rtn1 loss again did not alter resting ER luminal GCaMP fluorescence relative to WT (Fig. 7A-B), but often caused the initial rapid Ca²⁺ release to appear (Fig. 7C-D; Supplementary Fig. S13B), at every stimulation frequency tested (Fig. 7E),...". For this reviewer, it is not clear what is meant with 'the initial rapid Ca²⁺ release to appear'. Is there no word missing?

- Discussion, first sentence: the whole first sentence doesn't read well and words are missing (or at least the message is not getting through). A bit further: '... presynaptic ER is more dependent on the function of Rtn1...'

- Supplemental data are fragmented over an excessive number of figures, which makes it for the reader more difficult to correlate with the results. I suggest to reduce the number and combine sets of data related to the same sub-theme (or figure in

the main text) in the same supplemental figure.

- Video 5: it is not clear what this video is showing. The situation in WT or mutant flies. In any event, comparative video's of the different genotypes should be included to appreciate the differences. Similar comments to video 6 to 8: live imaging data should be presented in pairs (wt vs mutant), even if there are no significant differences.

We thank the reviewers for their helpful comments on the manuscript.

Reviewer #1:

The overall findings of the paper are of potential interest to a wide readership, as it is relevant for studies of synaptic function and the HSP field as well. However, there are a number of concerns the authors must address, in particular the interpretation of protein changes and ER membrane architecture. I also believe the study as presented, lacks some novel insights or clarity into the mechanisms behind the synaptic Ca²⁺ changes.

General concerns

1) "One would need higher resolution or EM to really show these ER network characteristics and changes. Furthermore, it is difficult to imagine what the proposed ER organization changes are. How can the amount of tubules change so drastically while keeping ER volume intact? Are there so many cisternae in the terminals? Again, EM might clarify this."

"EM might clarify this"

We have in fact attempted serial EM on NMJ boutons, using the ATUMtome approach that we previously used very productively for axons. We were unable to follow ER tubules reliably in boutons due to the large excess of synaptic vesicles and the difficulty of distinguishing them from ER tubules due to their overlapping range of diameters and the staining of both types of membrane by ROTO staining. Because of this, we can only partially reconstruct the ER network in synaptic terminals, and have not been able to reliably compare wildtype with *Rtn1* mutant ER network at EM level. However, we have been able to identify both ER cisternae and connected tubules at terminals (Fig. 3), supporting our interpretation that the "beads on a string" that we see with GFP::HDEL are cisternae linked by tubules. Also, Wu et al (PNAS, 2017, cited) have already performed FIB-SEM (which has a similar x-y resolution to ours, but a better z-resolution) on a different source of neurons, and their findings already address some of the reviewer's points, and support our interpretation of the markers we use for confocal microscopy. However, "How can the amount of tubules change so drastically while keeping ER volume intact?" This is consistent with the tiny diameters of axonal ER tubules observed by ourselves, and by Terasaki (2018) and Wu et al (2017) - since the lumen diameter is so small that it is hard to estimate it even with EM, it is not surprising that most of the presynaptic ER luminal volume would be in cisternae, which have a larger volume, and which we also observe in axons by EM (Yalcin et al 2017).

"Are there so many cisternae in the terminals?"

Presynaptic ER stores (i.e. thapsigargin-sensitive) are well documented in neurophysiology, and will require a certain volume of lumen. Wu et al (2017) have indeed observed many ER cisternae in presynaptic terminals, also joined by narrow tubules. We have now added serial EM of a wildtype NMJ bouton (Fig. 3A), showing a high abundance of cisternae.

Regarding our use of markers, the luminal markers HDEL and GCaMP both give consistent "beads on a string" labeling, and this is consistent both with our previous EM observations in axons (Yalcin et al, 2017), and those of Wu et al (2017), of occasional irregularly shaped cisternae, joined by very narrow tubules. The continuity of our two membrane markers (Sturkopf and Sec61b), and the fact that they show less contrast between likely tubular and likely cisternae, are also consistent with our EM observations.

We have highlighted the presynaptic cisternae of Wu et al in our Introduction, and refer more explicitly to both their and our EM data when interpreting the localisation of our markers.

2) “Rtnl1 loss also causes large changes in postsynaptic Ca²⁺ responses. This could imply changes in presynaptic release but it could also suggest changes in postsynaptic Ca²⁺ handling. Is the ER architecture normal in the postsynaptic muscle?”

Specific expression of UAS-Rtnl1::HA in motor neurons fully rescues both presynaptic ER distribution (Fig. 2C) and presynaptic release in Rtnl1 mutants (Fig. 5E and Fig. 8E), demonstrating that the reported synaptic dysfunction is due to presynaptic ER defects. Furthermore we find no change in mini amplitude (Fig S7F), implying that postsynaptic sensitivity is unchanged.

“UAS-Rtnl1::GFP seems to phenocopy/work as dominant negative in both loss of SEC61 and decreased evoked Ca²⁺ responses but how is this reflected in ER network morphology changes?”
We have not tested for this since we do not believe this is central to our paper - we use our existing data as a way of assessing whether this is a suitable rescue construct and have concluded that it is not.

3) “The authors nicely show changes in Ca²⁺ handling in ER, mitochondria and synaptic cytosol upon depletion of ER-protein Rtnl1, however, there is little novel insights into how the depletion of synaptic ER tubules can affect these parameters or how they are related. Regarding the decrease in evoked ER Ca²⁺ fluxes, do the authors suggest this is solely dependent on the changes in ER-tubule number/surface or could, for instance, the ratio of Ca²⁺ release vs uptake channels be affected (as suggested by the fact that Ca²⁺ uptake but not Ca²⁺ release is affected)? In addition, it is unclear to me how the evoked-cytosolic Ca²⁺ fluxes (or reduced spontaneous release) can be changed, especially when authors do not expect an effect on SOCE. What is the potential contribution of ER, ER-calcium channels or SOCE here? Would increase/decrease in extracellular calcium enhance/reduce these phenotypes? Alternatively, how would inhibition/depletion of ER/PM Ca²⁺-channels or STIM affect these parameters?”

We addressed these questions by following up on our previous observation that STIM levels (like other ER markers) were also decreased in Rtnl1 mutants (Fig 4 A-B) by hypothesising that the effect of lowered ER tubule levels on presynaptic Ca²⁺ handling was due to lower levels of STIM signaling from ER to effectors (known or unknown) in the plasma membrane. This hypothesis predicts that the Rtnl1 mutant phenotypes in calcium handling would be rescued by STIM overexpression - and in fact we found this to be the case (Fig 5E), and that STIM overexpression alone has no effect on this phenotype, and that it thus abolishes this aspect of the Rtnl1 mutant phenotype.

Specific concerns

4) “The authors refer to (Oliva et al 2020) to indicate the unchanged GCAMP6-210 on loss of Rtnl1 (page 4) but could also refer here to their own data in Fig. 6.”

On page 4, we only refer to “Oliva et al., 2020” because we are still justifying why we performed the experiments shown in Fig. 3. We do not reference Fig. 6 since that data was subsequently performed and cannot be referred before the previous Figures are presented.

5) “The Ms describes no differences in resting cytosolic Ca²⁺, figure 5B however shows a significant increase of resting Ca²⁺ in Type Ib boutons.”

We now properly assess resting cytosolic Ca²⁺ in Type Ib boutons by using a ratiometric cytosolic Ca²⁺ sensor (UAS-tdTom-p2a-GCaMP56). We normalized GCaMP signal to the sensor levels (tdTom signal), finding no differences between WT and Rtnl1 mutant. This is shown in Fig. 5B, replacing the panel of the previous version.

6) “The reduction in evoked responses could be (partially) rescued by UAS-Rntl1::Ha (Fig. 5E), statistics between WT and Rntl1 are however not indicated. Similar for Fig. 8E.”

This is now shown in both Figures

7) “According to the result section, loss of Rtnl1 has no effect on the dynamics of Ca²⁺ responses (Figure S11D-F). In the figure however, while Rtnl1-18/Rtnl1-1 flies exhibit no changes, Rtnl1-18 flies seem to show significant changes in time to recover (faster, both in Ib and Is boutons). For clarity it might be better to only show and do the statistics on the WT and Rtnl1-18 flies (similar as in S12 B-D). Graph titles should be the same for similar experiments.”

This has been fixed

8) “Fig 6C,D. The authors show in 6D that ER-Ca²⁺ release is not affected in Rtnl-18 flies. The example traces in Figure 6C (and additionally in Fig 9) however, show a large decrease compared to WT.”

Yes, our graph in Fig 6C was perhaps a little confusing. There is a slight apparent decrease in ER Ca²⁺ release in Rtnl1 mutants - however, the ANOVA analysis in Fig 6D shows that there is no significant effect of genotype, and therefore that this apparent decrease is not significant. To remove this confusion, we now instead show recordings from 20Hz, where there is neither a real nor an apparent decrease in ER Ca²⁺ release.

9) “Video 6: split channels for clarity”

Brightness on (now) video 10 has been adjusted for clarity. ER in Is boutons are very narrow, so with the brightness settings it is impossible to get clear videos comparing WT and Rtnl1 mutants – we have therefore included a “brightness up” version of the Rtnl1 mutant response so it is easy to see the release of calcium in the cisternae to the right of the video with stimulation. We prefer this to splitting channels, since splitting channels on all multichannel videos would generate several new supplementary video files.

Other

“Ejp amplitude= Excitatory Junction Potential”

The abbreviation has now been defined.

“S9 E,F: name of statistical test is behind graph”

This is no longer behind the graph.

“S11A,B: labels/stats are shifted”

Labels and stats are no longer shifted.

Reviewer #2:

The experiments presented in this study are of high quality, the writing is clear, and there are several intriguing findings. The authors are also to be commended for investigating presynaptic ER and related Ca²⁺ dynamics separately at Is vs Ib terminals, distinct neuronal subtypes that

have largely been ignored by the field. The author's finding that the tubular ER network, if that is indeed what is specifically perturbed at presynaptic terminals in *Rtnl1* mutants, has an underappreciated role in regulating Ca^{2+} dynamics at cytosolic, ER, and mitochondrial compartments, changes which may contribute to defective synaptic transmission and even HSP disease etiology. However, much of the study is rather descriptive, lacking a mechanistic hypothesis for why tubular ER is selectively disrupted in *Rtnl1* mutants and what the role of this structure is, distinct from cisternal ER, in regulating presynaptic Ca^{2+} . There are some areas in which major improvements can be made (see below), but at this point the study is lacking in the significant conceptual advance that one would expect.

Major points:

1. "Framing of the study: This study really uses *Rtnl1* mutants as an opportunity to probe the role of tubular ER in presynaptic Ca^{2+} dynamics at distinct compartments. However, it wasn't until the Discussion that this motivation was clearly articulated, and important background information is missing from the Introduction. Why is it important to disentangle functions of tubular vs cisternal ER in presynaptic Ca^{2+} dynamics? What is known currently about the role of tubular ER regulating synaptic transmission? It is clear from the highly cited and discussed de Juan-Sanz 2017 paper that presynaptic ER plays a major role in modulating neurotransmitter release, and a specific mechanism is proposed in that paper as the authors discuss. How does the tubular vs cisternal ER fit into this previously published work, and what is learned from this current study? The answers to these fundamental questions needs to be addressed up front in the Introduction and further in the Discussion."

To address this point, we have now added new text in the 4th paragraph of the Introduction and updated the related text in the Discussion section (4th paragraph).

2. "*Rtnl1* and the impact on tubular ER: It would be of significant interest to know why *Rtnl1* would be selectively needed for proper tubular ER formation. More importantly, it seems important to more fully define what aspects of the presynaptic ER network are perturbed in *Rtnl1* mutants. Although this does not seem to be the central interest of the authors in this study, is not well defined what particular aspects of presynaptic ER are perturbed by loss of *Rtnl1*. There are clearly reductions in presynaptic ER in *Rtnl1* mutants, although it is not clear to what extent this is due specifically to what the authors call "tubular ER". In the absence of clear markers to define tubular vs cisternal ER, it would appear that other approaches not used in this study, such as serial EM reconstruction, would be necessary to really make this claim."

As requested, we tried to perform serial EM reconstruction, but we were unable to follow the ER into the pool of synaptic vesicles. Despite this, we identified connected ER cisternae and tubules at presynaptic terminals (Fig. 3A), supporting our interpretation that the "beads on a string" that we see with GFP::HDEL are cisternae linked by tubules. Therefore, we believe that the performed analysis of ER membrane markers and an ER luminal marker provide anatomical markers for a correct understanding of the morphological changes in presynaptic ER, showing the local surface and volume of the network. As explained in our response to Reviewer 1, the interpretation of tubules and cisternae is based also on published ultrastructural analysis of presynaptic ER. This is now highlighted in the 4th paragraph in the Introduction.

"The changes in axonal ER morphology the authors present in *Rtnl1* mutants are interesting and consistent with previous work, although it is not clear what the physiological importance is of the inter-boutonal changes and whether they indeed reflect selective "tubular ER" reductions. Nor is it apparent what the physiological function of tubular ER vs cisternal ER are to presynaptic Ca^{2+} physiology. For example, it is not clear how an apparent change in axonal ER (tubule depletion)

would lower evoked cytosolic Ca²⁺ influx, particularly given the mechanisms defined by the de Juan-Sanz paper. From the data presented in this study, we don't learn much about Rtnl1 function beyond simply reducing tubular ER surface area, and having more insight into why loss of Rtnl1 would selectively disrupt tubular but not cisternal ER would really improve and extend the impact of this manuscript.

Beyond this phenotype or reduced tubular ER in Rtnl1 mutants, all of the Ca²⁺ imaging expts are simply different ways of showing less response due to reduced ER surface area, without a coherent model for what tubular ER, distinct from cisternal ER, is doing to regulate presynaptic Ca²⁺ dynamics."

As it is highlighted in the Introduction, Reticulon proteins induce curvature of the ER membrane, mediating tubulation. This is well supported by the literature. Therefore, it was highly expected to see a particular impact of Rtnl1 loss on ER tubules and not on ER cisternae. We have now added explanatory text about the hypothetical roles of presynaptic ER tubules and cisternae (4th paragraph in the Introduction and 4th paragraph in the Discussion). These roles are supported by new data (Fig. 5E) that supports STIM-mediated ER-PM signaling as the mechanism of the model presented in Fig. 10.

3. "MN-1s vs -1b physiology: The authors do a great job of uncoupling ER changes and Ca²⁺ dynamics at MN-1s vs -1b terminals, something that has been largely ignored by the field. It would be very interesting if the authors could show specialized functions for presynaptic ER in contributing to tonic vs phasic physiology at these distinct motor neuron subtypes. However, this doesn't seem to be the case, and no particularly notable phenotypic differences are reported.

The major approach missing from this current study is electrophysiology. The Ca²⁺ imaging does not add much and is focused on miniature transmission, when the major phenotype is reduced evoked release as reported in previous studies, which failed to separate evoked transmission from 1s vs 1b. As the authors know, recent work combining optogenetic stimulation using the 1s/1b Gal4 drivers the authors identified is capable of recording transmission selectively at 1s vs 1b (published in recent studies from Graeme Davis and Troy Littleton, among others). This should be done here and may lead to differential functions in presynaptic tubular ER in regulating release from the strong 1s vs the weak 1b terminals."

We respectfully disagree that the calcium imaging doesn't add much - we show extensive comparisons of calcium fluxes in 1b and 1s terminals. We acknowledge that specific optogenetic stimulation of 1s or 1b would allow postsynaptic recordings at either bouton type, but argue that this would take the balance of the manuscript further away from understanding the role of presynaptic ER in presynaptic calcium fluxes, on which we already present extensive analyses.

Minor points:

1. "Given the observed reduction in ER-PM contact sites reported in Rtnl1 mutants, it would seem of interest to characterize Esyt2 localization in Rtnl1 mutants and maybe to define Rtnl1::YFP localization in Esyt2 mutants. Tagged constructs and Drosophila mutants have been reported (Kikuma et al., 2017)."

As reported in Kikuma et al., 2017, Esyt2 localization is quite homogenous along presynaptic ER at Drosophila NMJs, showing good colocalization with the general ER marker KDEL::GFP. Therefore, Esyt2 would not be useful to explore ER contact sites.

Regarding the converse experiment, the effect of Esyt2 loss on Rtnl1 localization, we argue that this is beyond the aim of this work. We have studied how changes in ER morphology affect its

function, but how ESyt2 might impact ER morphology (i.e., tubulation) would be a different biological problem.

2. "Ca imaging in Fig. 5 and other areas: Accurate estimates of resting Ca²⁺ cannot be obtained by simply measuring basal GCaMP fluorescence between NMJs. A ratiometric indicator is needed, either by dual loading of dyes or co-expression of GCaMP and a Ca²⁺ insensitive fluorophore like mCherry (see work by Greg MacLeod). There are even new ratiometric presynaptic indicator using the newest GCaMP (UAS-Syt::mScarlet::GCaMP8f) recently published by Dion Dickman's lab (Li et al., Science Advances 2021). Doing presynaptic Ca²⁺ imaging using one of these ratiometric approaches is necessary to estimate differences in resting Ca²⁺ levels between Is and Ib and how they are impacted in Rtnl1 mutants.

Similarly, using the newer GCaMP8 variants would provide single action potential resolution of differences in the Ca response to stimulation that is more easily interpreted compared to the 50 Hz stimulation shown in Fig. 5C."

We already presented this approach for mitochondria, using a combination of a calcium sensor and an epitope tag. We have now included new data using a ratiometric cytoplasmic Ca²⁺ sensor tagged with tdTom (UAS-tdTom-p2a-GCaMP56), validating that resting/basal Ca²⁺ levels are similar for WT and Rtnl1 mutants. Panel Fig. 5B has been updated to show this data.

Reviewer #3:

The manuscript builds further on previous work from the group, exploring the role of reticulon1 in the axonal ER, and overall the experiments are well conducted and provide an original insight in connecting ER architecture with presynaptic Ca²⁺ handling. However, essentially a single approach (Ca²⁺ fluxes in different compartments) is used throughout the manuscript, and the data would become more conclusive if independent or orthogonal experiments were included. The provided data need more scrutiny and mechanistic detail to be considered for publication at this level. I can summarize the major concerns as follows.

- "Membrane contact sites play an important role in the exchange of lipids and Ca²⁺ handling between associated subcellular compartments. With respect to lipid transfer in the light of mutant Rtnl1, the authors limit the study to inquiring PIP2 levels. As these are negative, the remainder of the work is focused on Ca²⁺ handling. However, the authors in addition conclude that on the basis of their findings, lipid handling in general is not affected (also indirectly based on the notion that other features reflecting failed lipid transfer are not observed). Without further support, this remains to my opinion a too bold statement, not supported by the data. Could the authors explore at least one more lipid reporter to corroborate more convincingly their statement?"

We agree with the reviewer that the analysis of PIP2 levels is not enough to conclude that Rtnl1 loss is not affecting any lipid transfer, and in coherence we already indicated the following in the Results section:

"Although we cannot exclude that other ER-PM contact-site-dependent lipids are affected in Rtnl1 mutants, a decrease in presynaptic tubular ER does not necessarily impact lipid homeostasis."

and in the Discussion section:

"Although we do not observe any PI(4,5)P2 alterations in Rtnl1 mutants, it is still possible that other ER-dependent lipids might be affected"

We, therefore, think we indicate fair conclusions considering our data, which supports why, at these early stages of our work, we decided to expand our analysis towards Ca²⁺ exchange instead of lipid transfer.

- “Mutant Rtn1 results in less tubular ER, whereas ER cisternae appear unaffected. This is essentially concluded from the reduced fluorescence intensity of bona fide (axonal) ER markers. The authors go even a step further and conclude that regions with essentially no visible staining represent thinned axonal ER tubules. E.g. in figure 3B, arrowheads point in both WT and mutant Rtn1 to gaps in the staining: from these (representative) pictures it is for this reviewer impossible to conclude that the fluorescence intensity is reduced. If reduced, can the authors exclude that in mutant Rtn1 neurons, the axonal/presynaptic ER is not more fragmented (instead of thinned tubular ER)? Likewise for the presynaptic areas: although the fluorescence intensity is reduced, it is difficult to deduce from the figures that actually the presynaptic ER network is reduced (with less tubular ER) (as concluded in their model in figure 10). I assume the authors have in their institute access to super-resolution imaging to provide a better resolution for the ER architecture in the presynapse, or to perform ultrastructural data using TEM to demonstrate that indeed the ER network is significantly affected (overall reduced, less/thinner ER tubules,...).”

We have added new panels in Supp. Fig. S5C to provide a better observation of those regions with low signal levels. We highlight those pixels with 0 intensity value. In addition, we already provided plot intensity profiles of Fig. 3B images to directly see the quantified fluorescence intensity along the NMJ (Fig. 3C), providing a more accurate approach than the comparison of the images by eye.

We are confidently able to exclude an increase of ER fragmentation in Rtn1 mutants, due to the consistent continuity observed with different ER membrane markers (Sec61B, CG9186 or STIM). Therefore, our conclusion is that the ER lumen is what might be discontinuous or quite narrow, but not the ER network. It is important to highlight that the apparent gaps observed with the luminal marker are also shown by the WT, indicating that these narrow regions are normal elements of the ER network.

While we think the confocal analysis of ER membrane markers and a ER luminal marker is solid enough to reach our conclusions, we tried to perform serial EM as requested by the reviewer. Although, as described in our reply to Reviewer 1, we were unable to identify the ER in the presence of surrounding synaptic vesicles, and thus to reliably characterize the impact of loss of Rtn1 at this level, we observed connected ER cisternae and tubules at presynaptic terminals (Fig. 3A), supporting our interpretation that the “beads on a string” that we see with GFP::HDEL are cisternae linked by tubules

- “The authors base a major part of their conclusions on an altered ER-PM communication, suggesting alterations in membrane contact site functioning. However, the only direct support is a reduced detection of STIM in motor neuron axons. The authors should include a more detailed analysis that morphologically the ER-PM MCSs are affected (altered size, number, frequency) using more advanced imaging with ER-PM MCSs marker proteins.”

As described in our response to Reviewer 1, we now provide data showing that STIM overexpression rescues presynaptic Ca²⁺ release (Fig. 5E). This data supports that altered ER-PM communication in Rtn1 mutants is causing the reported defects on synaptic function.

- “With respect to the observed reduced Ca²⁺ uptake by the ER, the authors do not scrutinize the source of the Ca²⁺. Given the role of SOCE in re-filling ER Ca²⁺ stores, this lead should be investigated to provide more mechanistic insights in the observed reduced Ca²⁺ fluxes. A similar

remark concerns the mitochondrial Ca²⁺ flux: herein also ER-mito MCSs may play a significant role. The authors might consider exploring specific MCS tethers to test whether indeed these contact sites are affected. Likewise, data could be supplemented with TEM.”

The new data showing rescue of presynaptic Ca²⁺ release by overexpressing STIM (Fig. 5E), suggests that extracellular Ca²⁺ uptake through ER-PM contacts is key to explain this phenotype in Rtnl1 mutants. While we might expect the existence of defects in ER contacts with other organelles, this is beyond the scope of this work.

- “The authors focus on the role of Rtnl1 in axonal ER. For this reviewer it is not clear whether Rtnl1 is uniquely expressed in axons, and for instance not in dendritic arbors: if so, mutant Rtnl1 could generate combined defects in axonal and dendritic Ca²⁺ handling (maybe more clear in CNS neurons?).”

We have not specifically explored Rtnl1 location at the dendritic arbors, but given its continuity along the axon and the cell body (Supp. Fig. S2B), we assume it is a general marker of the network. This is not relevant in our data, since Ca²⁺ recordings were performed after cutting the axons, leaving them without cell body and dendrites. However, we agree additional roles might be found for Rtnl1 and/or tubular ER at dendrites, with potential implications in spastic paraplegia disease.

Minor comments:

- “p.6: “Rtnl1 loss again did not alter resting ER luminal GCaMP fluorescence relative to WT (Fig. 7A-B), but often caused the initial rapid Ca²⁺ release to appear (Fig. 7C-D; Supplementary Fig. S13B), at every stimulation frequency tested (Fig. 7E),...”. For this reviewer, it is not clear what is meant with 'the initial rapid Ca²⁺ release to appear'. Is there no word missing?”

We have revised the text to: “but often led to initial rapid Ca²⁺ release as in wildtype and mutant Type 1s boutons”

- “Discussion, first sentence: the whole first sentence doesn't read well and words are missing (or at least the message is not getting through). A bit further: ‘... presynaptic ER is more dependent on the function of Rtnl1...’”

We have modified this text to clarify the message

- “Supplemental data are fragmented over an excessive number of figures, which makes it for the reader more difficult to correlate with the results. I suggest to reduce the number and combine sets of data related to the same sub-theme (or figure in the main text) in the same supplemental figure.”

We have now combined some Supplementary Figures, reducing their number by 3.

- “Video 5: it is not clear what this video is showing. The situation in WT or mutant flies. In any event, comparative video's of the different genotypes should be included to appreciate the differences. Similar comments to video 6 to 8: live imaging data should be presented in pairs (wt vs mutant), even if there are no significant differences.”

There is now a WT and Rtnl1 example provided for each calcium sensor.

February 8, 2023

RE: JCB Manuscript #202112101R-A

Prof. Cahir J O'Kane
University of Cambridge
Department of Genetics
Downing Street
Cambridge, England CB2 3EH
United Kingdom

Dear Prof. O'Kane:

Thank you for submitting your revised manuscript entitled "Spastic paraplegia protein ortholog, reticulon-like 1, governs presynaptic ER organization and Ca²⁺ dynamics". The paper has now been seen again by the original reviewers, all of whom not recommend acceptance. Therefore, we would be happy to publish your paper in JCB pending final revisions necessary to meet our formatting guidelines (see details below).

****As you will see, reviewers #2 and #3 have a few minor comments that we would like for you to address in the final revision. In particular, reviewer #2 feels that generating a 3D rendering of the structures would be useful for readers (particularly the non-specialists). In addition, reviewer #3 feels that you may need to tone down your conclusion that "Rtnl1 loss results in a specific decrease of presynaptic ER tubules".**

Please be sure to provide a point-by-point rebuttal to these remaining reviewer comments along with your final revision.**

A. MANUSCRIPT ORGANIZATION AND FORMATTING:

- 1) Text limits: Character count for Articles and Tools is < 40,000, not including spaces. Count includes the abstract, introduction, results, discussion, and acknowledgments. Count does not include title page, materials and methods, figure legends, references, tables, or supplemental legends. You are below the limit at this time but please try not to exceed it when performing the final revisions, if possible.
- 2) Figures limits: Articles and Tools may have up to 10 main text figures.
- 3) Figure formatting: Scale bars must be present on all microscopy images, including inset magnifications. Molecular weight or nucleic acid size markers must be included on all gel electrophoresis.
- 4) Statistical analysis: Error bars on graphic representations of numerical data must be clearly described in the figure legend. The number of independent data points (n) represented in a graph must be indicated in the legend. Statistical methods should be explained in full in the materials and methods. For figures presenting pooled data the statistical measure should be defined in the figure legends. Please also be sure to indicate the statistical tests used in each of your experiments (both in the figure legend itself and in a separate methods section) as well as the parameters of the test (for example, if you ran a t-test, please indicate if it was one- or two-sided, etc.). Also, if you used parametric tests, please indicate if the data distribution was tested for normality (and if so, how). If not, you must state something to the effect that "Data distribution was assumed to be normal but this was not formally tested."
- 5) Materials and methods: Should be comprehensive and not simply reference a previous publication for details on how an experiment was performed. Please provide full descriptions (at least in brief) in the text for readers who may not have access to referenced manuscripts. The text should not refer to methods "...as previously described."
- 6) Please be sure to provide the sequences for all of your primers/oligos and RNAi constructs in the materials and methods. You must also indicate in the methods the source, species, and catalog numbers (where appropriate) for all of your antibodies.
- 7) Microscope image acquisition: The following information must be provided about the acquisition and processing of images:
 - a. Make and model of microscope
 - b. Type, magnification, and numerical aperture of the objective lenses

- c. Temperature
- d. imaging medium
- e. Fluorochromes
- f. Camera make and model
- g. Acquisition software
- h. Any software used for image processing subsequent to data acquisition. Please include details and types of operations involved (e.g., type of deconvolution, 3D reconstitutions, surface or volume rendering, gamma adjustments, etc.).

8) References: There is no limit to the number of references cited in a manuscript. References should be cited parenthetically in the text by author and year of publication. Abbreviate the names of journals according to PubMed.

****Please note that we do not allow a separate reference list in the supplementary materials. Thus, we ask that you remove the supplemental reference list and add any non-duplicated references into the main reference list.****

9) Supplemental materials: There are normally strict limits on the allowable amount of supplemental data. Article may usually have up to 5 supplemental figures. However, given the circumstances, we will be able to give you the extra space this time. While it may be necessary to add one more supplemental figure (the 3D rendering mentioned by reviewer #2), please do not add any more figures than this one.

A summary of all supplemental material (that is, in addition to the supplementary figure legends) should appear at the end of the Materials and methods section. Please see any recent JCB paper for an example of this.

****Also, as noted above, please remove the supplementary reference list and add any non-duplicated references to the main reference list.****

10) eTOC summary: A ~40-50 word summary that describes the context and significance of the findings for a general readership should be included on the title page. The statement should be written in the present tense and refer to the work in the third person. It should begin with "First author name(s) et al..." to match our preferred style.

11) Conflict of interest statement: JCB requires inclusion of a statement in the acknowledgements regarding competing financial interests. If no competing financial interests exist, please include the following statement: "The authors declare no competing financial interests." If competing interests are declared, please follow your statement of these competing interests with the following statement: "The authors declare no further competing financial interests."

12) A separate author contribution section is required following the Acknowledgments in all research manuscripts. All authors should be mentioned and designated by their first and middle initials and full surnames. We encourage use of the CRediT nomenclature (<https://casrai.org/credit/>).

13) ORCID IDs: ORCID IDs are unique identifiers allowing researchers to create a record of their various scholarly contributions in a single place. At resubmission of your final files, please consider providing an ORCID ID for as many contributing authors as possible.

14) Please note that JCB now requires authors to submit Source Data used to generate figures containing gels and Western blots with all revised manuscripts. This Source Data consists of fully uncropped and unprocessed images for each gel/blot displayed in the main and supplemental figures. Since your paper includes cropped gel and/or blot images, please be sure to provide one Source Data file for each figure that contains gels and/or blots along with your revised manuscript files. File names for Source Data figures should be alphanumeric without any spaces or special characters (i.e., SourceDataF#, where F# refers to the associated main figure number or SourceDataFS# for those associated with Supplementary figures). The lanes of the gels/blots should be labeled as they are in the associated figure, the place where cropping was applied should be marked (with a box), and molecular weight/size standards should be labeled wherever possible.

15) Journal of Cell Biology now requires a data availability statement for all research article submissions. These statements will be published in the article directly above the Acknowledgments. The statement should address all data underlying the research presented in the manuscript. Please visit the JCB instructions for authors for guidelines and examples of statements at (<https://rupress.org/jcb/pages/editorial-policies#data-availability-statement>).

B. FINAL FILES:

Thank you for your attention to these final processing requirements. Please revise and format the manuscript and upload materials within 7-14 days. If complications arising from measures taken to prevent the spread of COVID-19 will prevent you from meeting this deadline (e.g. if you cannot retrieve necessary files from your laboratory, etc.), please let us know and we can work with you to determine a suitable revision period.

Please contact the journal office with any questions, cellbio@rockefeller.edu.

Thank you for this interesting contribution, we look forward to publishing your paper in Journal of Cell Biology.

Sincerely,

Patrik Verstreken, PhD
Monitoring Editor
Journal of Cell Biology

Tim Spencer, PhD
Executive Editor
Journal of Cell Biology

Reviewer #1 (Comments to the Authors (Required)):

The authors make a reasonable attempt to respond to my previous critiques and the revised manuscript has been improved compared to the originally submitted version. My 2 main concerns were the lack of more detailed ER network studies and mechanistic insight into the relationship between ER network changes and presynaptic calcium levels. Both concerns were partially addressed. The authors have added an important experiment (STIM overexpression), to now show some mechanistic insight on the requirement of STIM and ER-PM contact sites in presynaptic calcium changes. They discuss that STIM might promote calcium entry via the coupling with calcium channels on the PM. The authors however, try little to further determine what is (or what is not) the effector at the plasma membrane nor do they show that extracellular Ca²⁺ uptake is indeed key to the observed changes. EM data has been incorporated to try to address the ER architecture that previously was only tackled by (confocal) light microscopy. That they were not able to provide more detail on ER architecture, and couldn't provide any data on ER network changes in the Rtnl1 mutant, is a pity. I understand that such imaging can be challenging, however, as a large part of the manuscript and figures is devoted to describing the change in ER-architecture, additional analysis using for instance targeted ER labeling such as HRP/APX/miniSOGs or super-resolution light microscopy techniques would strengthen the claims made.

Reviewer #2 (Comments to the Authors (Required)):

The authors have completed additional experiments, analyses, and textual revisions that have significantly improved the study in this revised manuscript. In particular, the serial EM reconstruction now presented in Fig. 3A help to define the ER cisternae and tubules more fully at presynaptic terminals, and the ratiometric cytoplasmic Ca²⁺ sensor presented in Fig. 5B helps to more accurately assess Ca²⁺ levels. I still think having shown input-specific electrophysiological recordings would have helped to define functional consequences at tonic vs phasic synapses, but it is understood these recordings are not trivial and not central to the current study. I only have a few minor comments for the authors to consider in what is an important study that will be of broad interest to researchers.

1. There appears to be something wrong with resolution of the WT NMJ shown in Fig. 2B.
2. Serial EM in Fig. 3A: It would be beneficial for readers to show a 3D rendering of the structures of interest; there are several free software programs available that can do this (with color coding of the organelles, etc).

Reviewer #3 (Comments to the Authors (Required)):

Pérez-Moreno and co-workers have revised their manuscript on the contribution of the ER tubular network on presynaptic ER organization and synapse function using calcium dynamics as the readout. With regard to my criticisms, the authors have done an attempt to provide higher resolution EM on the synaptic boutons to better resolve the ER architecture. Although a full 3D FIB-SEM seemed unfeasible, they provided serial sections that support the beads-on-a-string interpretation. It is a pity that no comparison is made between WT and mutant flies, but these data together with the confocal analysis of ER membrane and luminal markers make it sufficient solid. They also provided more support for the functional involvement of ER-PM MCSs by demonstrating that STIM OE rescues calcium release. Overall the authors have significantly improved the manuscript, including addressing several of the criticisms of the other reviewers. Nevertheless the overall conclusion that Rtnl1 loss results in a specific decrease of presynaptic ER tubules (p.12, line 6 of the discussion) remains a too strict interpretation. Given that ER markers are overall down along the axon as well as a more general effect of STIM signaling along axons (STIM-positive ER-PM MCSs function beyond the synaptic bouton, the role of Rtnl1 impacts the axonal and not only the presynaptic ER tubules.

We thank the reviewers for their comments, which have helped us to substantially improve the manuscript. Find below the response to the last minor comments by reviewers 2 and 3.

Reviewer #2 (Comments to the Authors (Required)):

1. There appears to be something wrong with resolution of the WT NMJ shown in Fig. 2B.

We have substituted these panels by new versions with higher resolution

2. Serial EM in Fig. 3A: It would be beneficial for readers to show a 3D rendering of the structures of interest; there are several free software programs available that can do this (with color coding of the organelles, etc).

We have added new panels in Fig. 3A showing a 3D rendering of the highlighted ER network from the EM images.

Reviewer #3 (Comments to the Authors (Required)):

Nevertheless the overall conclusion that *Rtnl1* loss results in a specific decrease of presynaptic ER tubules (p.12, line 6 of the discussion) remains a too strict interpretation. Given that ER markers are overall down along the axon as well as a more general effect of STIM signaling along axons (STIM-positive ER-PM MCSs function beyond the synaptic bouton, the role of *Rtnl1* impacts the axonal and not only the presynaptic ER tubules.

We have deleted the word “specific” from this part of the text, leaving more open our interpretation of the data.